# Towards Understanding Token Selection in Self-Attention: Successes and Pitfalls in Learning Random Walks

## Abstract

As a key component of the transformer architecture, the self-attention mechanism is known for its capability to perform token selection, which can often significantly enhance model performance. However, when and how self-attention can be trained to perform effective token selection remains poorly understood in theory. In this paper, we study the problem of using a single self-attention layer to learn random walks on circles. We theoretically demonstrate that, after training with gradient descent, the self-attention layer can successfully learn the Markov property of the random walk, and achieve optimal next-token prediction accuracy by focusing on the correct parent token. In addition, we also study the performance of a single self-attention layer in learning relatively simpler "deterministic walks" on circles. Surprisingly, in this case, our findings indicate that the self-attention model trained with gradient descent consistently yields next-token prediction accuracy no better than a random guess. This counter-intuitive observation that self-attention can learn random walks but struggles with deterministic walks reveals a potential issue in self-attention: when there are multiple highly informative tokens, self-attention may fail to properly utilize any of them.

## 1 Introduction

In recent years, transformers (Vaswani et al., 2017) have revolutionized many fields such as natural language processing, and have rapidly emerged as a key component in state-of-the-art deep learning models due to their ability to capture complex dependencies in data. At the heart of transformers lies the self-attention mechanism, which allows the model to assign different weights or importance to each input token based on its relevance to the context or task at hand. This process of assigning weights to tokens based on their "importance" can be seen as a form of token selection, since it determines which tokens contribute more significantly to the model's prediction. However, the exact mechanisms behind token selection and how it impacts model performance are still not well understood.

A line of recent works has studied token selection of the self-attention mechanism from different perspectives. Tarzanagh et al. (2023); Ataee Tarzanagh et al. (2023) propose an equivalence between the optimization dynamic of one self-attention layer and an SVM problem and prove the global convergence under certain assumptions. Li et al. (2024a) shows that when training a self-attention layer, the priority in token selection is determined by a directed graph extracted from the training data. Wang et al. (2024) demonstrates that transformer models can learn the sparse token selection task effectively while fully connected networks fail in the worst case. Li et al. (2024b) shows that a self-attention layer can be trained to perform proper token selection so that the model acts as a one-nearest neighbor classifier in context.

Several more recent works have also studied the performance of transformers in learning sequential data generated from Markov models or Bayesian network models. In these studies, token selection is also the key, as an ideal self-attention layer should properly select the token(s) that is/are the 'parent(s)' of the token to be predicted. Specifically, Makkuva et al. (2024) characterizes the loss landscape of a single-layer transformer and demonstrates the existence of global minima and bad local minima in learning Markovian data with vocabularies of size two. Ildiz et al. (2024) shows the

connection between a context-conditioned Markov chain and the self-attention mechanism. Nichani et al. (2024) studies the mechanism through which transformer models encode a specific causal structure in their representations for in-context learning.

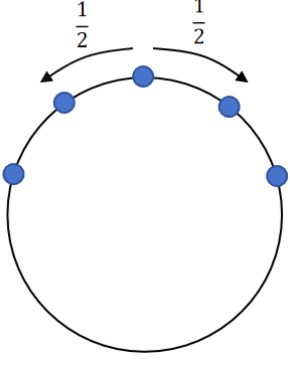 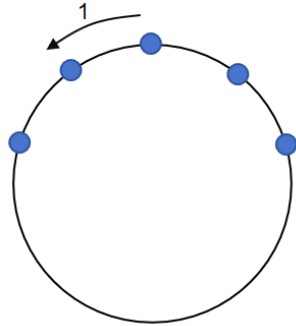

(a) Task 1: random walk      (b) Task 2: deterministic walk

Figure 1: Illustration of the tasks on learning random walks and deterministic walks. The first task involves a random walk along circular paths, where each step has an equal chance of moving clockwise or counterclockwise, as illustrated in (a). The second task involves a deterministic movement along circular paths, with the rule of always moving counterclockwise, as illustrated in (b).

In this paper, we introduce two simple case studies on how transformers learn sequential data. Specifically, we train a one-layer transformer model to predict sequences generated by "random walks" and "deterministic walks" on circles (see Figure 1 for an illustration). With a precise analysis on the training dynamics of gradient descent, we surprisingly find that the performance of the transformer on these two tasks can be drastically different.

The main contributions of this paper are summarized as follows:

- We theoretically demonstrate that, a one-layer transformer can be trained by gradient descent to optimally predict the next location of a random walk (illustrated in Figure 1(a)). In addition, our analysis also precisely reveals that the self-attention can be trained to select the correct token (the 'parent' token), and make prediction based on it. Our analysis sheds light on how the self-attention mechanism can adapt to sequential data patterns with proper token selection.

- We also show that, when learning to predict deterministic walks (illustrated in Figure 1(b)), the training of the same one-layer transformer model with any loss function and any step size will always fail, resulting in a transformer model whose performance is no better than a random guess. This result highlights a potential limitation of self-attention: when all tokens are equally 'informative', the self-attention mechanism may fail to utilize any of them.

- Simulations demonstrate that our theoretical characterization of one-layer transformers is accurate. Even when the trainable parameters of the transformer are initialized with Gaussian random values that do not satisfy our theoretical assumptions, we observe that the transformer struggles to learn deterministic walks, which aligns with our theory. Furthermore, motivated by our theories and explanations, we construct two simple question answering tasks in natural language processing (NLP) and successfully predict the performance of transformers on these tasks. This confirms the validity of our theory and highlights the insights provided by our study.

## 2 PROBLEM SETUP

In this section, we present our problem formulations, including the construction of the next-token prediction tasks we focus on, the transformer architecture with one self-attention layer, and the training algorithm.

### 2.1 RANDOM AND DETERMINISTIC WALKS ON CIRCLES

We study the procedures of random and deterministic walks on circles. Specifically, consider $K$ nodes (possible locations) that are arranged on a circle so that each node has two neighbors. Without

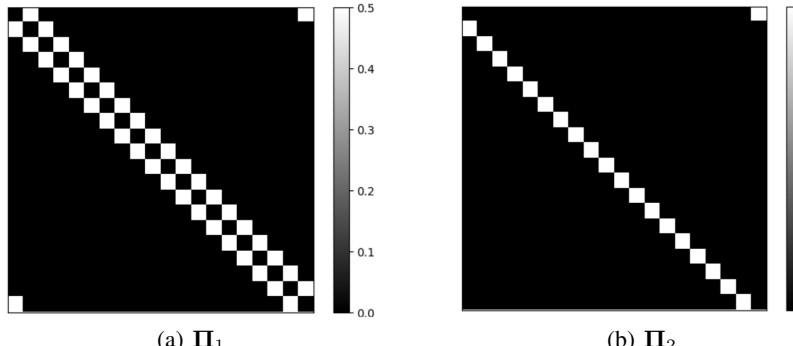

(a) $\mathbf{\Pi}_1$          (b) $\mathbf{\Pi}_2$

Figure 2: Visualization of two transition matrices $\mathbf{\Pi}_1$ and $\mathbf{\Pi}_2$ for Task 1 and Task 2. In $\mathbf{\Pi}_1$, the white block represents 0.5, and the black block represents 0. In $\mathbf{\Pi}_2$, the white block represents 1, and the black block represents 0.

loss of generality, we suppose that the nodes are assigned with node IDs $1, 2, \ldots, K$ in a clockwise manner. A 'walk' on the circle refers to the process where a 'walker' moves step-by-step among the nodes of the circle. At each step, the walker moves to a neighboring node of its current position. In this way, a walk of length $N$ generates a sequence of 'states' $s_1, \ldots s_N$, where $s_i \in [K]$ denotes the location (node ID) of the walker at the $i$-th step.

With such sequential data generated by random or deterministic walks, we can consider the problem of predicting the location of the walker ($s_N$) based on the historical locations $s_1, \ldots s_{N-1}$. Specifically, for $i \in [N-1]$, we denote by $\boldsymbol{x}_i = \boldsymbol{e}_{s_i} \in \mathbb{R}^K$ the one-hot embedding of $s_i$, and denote $y = s_N$. Our goal is then to train a model to predict $y$ based on $\boldsymbol{x}_1, \ldots, \boldsymbol{x}_{N-1}$.

As mentioned in the introduction, we consider two walks, which we call "random walk" and "deterministic walk" respectively for simplicity. In the following, we give their detailed definitions and discuss some basic properties respectively.

**Random walk.** In the case of random walk, starting from a random location, the walker randomly decides to move clockwise or counterclockwise at each step. For any integers $s$, we define $\langle s \rangle_K$ as the integer satisfying $\langle s \rangle_K \in [K]$ and $s \equiv \langle s \rangle_K \pmod{K}$. With this definition, the probabilistic model is defined as follows.

**Task 1** (Random Walk). Suppose that $\boldsymbol{x}_1, \ldots, \boldsymbol{x}_{N-1}, y$ are generated as follows:

1. Draw $s_1 \sim \text{Unif}([K])$.

2. For $i = 2, \ldots, N$, sample $s_i = \langle s_{i-1} - 1 \rangle_K$ or $s_i = \langle s_{i-1} + 1 \rangle_K$ equally likely.

3. Set $\boldsymbol{x}_i = \boldsymbol{e}_{s_i}$, $i \in [N-1]$, and $y = s_N$.

By the definition above, it is clear that the sequence $\boldsymbol{x}_1, \ldots, \boldsymbol{x}_{N-1}, \boldsymbol{e}_y$ form a Markov chain, and $\mathbb{P}(y|\boldsymbol{x}_1, \ldots, \boldsymbol{x}_{N-1}) = \mathbb{P}(y|\boldsymbol{x}_{N-1})$. Moreover, the transition matrix of the Markov model is

$$\mathbf{\Pi}_1 = (\pi_{ij}^{(1)})_{K \times K}, \text{ where } \pi_{i,j}^{(1)} = 1/2 \cdot \mathbb{1}\{i \equiv j - 1 (\text{mod } K)\} + 1/2 \cdot \mathbb{1}\{i \equiv j + 1 (\text{mod } K)\}.$$

An visualization of $\mathbf{\Pi}_1$ is given in Figure 2(a). The Markov property indicates that the optimal predictor of $y$ is given by

$$f_{\text{Task1}}^{\text{OPT}}(\boldsymbol{x}_1, \ldots, \boldsymbol{x}_{N-1}) = \mathbf{\Pi}_1^\top \boldsymbol{x}_{N-1},$$

and the optimal prediction accuracy any predictor can achieve is $\text{OPT}_{\text{Task1}} = 1/2$.

**Deterministic walk.** In the case of deterministic walk, starting from a random location, the walker deterministically moves counterclockwise at each step. The corresponding probabilistic model is defined as follows.

**Task 2** (Deterministic Walk). Suppose that $\boldsymbol{x}_1, \ldots, \boldsymbol{x}_{N-1}, y$ are generated as follows:

1. Draw $s_1 \sim \text{Unif}([K])$.

2. For $i = 2, \ldots, N$, set $s_i = \langle s_{i-1} - 1 \rangle_K$.

3. Set $\boldsymbol{x}_i = \boldsymbol{e}_{s_i}$, $i \in [N-1]$, and $y = s_N$.

The only randomness in the case of deterministic walk is the initial state. Moreover, the transition matrix of the deterministic walk is

$$\boldsymbol{\Pi}_2 = (\pi_{ij}^{(2)})_{K \times K}, \text{ where } \pi_{i,j}^{(2)} = \mathbb{1}\{i \equiv j - 1 \pmod{K}\}.$$

An visualization of $\boldsymbol{\Pi}_2$ is given in Figure 2(b). It is natural that the prediction of $y$ in deterministic walk is relatively easy compared with the case of random walk. In fact, as long as one of the historical locations is known, there should exist a perfect predictor of $y$:

$$\text{For any } i \in [N-1], \ f_{\text{Task2},i}^{\text{OPT}}(\boldsymbol{x}_i) = (\boldsymbol{\Pi}_2^\top)^{N-i}\boldsymbol{x}_i = \boldsymbol{e}_y \text{ with probability 1.}$$

Therefore, the optimal prediction accuracy any predictor can achieve is $\text{OPT}_{\text{Task2}} = 1$.

## 2.2 Transformer Architecture

We consider learning the prediction tasks in random and deterministic walks introduced in the previous section with a simple one-layer transformer model. By naturally treating the one-hot vectors $\boldsymbol{x}_1, \ldots, \boldsymbol{x}_{N-1}$ as tokens, then the task to predict the next position $y$ is exactly a problem of next token prediction.

Define the data matrix $\boldsymbol{X} = [\boldsymbol{x}_1, \boldsymbol{x}_2, \ldots, \boldsymbol{x}_{N-1}, \boldsymbol{0}] \in \mathbb{R}^{K \times N}$. We also consider a positional embedding matrix $\boldsymbol{P} = [\boldsymbol{p}_1, \boldsymbol{p}_2, \ldots, \boldsymbol{p}_N] \in \mathbb{R}^{M \times N}$, where $M$ is the embedding dimension with $M = \Omega(N^{3/2})$ and $\boldsymbol{p}_i \in \mathbb{R}^M$ is defined as

$$\boldsymbol{p}_i = \left[\sin\left(\frac{i\pi}{M+1}\right), \sin\left(\frac{2i\pi}{M+1}\right), \ldots, \sin\left(\frac{Mi\pi}{M+1}\right)\right]^\top$$

for $i = 1, 2, \ldots, N$. The positional embeddings above are inspired by the fact that $\langle \boldsymbol{p}_i, \boldsymbol{p}_j \rangle = 0$ for all $i \neq j$, which significantly helps to simplify our theoretical analysis (see Lemma F.5 in the appendix). Then, we define the matrix $\widetilde{\boldsymbol{X}}$ by concatenating the input matrix $\boldsymbol{X}$ and the position matrix $\boldsymbol{P}$ as

$$\widetilde{\boldsymbol{X}} = \begin{bmatrix} \boldsymbol{X} \\ \boldsymbol{P} \end{bmatrix} = \begin{bmatrix} \boldsymbol{x}_1 & \boldsymbol{x}_2 & \cdots & \boldsymbol{x}_{N-1} & \boldsymbol{0} \\ \boldsymbol{p}_1 & \boldsymbol{p}_2 & \cdots & \boldsymbol{p}_{N-1} & \boldsymbol{p}_N \end{bmatrix} := [\widetilde{\boldsymbol{x}}_1, \widetilde{\boldsymbol{x}}_2, \ldots, \widetilde{\boldsymbol{x}}_N] \in \mathbb{R}^{(K+M) \times N}.$$

We consider a single-layer transformer model to make a prediction on a given input matrix $\boldsymbol{X}$. The transformer is defined as follows:

$$f_\theta(\boldsymbol{X}) = \boldsymbol{V}\boldsymbol{X}\mathcal{S}(\widetilde{\boldsymbol{X}}^\top \boldsymbol{W} \widetilde{\boldsymbol{x}}_N), \tag{2.1}$$

where $\boldsymbol{V} \in \mathbb{R}^{K \times K}$, $\boldsymbol{W} \in \mathbb{R}^{(K+M) \times (K+M)}$ are the trainable parameter matrices, $\mathcal{S} : \mathbb{R}^N \to \mathbb{R}^N$ is the softmax function defined by $[\mathcal{S}(\boldsymbol{z})]_i = \frac{\exp(z_i)}{\sum_{j=1}^N \exp(z_j)}$, and $\theta = (\boldsymbol{V}, \boldsymbol{W})$ denotes the collection of all the trainable parameters. In this definition, we consider a reparameterization where we use a single matrix $\boldsymbol{W}$ to denote the product of the commonly considered key and query matrices in practice (Vaswani et al., 2017). Such kind of reparameterizations is commonly considered in theoretical studies of transformer models (Jelassi et al., 2022; Tian et al., 2023a; Huang et al., 2024; Zhang et al., 2024; Nichani et al., 2024; Li et al., 2024a; Wang et al., 2024; Ildiz et al., 2024).

Note that by the definition in (2.1), given any input matrix $\boldsymbol{X}$, the transformer model outputs a $K$-dimensional vector. This follows the standard practice of $K$-class classification – for $i \in [K]$, $[f_\theta(\boldsymbol{X})]_i$ can be treated as a predicted "score" of the $i$-th class. More specifically, we can define the prediction rule as follows.

**Definition 2.1.** For any predictor $f(\boldsymbol{X}) : \mathbb{R}^{K \times N} \to \mathbb{R}^K$, the predicted label is given as

$$\text{Pred}(f(\boldsymbol{X})) := \min\left\{j \in [K] : [f(\boldsymbol{X})]_j = \max_{i \in [K]}\{[f(\boldsymbol{X})]_i\}\right\}.$$

The definition above matches the common practice to predict the label that corresponds to the entry in $f(\boldsymbol{X})$ with the maximum function value. It also gives a naive way to handle ties – when $f(\boldsymbol{X})$ contains multiple dimensions with the same (and maximum) function value, we always predict the dimension corresponding to the smallest label. We remark that this definition to handle ties is just to exclude ambiguity, and the detailed rule to handle ties is not essential. Our result works for all reasonable methods to handle ties.

## 2.3 TRAINING METHOD

We consider training the transformer model defined in (2.1) by gradient descent. We consider to minimize the loss function

$$L(\theta) = \mathbb{E}_{(\boldsymbol{X},y)} \left[ \ell \left( \boldsymbol{e}_y^\top f_\theta(\boldsymbol{X}) \right) \right], \tag{2.2}$$

where $\ell(\cdot)$ is a loss function. In terms of the specific choice of $\ell(\cdot)$, our analysis will cover (i) learning the random walk defined in Task 1 by minimizing the log-loss $\ell(z) = -\log(z + \epsilon)$, which has been considered in a series of recent works (Li et al., 2024a; Ildiz et al., 2024; Makkuva et al., 2024; Thrampoulidis, 2024), and also (ii) learning the deterministic walk defined in Task 2 by minimizing *any* loss function $\ell(\cdot)$.

We consider gradient descent with zero initialization $\boldsymbol{V}^{(0)} = \boldsymbol{0}^{K \times K}$, $\boldsymbol{W}^{(0)} = \boldsymbol{0}^{(K+M) \times (K+M)}$ to train the model. The update rule for the parameter matrices $\boldsymbol{V}$ and $\boldsymbol{W}$ are as follows:

$$\boldsymbol{V}^{(t+1)} = \boldsymbol{V}^{(t)} - \eta \nabla_{\boldsymbol{V}} L(\theta^{(t)}) \text{ and } \boldsymbol{W}^{(t+1)} = \boldsymbol{W}^{(t)} - \eta \nabla_{\boldsymbol{W}} L(\theta^{(t)}), \tag{2.3}$$

where $\eta > 0$ is the learning rate and $t \geq 0$ is the iteration number.

## 3 MAIN RESULTS

In this section, we present our main theoretical results on using a self-attention layer to learn the random and deterministic walks defined in Task 1 and Task 2. In our result, we can choose any $T^* = \text{poly}(\eta, \epsilon^{-1}, K, N, M)$ as the maximum admissible number of iterations, and only consider the training period $0 \leq t \leq T^*$. This technical assumption regarding a polynomially large maximum admissible number prevents training from becoming exponentially long and is a mild assumption since exponentially long training is impractical.

Our main results for learning the random walk in Task 1 is given in the following theorem.

**Theorem 3.1.** Suppose that $K$ is a constant even integer, and $N = \omega(1)$. Further suppose that the transformer is trained by gradient descent (2.3) to minimize the loss (2.2) with $\ell(z) = -\log(z + \epsilon)$, and $\eta, \epsilon = \Theta(1)$. Then there exists $T_0 = \Theta(1)$, such that for all $T_0 \leq T \leq T^*$, it holds that:

1. The trained transformer achieves optimal prediction accuracy:

$$\mathbb{P}_{(\boldsymbol{X},y) \sim \text{Task1}} \left[ \text{Pred}(f_{\theta^{(T)}}(\boldsymbol{X})) = y \right] = \text{OPT}_{\text{Task1}} = \frac{1}{2}.$$

2. The transformer converges to the optimal predictor. Suppose that $(\boldsymbol{X}, y)$ is generated by Task 1. Then with probability 1, it holds that

$$\left\| \frac{f_{\theta^{(T)}}(\boldsymbol{X})}{\|f_{\theta^{(T)}}(\boldsymbol{X})\|_2} - f_{\text{Task1}}^{\text{OPT}}(\boldsymbol{X}) \right\|_2 = \mathcal{O}\left( \frac{1}{\sqrt{T}} \right).$$

3. The value matrix converges to the true transition matrix in direction:

$$\left\| \frac{\boldsymbol{V}^{(T)}}{\|\boldsymbol{V}^{(T)}\|_F} - \frac{\boldsymbol{\Pi}_1^\top}{\|\boldsymbol{\Pi}_1^\top\|_F} \right\|_F = \mathcal{O}\left( \frac{1}{\sqrt{T}} \right).$$

4. The softmax attention selects the correct token. Suppose that $(\boldsymbol{X}, y)$ is generated by Task 1. Then with probability 1, it holds that

$$\left[ \mathcal{S}(\widetilde{\boldsymbol{X}}^\top \boldsymbol{W}^{(T)} \widetilde{\boldsymbol{x}}_N) \right]_{N-1} \geq 1 - \exp(-\Omega(N)), \text{ and } \left[ \mathcal{S}(\widetilde{\boldsymbol{X}}^\top \boldsymbol{W}^{(T)} \widetilde{\boldsymbol{x}}_N) \right]_j \leq \exp(-\Omega(N))$$

for all $j \neq N - 1$.

In terms of the prediction, the first result in Theorem 3.1 states that the transformer trained by gradient descent for a constant number of iterations can achieve a prediction accuracy $1/2$, which matches the optimal accuracy $\text{OPT}_{\text{Task1}}$ for Task 1. The second result in Theorem 3.1 further gives a more detailed characterization of the trained transformer, and demonstrates that the normalized

model converges to the optimal prediction model $f_{\text{Task1}}^{\text{OPT}}(\boldsymbol{X}) = \boldsymbol{\Pi}_1^\top \boldsymbol{x}_{N-1}$. This convergence result strongly indicates that the transformer model learns token selection – it successfully learns to focus on the direct parent of $y$, and then makes a prediction based on this direct parent.

The third and fourth results in Theorem 3.1 further back up the first two results by a precise characterization on how the self-attention mechanism works in predicting random walks. Specifically, the third result demonstrates that in direction, the value matrix $\boldsymbol{V}^{(T)}$ converges to the ground-truth transition matrix $\boldsymbol{\Pi}_1$, and the last result indicates that the softmax score assigned to the $(N-1)$-th token is close to 1, demonstrating that the attention layer can correctly select the parent token. Together, these two results illustrate that the trained one-layer transformer model makes predictions by (i) selecting the correct parent token $\boldsymbol{x}_{N-1}$ of $y$ by assigning a softmax weighting close to 1 to it, and (ii) predicting $y$ by applying a one-step transition model to $\boldsymbol{x}_{N-1}$ through the linear mapping defined by the value matrix.

We would also like to remark that Theorem 3.1 assumes that the number of nodes $K$ on the circle is an even integer. This assumption is to simplify our analysis and avoid tedious discussions on whether $K$ is even or odd. Our result should also hold for odd values of $K$, but some parts of the proof may need to be changed. We believe that demonstrating the results for even $K$ can already clearly and convincingly demonstrate the performance of transformers in learning random walks.

The following theorem summarizes our main results on learning the deterministic walk defined in Task 2 with a one-layer transformer model.

**Theorem 3.2.** Suppose that $K$ is a constant integer, and $N = rK + 1$ with $r \geq 1$. Further suppose that the transformer is trained by gradient descent (2.3) to minimize the loss (2.2). Then for any loss function $\ell(\cdot)$, any learning rate $\eta > 0$, and any $T \geq 0$, it holds that

$$\mathbb{P}_{(\boldsymbol{X},y)\sim\text{Task2}}\big(\text{Pred}(f_{\theta^{(T)}}(\boldsymbol{X})) = y\big) = \frac{1}{K}.$$

Moreover, suppose that $(\boldsymbol{X}, y)$ is generated by Task 2. Then with probability 1, for all $T \geq 0$, it holds that

$$\boldsymbol{V}^{(T)} \propto \mathbf{1}_{K \times K}, \text{ and } \left[\mathcal{S}(\widetilde{\boldsymbol{X}}^\top \boldsymbol{W}^{(T)} \widetilde{\boldsymbol{x}}_N)\right]_1 = \cdots = \left[\mathcal{S}(\widetilde{\boldsymbol{X}}^\top \boldsymbol{W}^{(T)} \widetilde{\boldsymbol{x}}_N)\right]_{N-1}.$$

Theorem 3.2 shows that the prediction accuracy of the trained transformer for Task 2 is $1/K$, which is the same as the accuracy of a random guess. Moreover, the characterizations of the value matrix $\boldsymbol{V}^{(T)}$ and the softmax scores further demonstrate that the transformer takes average over all tokens, and then gives the same prediction scores for all possible values of $y$. Notably, these results hold for any choice of the loss function and any learning rate, indicating that this failure case of the transformer cannot be resolved by simply adjusting these training setups.

As we have discussed in Section 2.1, Task 2 on the deterministic walk is naturally easier compared with Task 1 on the random walk. However, Theorems 3.1 and 3.2 together lead a surprising conclusion: a one-layer transformer trained by gradient descent can successfully learn to predict random walks, but provably fails to predict deterministic walks. Here we remark that this counter-intuitive result is due to the fact that self-attention may fail when there are multiple highly informative tokens. We will give a detailed discussion in the next section.

## 4 EXPERIMENT

In this section, we present simulation results training on synthetic data to support our theoretical analysis. We consider two cases: the first one is the zero initialization case which aligns with the setting used in our theoretical analysis, and the second one is the random initialization case which is more commonly used in the practical scenario. In all experiments introduced in this section, we set the number of nodes $K = 6$ and the length of each sequence $N = 97$. We utilize the transformer model introduced in Section 2 and utilize the gradient method to train the model. The prediction accuracy is calculated based on 1000 test data.

**Zero initialization.** In this case, we set the length of the positional embedding $M = 1000$, the initialization $\boldsymbol{V}^{(0)} = \mathbf{0}_{K \times K}, \boldsymbol{W}^{(0)} = \mathbf{0}_{(K+M) \times (K+M)}$, and the learning rate $\eta = 1$. The constant $\epsilon$ in the log-loss is set as $\epsilon = 0.1$. For both tasks, we generate 1000 sequences to train the model.

Figure 3 and Figure 4 illustrate the results of the experiment for Task 1 and Task 2 respectively: Figure 3(a) and Figure 4(a) present the prediction accuracy; Figure 3(b) and Figure 4(b) visualize the value matrix $\boldsymbol{V}^{(T)}$ after 50 iterations; Figure 3(c) and Figure 4(c) display the attention scores attached to each token after 50 iterations. To clearly observe the results, we also provide Figure 3(d) that represents the part of Figure 3(c).

We can observe that these experimental results for Task 1 provide strong support for Theorem 3.1. Figure 3(a) shows that the prediction accuracy is close to the optimal accuracy (50%) within constant iterations. Figure 3(b) indicates that $\boldsymbol{V}^{(T)}$ can recover the transition matrix $\boldsymbol{\Pi}_1$ as shown in Figure 2(a). Figure 3(c) presents that the $(N-1)$-th attention score is the highest and close to 1, indicating that the self-attention layer is able to select the true parent token. All of these experimental results demonstrate the performance of transformers in learning random walks.

In addition, we can find that the experimental outcomes for Task 2 match the theoretical results stated in Theorem 3.2. We obtain an accuracy close to 0.167 from Figure 4(a), which suggests that the prediction accuracy for learning Task 2 is approximately equal to $1/K$, far away from the optimal accuracy (100%) and no better than a random guess. Figure 4(b) indicates that $\boldsymbol{V}^{(T)}$ is approximately proportional to $\boldsymbol{1}_{K \times K}$. Figure 4(c) shows that the attention scores attached to all tokens are identical, which proves that the self-attention layer cannot select any of the tokens when learning Task 2. These experimental results for Task 2 demonstrate that the self-attention mechanism struggles in learning deterministic walks.

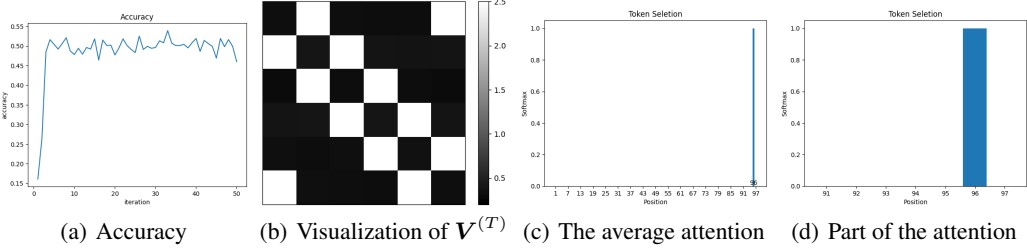

(a) Accuracy     (b) Visualization of $\boldsymbol{V}^{(T)}$    (c) The average attention    (d) Part of the attention

Figure 3: The results of the experiments for Task 1 with zero initialization: (a) is the test accuracy; (b) is the visualization of $\boldsymbol{V}$; (c) and (d) present the average attention of the test data with $x$-axis representing the position of the token and $y$-axis representing the attention score.

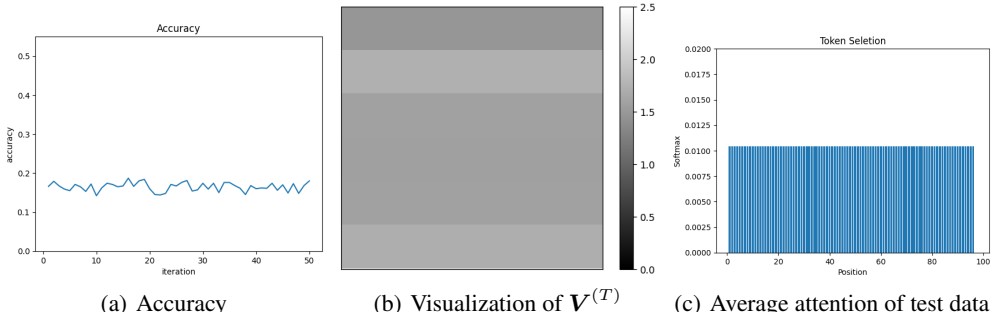

(a) Accuracy        (b) Visualization of $\boldsymbol{V}^{(T)}$      (c) Average attention of test data

Figure 4: The results of the experiment for Task 2 with zero initialization. (a) is the prediction accuracy with $x$-axis representing the iteration and $y$-axis representing the accuracy. (b) is the visualization of $\boldsymbol{V}$. (c) is the average attention of the test data with $x$-axis representing the position of the token and $y$-axis representing the attention score.

**Random initialization.** In this case, we set the length of the positional embedding $M = 1000$, the initialization $\boldsymbol{V}_{ij}^{(0)}, \boldsymbol{W}_{ij}^{(0)} \sim N(0, \sigma^2)$ with $\sigma = 0.01$, and the learning rate $\eta = 0.01$. The constant $\epsilon$ in the log-loss is set as $\epsilon = 0.1$. For both tasks, we generate 1000 sequences to train the model.

Figure 5 illustrates the results of the experiment for Task 1 and Task 2. Figure 5(a) and Figure 5(c) show the prediction accuracy within 1000 iterations for Task 1 and Task 2 respectively. In Figure 5(b) and Figure 5(d), we first normalize the output of the trained transformer model to get a $K$-dimensional vector, which can be regarded as the prediction distribution of $K$ locations. The KL-divergence between this prediction distribution and the true distribution of $y|\boldsymbol{x}_{N-1}$ is illustrated in Figure 5(b) and Figure 5(d).

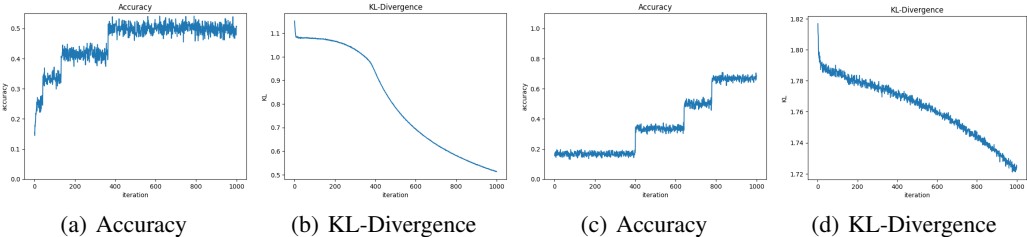

|                |                |                |                |
|:--------------:|:--------------:|:--------------:|:--------------:|
| (a) Accuracy   | (b) KL-Divergence | (c) Accuracy | (d) KL-Divergence |

Figure 5: The results of the synthetic experiment with random initialization: (a) and (b) correspond to the experiment for Task 1; (c) and (d) correspond to the experiment for Task 2. (a) and (c) present the prediction accuracy. In (b) and (d), we first normalize the output of the trained transformer model to get a $K$-dimensional vector, representing the prediction distribution of $K$ locations. Then, we display the KL-divergence between this prediction distribution and the true distribution of $y|\boldsymbol{x}_{N-1}$ in (b) and (d).

Figure 5(a) clearly shows that in the experiment for Task 1, the accuracy is close to the optimal accuracy (50%) after around 400 iterations. However, as shown in Figure 5(c), for Task 2, the prediction accuracy cannot reach the optimal accuracy (100%) within 1000 iterations. Based on the plots of KL-divergence, we can also see that the transformer learns the true prediction distribution of random walks much faster than learning that of deterministic walks. Note that these results are for training with random initialization, and hence the results do not perfectly match our theory for zero initialization in Section 3. However, the experiment results still clearly demonstrate that Task 2 for learning deterministic walks is significantly more challenging even with random initialization.

## 5 Successes & Pitfalls Beyond Random/Deterministic Walks

In Sections 3 and 4, we demonstrate that a one-layer transformer can be trained to optimally predict random walks, but fails in the arguably easier task of predicting deterministic walks. In this section, we provide a concise explanation for the counter-intuitive phenomenon, and discuss other learning tasks beyond random and deterministic walks, where transformer training may also be challenging.

### 5.1 An Intuitive Explanation

Here we aim to give an intuitive explanation on why transformers fail in learning deterministic walks. A natural starting point is to study the differences between random and deterministic walks.

**Difference between random and deterministic walks.** As is discussed in Section 2.1, a key difference between random and deterministic walks is that, in random walks, the optimal predictor must rely on $\boldsymbol{x}_{N-1}$, which is the direct 'parent' of $y$, to make a prediction. On the other hand, in deterministic walks, knowing any one of the historical locations $\boldsymbol{x}_i$, $i \in [N-1]$ can provide sufficient information to achieve perfect prediction. In other words, in random walk, there is a unique token that is the most 'informative', while in deterministic walks, all tokens are equally 'informative'.

Motivated by the discussion above, we consider using *entropy*[1] as a more concrete characterization of how 'informative' each token is. Specifically, we take the case $(K, N) = (6, 7)$ as an example. Denoting by $\overline{\boldsymbol{x}}$ the average over $\boldsymbol{x}_1, \ldots, \boldsymbol{x}_6$, we report the values of

$$\text{Entropy}(y|\boldsymbol{x}_i) = \mathbb{E}_{(\boldsymbol{X}, y)}\left[-\log \mathbb{P}(y|\boldsymbol{x}_i)\right], \; i = 1, \ldots, 6,$$

$$\text{Entropy}(y|\overline{\boldsymbol{x}}) = \mathbb{E}_{(\boldsymbol{X}, y)}\left[-\log \mathbb{P}(y|\overline{\boldsymbol{x}})\right], \text{ and } \text{Entropy}(y) = \mathbb{E}_{(\boldsymbol{X}, y)}\left[-\log \mathbb{P}(y)\right]$$

in Figure 6 for both random (Task 1) and deterministic (Task 2) walks. For Task 1, we can observe that $\text{Entropy}(y|\boldsymbol{x}_6)$ is significantly smaller than the others. Thus, $\boldsymbol{x}_6$ can be regarded as the most informative token in predicting $y$ in Task 1. However, for Task 2, the values of all $\text{Entropy}(y|\boldsymbol{x}_i)$'s are the same and are all zero, indicating that all the tokens are perfectly informative in predicting $y$. More importantly, we note that in Task 1, $\text{Entropy}(y|\overline{\boldsymbol{x}})$ is smaller than $\text{Entropy}(y)$, which implies that knowing $\overline{\boldsymbol{x}}$ can help predicting $y$ to a certain extent. However, in Task 2, we can see

---

[1]We clarify that entropy is not directly utilized in our proof. Nevertheless, it can provide us the tool to clearly explain the intuition of our proof.

that $\text{Entropy}(y|\overline{\boldsymbol{x}}) = \text{Entropy}(y)$, meaning that *the token average $\overline{\boldsymbol{x}}$ does not provide any useful information in predictiing $y$ in Task 2.*

| | $x_1 \longrightarrow$ | $x_2 \longrightarrow$ | $x_3 \longrightarrow$ | $x_4 \longrightarrow$ | $x_5 \longrightarrow$ | $x_6 \longrightarrow$ | $y$ | |
|---|---|---|---|---|---|---|---|---|
| | Entropy($y|x_1$) | Entropy($y|x_2$) | Entropy($y|x_3$) | Entropy($y|x_4$) | Entropy($y|x_5$) | Entropy($y|x_6$) | Entropy($y$) | Entropy($y|\overline{x}$) |
| **Task 1** | 1.098 | 1.098 | 1.095 | 1.082 | 1.040 | 0.693 | 1.792 | 1.558 |
| **Task 2** | 0 | 0 | 0 | 0 | 0 | 0 | 1.792 | 1.792 |

Figure 6: An illustration of the values of $\text{Entropy}(y)$, $\text{Entropy}(y|\overline{\boldsymbol{x}})$, and $\text{Entropy}(y|\boldsymbol{x}_i)$ for each token $\boldsymbol{x}_i$ in both Task 1 and Task 2 under the condition that $K = 6, N = 7$.

Figure 6 leads to an explanation on why transformers struggle in learning deterministic walks:

*At small (random) initialization, the initial softmax scores on each token are almost equal and the initial output of the transformer is approximately $\boldsymbol{V}\overline{\boldsymbol{x}}$, where $\overline{\boldsymbol{x}}$ is the average over all tokens. However, $\overline{\boldsymbol{x}}$ contains no useful information that can help prediction, and hence the transformer can never (or at least cannot efficiently) be trained to make correct predictions.*

## 5.2 BEYOND RANDON/DETERMINISTIC WALKS: EXAMPLES IN SIMPLE NLP TASKS

In Section 5.1, we provide an intuitive explanation of why one-layer transformers can hardly learn the simple task of deterministic walks. Here, motivated by this intuitive explanation, we discuss other tasks which transformers may also struggle to learn.

We construct two simple tasks in natural language processing (NLP). The detailed descriptions of these two new tasks are given as follows.

**Task 3.** We consider a very simple question answering task. Specifically, possible input questions are all of the form:

*Based on the list 'apple, orange, apple, apple, orange', which type of fruit appears most frequently?*

Here, the list stated in the question can be any combination of 'apple' and 'orange' with a fixed length of 5. Therefore, there are a total of 32 possible questions the model may see, and each of these questions occur with probability $1/32$. Ignoring punctuation marks, each input sample is assumed to be 16 words involving the list and other words in the inquiry sentence. The correct response (the 'label' for classification) is the fruit that appears most frequently in the list. For example, for the question *"Based on the list 'apple, orange, apple, apple, orange', which type of fruit appears most frequently?"*, the correct response is *apple*.

**Task 4.** We again consider a very simple question answering task with only two possible questions:

> *Based on the sentence 'I prefer an apple to an orange', which type of fruit do I prefer?*
> *Based on the sentence 'I prefer an orange to an apple', which type of fruit do I prefer?*

Here, each of the two questions above occurs with probability $1/2$. Similar to Task 3, we ignore punctuation marks and the input is the 18 words in the sentence. The correct response (the 'label' for classification) is *apple* for the first question above, and *orange* for the second question above.

Task 3 and Task 4 above are motivated by our discussion and explanation in Section 5.1. Intuitively, in Task 3, the average of the word embeddings $\overline{\boldsymbol{x}}$ in a question can still help the model to find the correct response. In contrast, in Task 4, we can see that the two questions give the *same* average of word embeddings $\overline{\boldsymbol{x}}$, and therefore, it is impossible to give the correct response based on $\overline{\boldsymbol{x}}$. Below, we experimentally study the capability of one-layer transformers in learning these two tasks.

Combining all the words appearing in two tasks, we attain a vocabulary with a length of 19: {'apple', 'orange', 'Based', 'on', 'the', 'which', 'type', 'of', 'fruit', 'list', 'appears', 'most', 'frequently', 'sentence', 'I', 'prefer', 'an', 'to', 'do'}. We embed this sequence as a matrix $\boldsymbol{E} = [\boldsymbol{e}_1, \boldsymbol{e}_2, ..., \boldsymbol{e}_{19}] \in \mathbb{R}^{19 \times 19}$, where each word is embedded as a one-hot vector $\boldsymbol{e}_i$. Then, we know that the length of the vocabulary $K$ and the length of each input sequence $N$ are set as $(K, N) = (19, 17), (19, 19)$ for Task 3 and Task 4 respectively.

In the experiment for these two NLP tasks, we consider the similar transformer model as we introduced in our theoretical analysis. To train the model, we consider Gaussian random initialization $\boldsymbol{V}_{ij}^{(0)}, \boldsymbol{W}_{ij}^{(0)} \sim N(0, \sigma^2)$ with $\sigma = 0.01$, and we use gradient descent with learning rate $\eta = 0.1$ to train the model. The constant $\epsilon$ in the log-loss is set as $\epsilon = 0.1$. Both experiments are conducted on 1000 training data and 1000 test data.

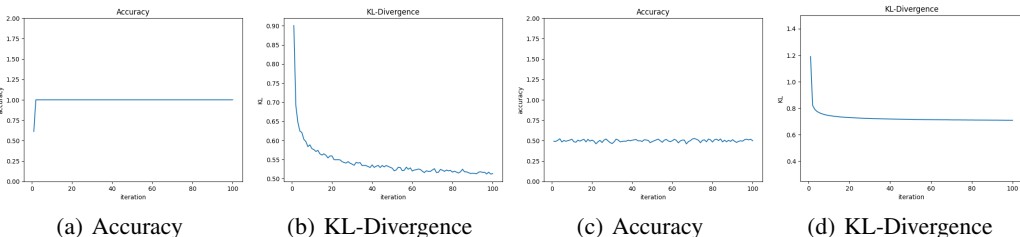

| (a) Accuracy | (b) KL-Divergence | (c) Accuracy | (d) KL-Divergence |

Figure 7: The results of the experiment for Task 3 and Task 4: (a) and (b) correspond to the experiment for Task 3; (c) and (d) correspond to the experiment for Task 4.

Figure 7 shows the experiment results for Task 3 and Task 4. Figure 7(a) and Figure 7(c) present the test accuracy. In Figure 7(b) and Figure 7(d), we first normalize the output of the trained transformer model to get a $K$-dimensional vector, representing the prediction distribution of $K$ words. Then, we report the KL-divergence between this prediction distribution and the true distribution of $y|\boldsymbol{x}_1, \boldsymbol{x}_2, ..., \boldsymbol{x}_{N-1}$ in Figure 7(b) and Figure 7(d). The experiment results show a clear difference between the performances of the transformer model in the two tasks. In Task 3, the trained transformer model can successfully approach the optimal accuracy (100%) within 100 iterations. However, in Task 4, the test accuracy always remains around 50%, which is the accuracy of a random guess.

Comparing these two NLP tasks, we observe that in Task 3, no single word can determine the answer; instead, we must combine all five words in the list to solve the question. In contrast, in Task 4, the single word in the 8th or 11th position can uniquely determine the answer. Thus, Task 4 can be naturally considered a 'simpler' task and easier to learn. However, the experiment results show a counter-intuitive phenomenon that the transformer fails to learn the relatively 'simple' Task 4 but can learn the relatively 'difficult' Task 3. This phenomenon can also be explained by our discussion in Section 5.1: *the self-attention mechanism struggles in the case that there are multiple highly informative tokens but the average of them is not informative.*

The experiment results for these simple but intuitive NLP tasks demonstrate that our theories and explanations for random and deterministic walks can guide the construction of various other learning tasks and predict the performance of a transformer model in these tasks. This confirms the validity of our theories and explanations, and highlights the insights provided by our study.

## 6 CONCLUSION

This paper studies the self-attention mechanism via a random walk and a deterministic walk, where we consider a transformer with a single self-attention layer. It can be demonstrated that the self-attention layer can learn random walks well by effectively selecting the correct parent token and obtaining the optimal next-token prediction accuracy. However, when learning the simpler deterministic task, the self-attention layer fails to select any token; instead, the self-attention layer assigns the same attention score to all the tokens. As a result, the trained transformer shows no improvement over a random guess. We thus discover that multiple informative tokens may hinder the performance of the self-attention mechanism by failing to select any specific token.

This work performs two specific cases studies on learning random and deterministic walks with one-layer transformers. While the conclusions of these case studies provide valuable insights, it is important to extend the results and study the performance of deeper transformer architectures, which may require more advanced theoretical tools. Moreover, extending the finding to more complicated learning tasks, such as random sequences generated by Bayesian networks, is also an important future work direction.

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

## A    ADDITIONAL RELATED WORK

In this section, we give an overview of some additional related works.

**Next-token prediction.** Thrampoulidis (2024) explores the implicit bias of next-token prediction employing a related SVM formulation. Lu et al. (2024) demonstrates that transformers fail to solve the Partially Observable Markov Decision Processes problem (POMDP) even with sufficient data. He & Su (2024) observes a phenomenon of next-token prediction in LLM that each layer contributes equally to enhancing the prediction accuracy. Tian et al. (2023a) studies the SGD training dynamics of a transformer with one self-attention layer and one decoder layer for next-token prediction, restricted to some specific assumptions like no positional encoding, long input sequences, and the fact that the decoder layer learns faster than the self-attention layer.

**Training dynamics of transformers.** Mahankali et al. (2023); Zhang et al. (2024) investigate the training dynamics of in-context learning in transformers with a single self-attention layer trained through gradient flow on linear regression tasks. Huang et al. (2024) solves in-context linear regression with the orthogonal input data by gradient descent on a single softmax attention layer. Jelassi et al. (2022) demonstrates that the position-position block of a single attention layer in a vision transformer can encode spatial structure by dealing with a binary classification task. Tian et al. (2023b) delves into the training process of transformers with multi-layers by analyzing the dynamics of the MLP layers. Bietti et al. (2024) analyzes a synthetic in-context learning task and emphasizes the significance of weight matrices as associative memories. Abbe et al. (2024) shows incremental learning dynamics in transformers with diagonal attention matrices.

## B    INFORMAL PROOF SKETCHES OF THE MAIN RESULTS

In this section, we discuss the training dynamics of the transformer model in learning Task 1 and Task 2. These characterizations of training dynamics also serve as informal proof sketches of Theorems 3.1 and 3.2. The proofs follow our discussion in Section 5 about the fact that in Task 1, the 'direct parent' of $y$ is more 'informative' than the other tokens. On the other hand, in Task 2 for deterministic walks, all tokens are perfectly and equally informative, but their average is not informative at all.

**Training dynamics in learning random walks.** We consider the training procedure of a one-layer transformer in learning Task 1. Recall that we train the transformer model with gradient descent starting from zero initialization. We can characterize the first three gradient steps as follows:

Step 1.   After the first gradient descent step, it can be shown that $V^{(1)}$ is a symmetric matrix whose largest entries appear exactly on the locations of the non-zero entries of $\Pi_2$ (see Lemma D.2 and Lemma D.3 in the appendix). $W^{(1)}$ is still a zero matrix due to the fact that $V^{(0)} = 0$.

Step 2.   With the same analysis as in Step 1, we can also show that $V^{(2)}$ is a symmetric matrix whose largest entries appear exactly on the locations of the non-zero entries of $\Pi_2$. Moreover, based on the result on $V^{(1)}$, it can be further shown that $W^{(2)}$ is updated so that higher softmax weightings will be put upon $x_{N-1}$ (see Lemma D.6 in the appendix).

Step 3.   The higher weighting on $x_{N-1}$ by $W^{(2)}$ further encourages $V^{(3)}$ to be updated towards $\Pi_2$ in direction. And the result in Step 2 on $V^{(2)}$ continues to encourage $W^{(3)}$ to continue placing a high weighting on $x_{N-1}$ (see Lemma D.8 in the appendix).

From the three gradient descent steps listed above, it is clear that $V^{(t)}$ will converge to the direction of $\Pi_2$, and $W^{(t)}$ will consistently place a high weighting on the most 'informative' token $x_{N-1}$. This is our key intuition for proving Theorem 3.1, and in our formal proof, we use an induction to characterize the whole training procedure.

**Training dynamics in learning deterministic walks.** We can also consider the training dynamics in learning deterministic walks. Starting from zero initialization, we can easily verify the following two initial gradient steps:

Step 1. Since the initial softmax weightings on all tokens are the same, $V^{(t)}$ is essentially trained based on the averaged token $\overline{x} = \frac{1}{N-1} \sum_{i=1}^{N-1} x_i$. Importantly, by definition we see that

$\overline{x}$ *is a constant vector that does not depend on the random initial location.*

It can then be shown that all entries in $V^{(1)}$ are equal (see Lemma E.1 in the appendix). $W^{(1)}$ is still a zero matrix due to the fact that $V^{(0)} = \mathbf{0}$.

Step 2. With the same analysis as Step 1, we can show that all entries in $V^{(2)}$ are equal. Moreover, due to the fact that the tokens are 'equally informative', it can be further shown that $W^{(2)}$ is updated so that the softmax weightings on all tokens $x_1, \ldots, x_{N-1}$ remain equal.

The above two steps clearly match our discussion in Section 5 on the reason transformers fail to learn the deterministic walk: *the deterministic walk is such a task, that each individual token can grant perfect prediction, but the average of the tokens provides no useful information.* We can then inductively show that throughout training, the value matrix $V^{(t)}$ is always proportional to the all-one matrix, and the softmax weights on all tokens are always the same.

## C GRADIENT DESCENT

Recall that the perturbed population loss is

$$L(\theta) = \mathbb{E}[\ell(\theta)] = \mathbb{E}[-\log(e_y^\top f_\theta(X) + \epsilon)] = \mathbb{E}[-\log(e_y^\top VX\mathcal{S}(\widetilde{X}^\top W\widetilde{x}_N) + \epsilon)].$$

We can compute the gradients as follows.

**Lemma C.1.** The gradients regarding $V$ and $W$ are

$$\nabla_V \ell(\theta) = -\frac{1}{e_y^\top VX\mathcal{S} + \epsilon} \cdot e_y \sum_{i=1}^{N-1} \mathcal{S}_i x_i^\top,$$

$$\nabla_W \ell(\theta) = -\frac{1}{e_y^\top VX\mathcal{S} + \epsilon} \cdot \begin{bmatrix} \mathbf{0} & \left(\sum_{i=1}^{N-1} \mathcal{S}_i x_i x_i^\top - \sum_{i=1}^{N-1} \mathcal{S}_i x_i \cdot \sum_{i=1}^{N-1} \mathcal{S}_i x_i^\top\right) V^\top e_y p_N^\top \\ \mathbf{0} & \left(\sum_{i=1}^{N-1} \mathcal{S}_i p_i x_i^\top - \sum_{i=1}^{N} \mathcal{S}_i p_i \cdot \sum_{i=1}^{N-1} \mathcal{S}_i x_i^\top\right) V^\top e_y p_N^\top \end{bmatrix},$$

where $\mathcal{S} = \mathcal{S}(\widetilde{X}^\top W\widetilde{x}_N)$, and $\mathcal{S}_i$ is the $i$-th element of $\mathcal{S}$.

***Proof of Lemma C.1.*** For $V$, we have

$$\nabla_V \ell(\theta) = -\frac{1}{e_y^\top f_\theta(X) + \epsilon} \cdot \frac{\partial e_y^\top VX\mathcal{S}}{\partial V}$$

$$= -\frac{1}{e_y^\top VX\mathcal{S} + \epsilon} \cdot e_y \mathcal{S}^\top X^\top$$

$$= -\frac{1}{e_y^\top VX\mathcal{S} + \epsilon} \cdot e_y \sum_{i=1}^{N-1} \mathcal{S}_i x_i^\top.$$

For $W$, we have

$$\nabla_W \ell(\theta) = -\frac{1}{e_y^\top f_\theta(X) + \epsilon} \cdot \frac{\partial e_y^\top VX\mathcal{S}(\widetilde{X}^\top W\widetilde{x}_N)}{\partial W}$$

$$= -\frac{1}{e_y^\top VXS + \epsilon} \cdot \widetilde{X} S'(\widetilde{X}^\top W\widetilde{x}_N) X^\top V^\top e_y \widetilde{x}_N^\top$$

$$= -\frac{1}{e_y^\top VXS + \epsilon} \cdot \widetilde{X}[\mathrm{diag}(S) - SS^\top]X^\top V^\top e_y \widetilde{x}_N^\top$$

$$= -\frac{1}{e_y^\top VXS + \epsilon} \cdot \begin{bmatrix} \sum_{i=1}^{N-1} S_i x_i x_i^\top - \sum_{i=1}^{N-1} S_i x_i \cdot \sum_{j=1}^{N-1} S_j x_i^\top \\ \sum_{i=1}^{N} S_i p_i x_i^\top - \sum_{i=1}^{N} S_i p_i \cdot \sum_{i=1}^{N-1} S_i x_i^\top \end{bmatrix} \cdot \begin{bmatrix} \mathbf{0} & V^\top e_y p_N^\top \end{bmatrix}$$

$$= -\frac{1}{e_y^\top VXS + \epsilon} \cdot \begin{bmatrix} \mathbf{0} & (\sum_{i=1}^{N-1} S_i x_i x_i^\top - \sum_{i=1}^{N-1} S_i x_i \cdot \sum_{i=1}^{N-1} S_i x_i^\top)V^\top e_y p_N^\top \\ \mathbf{0} & (\sum_{i=1}^{N-1} S_i p_i x_i^\top - \sum_{i=1}^{N} S_i p_i \cdot \sum_{i=1}^{N-1} S_i x_i^\top)V^\top e_y p_N^\top \end{bmatrix},$$

where we use the fact that $S'(\widetilde{X}^\top W\widetilde{x}_N) = [\mathrm{diag}(S) - SS^\top]$ and $\mathrm{diag}(S) :=$

$$\begin{bmatrix} S_1 & & & \\ & S_2 & & \\ & & \ddots & \\ & & & S_N \end{bmatrix}.$$

$\square$

To simplify the notation, we denote

$$\nabla_W \ell(\theta) = -\frac{1}{e_y^\top VXS + \epsilon} \cdot \begin{bmatrix} \mathbf{0} & (\sum_{i=1}^{N-1} S_i x_i x_i^\top - \sum_{i=1}^{N-1} S_i x_i \cdot \sum_{i=1}^{N-1} S_i x_i^\top)V^\top e_y p_N^\top \\ \mathbf{0} & (\sum_{i=1}^{N-1} S_i p_i x_i^\top - \sum_{i=1}^{N} S_i p_i \cdot \sum_{i=1}^{N-1} S_i x_i^\top)V^\top e_y p_N^\top \end{bmatrix}$$

$$:= -\frac{1}{e_y^\top VXS + \epsilon} \cdot \begin{bmatrix} \mathbf{0} & A \\ \mathbf{0} & B \end{bmatrix}, \tag{C.1}$$

and $W = \begin{bmatrix} W_{11} & W_{12} \\ W_{21} & W_{22} \end{bmatrix}$, where $W_{11} \in \mathbb{R}^{K \times K}$, $W_{12} \in \mathbb{R}^{K \times M}$, $W_{21} \in \mathbb{R}^{M \times K}$, and $W_{22} \in \mathbb{R}^{M \times M}$. By (C.1), we know that $W_{11}^{(t)} = \mathbf{0}_{K \times K}$ and $W_{21}^{(t)} = \mathbf{0}_{M \times K}$ for all $t \geq 1$.

By the definition of the transition matrix, we can write the transition matrices of Task 1 and Task 2 as $\Pi_1 = \frac{1}{2}\Pi_0 + \frac{1}{2}\Pi_0^\top$ and $\Pi_2 = \Pi_0$, where

$$\Pi_0 = \begin{bmatrix} 0 & & & & 1 \\ 1 & 0 & & & \\ & 1 & 0 & & \\ & & \ddots & \ddots & \\ & & & 1 & 0 \end{bmatrix}.$$

# D  TASK 1: RANDOM WALK

In this section, we consider the case of the random walk. We assume that the transition matrix is $\Pi = \Pi_1$, which means $y$ is generated by the transition probability $\Pi_1^\top x_{N-1}$. The following lemma presents the result of the first iteration.

**Lemma D.1.** If $\Pi = \Pi_1$, it holds that

$$V^{(1)} = \frac{\eta}{\epsilon NK} \sum_{i=1}^{N-1} \Pi_1^{N-i} \text{ and } W^{(1)} = \mathbf{0}_{(K+M) \times (K+M)}.$$

***Proof of Lemma D.1.*** By Lemma C.1, we have

$$\mathbb{E}[\nabla_V \ell(\theta^{(0)})] = -\frac{1}{\epsilon N} \sum_{i=1}^{N-1} \mathbb{E}[e_y x_i^\top]$$

$$= -\frac{1}{\epsilon N} \sum_{i=1}^{N-1} \mathbb{E}[(\Pi^\top)^{N-i} x_i x_i^\top]$$

$$= -\frac{1}{\epsilon N K} \sum_{i=1}^{N-1} (\mathbf{\Pi}_1^\top)^{N-i}$$

$$= -\frac{1}{\epsilon N K} \sum_{i=1}^{N-1} \mathbf{\Pi}_1^{N-i}$$

where the first equation is by the initialization of $\boldsymbol{V}^{(0)}$ and $\boldsymbol{W}^{(0)}$, the second equation is by the sampling method, the third equation is by $\mathbb{E}[\boldsymbol{x}_i \boldsymbol{x}_i^\top] = \frac{1}{K}\mathbf{I}_K$ for $i \in [N-1]$ since $\boldsymbol{x}_i$ is uniformly distributed in $\boldsymbol{E}$, and the last equation is by $\mathbf{\Pi}_1 = \mathbf{\Pi}_1^\top$. Thus, by the update, we can get

$$\boldsymbol{V}^{(1)} = \boldsymbol{V}^{(0)} - \eta \mathbb{E}[\nabla_{\boldsymbol{V}} \ell(\theta^{(0)})]$$

$$= \frac{\eta}{\epsilon N K} \sum_{i=1}^{N-1} \mathbf{\Pi}_1^{N-i}.$$

Since $\boldsymbol{V}^{(0)} = \mathbf{0}_{K \times K}$ and $\boldsymbol{W}^{(0)} = \mathbf{0}_{(K+M) \times (K+M)}$, we can get $\mathbb{E}[\nabla_{\boldsymbol{W}} \ell(\theta^{(0)})] = \mathbf{0}_{(K+M) \times (K+M)}$. Thus,

$$\boldsymbol{W}^{(1)} = \boldsymbol{W}^{(0)} - \eta \mathbb{E}[\nabla_{\boldsymbol{W}} \ell(\theta^{(0)})] = \mathbf{0}_{(K+M) \times (K+M)}.$$

$\square$

In Lemma D.2, Lemma D.3, and Lemma D.4, all the index $p$ of matrices and vectors represent $p'$, where $p \equiv p' \pmod{K}$ and $1 \le p' \le K$. And, these three lemmas provide some properties of $\boldsymbol{V}^{(t)}$.

**Lemma D.2.** If $\mathbf{\Pi} = \mathbf{\Pi}_1$, then it holds that $\boldsymbol{V}^{(1)}$ is a symmetric matrix and $[\boldsymbol{V}^{(1)}]_{i,j} = [\boldsymbol{V}^{(1)}]_{i,2i-j}$.

***Proof of Lemma D.2.*** We use induction to prove that for any $R \in \mathbb{N}$ and $i, j \in [K]$, $\mathbf{\Pi}_1^R$ is a symmetric matrix and $[\mathbf{\Pi}_1^R]_{i,j} = [\mathbf{\Pi}_1^R]_{i,2i-j}$. The results are obvious at $R = 1$. Suppose that the results hold for $\mathbf{\Pi}_1^R$. We aim to prove the results hold for $\mathbf{\Pi}_1^{R+1}$. Since $\mathbf{\Pi}_1^{R+1} = \mathbf{\Pi}_1^R \cdot \mathbf{\Pi}_1$, we have $[\mathbf{\Pi}_1^{R+1}]_{i,j} = \frac{1}{2}\left([\mathbf{\Pi}_1^R]_{i,j-1} + [\mathbf{\Pi}_1^R]_{i,j+1}\right) = \frac{1}{2}\left([\mathbf{\Pi}_1^R]_{i-1,j} + [\mathbf{\Pi}_1^R]_{i+1,j}\right)$. Thus, we can get that

$$[\mathbf{\Pi}_1^{R+1}]_{i,j} = \frac{1}{2}\left([\mathbf{\Pi}_1^R]_{i,j-1} + [\mathbf{\Pi}_1^R]_{i,j+1}\right)$$

$$= \frac{1}{2}\left([\mathbf{\Pi}_1^R]_{j-1,i} + [\mathbf{\Pi}_1^R]_{j+1,i}\right)$$

$$= [\mathbf{\Pi}_1^{R+1}]_{j,i},$$

and

$$[\mathbf{\Pi}_1^{R+1}]_{i,j} = \frac{1}{2}\left([\mathbf{\Pi}_1^R]_{i,j-1} + [\mathbf{\Pi}_1^R]_{i,j+1}\right)$$

$$= \frac{1}{2}\left([\mathbf{\Pi}_1^R]_{i,2i-j+1} + [\mathbf{\Pi}_1^R]_{i,2i-j-1}\right)$$

$$= [\mathbf{\Pi}_1^{R+1}]_{i,2i-j},$$

which completes the induction. By Lemma D.1, we know $\boldsymbol{V}^{(1)} = \frac{\eta}{\epsilon N K} \sum_{i=1}^{N-1} \mathbf{\Pi}_1^{N-i}$. Thus, $\boldsymbol{V}^{(1)}$ also has those properties. $\square$

**Lemma D.3.** If $\mathbf{\Pi} = \mathbf{\Pi}_1$ and $K$ is even, then $[\boldsymbol{V}^{(1)}]_{1,2} = \|\boldsymbol{V}^{(1)}\|_{\max}$.

***Proof of Lemma D.3.*** By Lemma D.2, we know that there are only $K/2 + 1$ different values in $\mathbf{\Pi}_1^t$ for all $t \in \mathbb{N}$. Denote that $a_j^{(t)} = [\mathbf{\Pi}_1^t]_{1,j+1}$ for $j \in \{0, 1, \ldots, K/2\}$. We are going to prove that $a_1^{(2k-1)} + a_1^{(2k)} \ge a_j^{(2k-1)} + a_j^{(2k)}$ for $k \in \mathbb{N}$ and $j \in \{0, 1, \ldots, K/2\}$. We use induction to prove that for $k \in \mathbb{N}$,

$$a_{2l_1-1}^{(2k-1)} \ge a_{2l_2-1}^{(2k-1)} \text{ for } l_1 < l_2 \text{ and } a_{2l}^{(2k-1)} = 0 \text{ for } l \in \mathbb{N};$$

$$a_{2l}^{(2k)} \le a_1^{(2k-1)} \text{ and } a_{2l-1}^{(2k)} = 0 \text{ for } l \in \mathbb{N}.$$

When $k = 1$, we have

$$a_1^{(1)} = 0.5, \text{ and } a_j^{(1)} = 0 \text{ for } j \neq 1;$$

$$a_0^{(2)} = 0.5, a_2^{(2)} = 0.25, \text{ and } a_j^{(2)} = 0 \text{ for } j \neq 0, 2.$$

We can easily check that the results hold for $k = 1$. Suppose that the results hold for $k = k'$. Then, we consider the condition of $k = k' + 1$. First, we can get that for $t \in \mathbb{N}$ and $j \in \{0, 1, \ldots, K/2\}$,

$$a_j^{(t+1)} = [\mathbf{\Pi}_1^{t+1}]_{1,j+1} = \frac{1}{2} \left( [\mathbf{\Pi}_1^t]_{1,j} + [\mathbf{\Pi}_1^t]_{1,j+2} \right) = \frac{1}{2} \left( a_{j-1}^{(t)} + a_{j+1}^{(t)} \right),$$

where $a_{-1}^{(t)} = a_1^{(t)}$ and $a_{K/2+1}^{(t)} = a_{K/2-1}^{(t)}$. Then, we can get that for $t \in \mathbb{N}$ and $j \in \{0, 1, \ldots, K/2\}$,

$$a_j^{(t+2)} = \frac{1}{2} \left( a_{j-1}^{(t+1)} + a_{j+1}^{(t+1)} \right) = \frac{1}{4} \left( a_{j-2}^{(t+1)} + 2a_j^{(t+1)} + a_{j+2}^{(t+1)} \right), \tag{D.1}$$

where $a_{-2}^{(t)} = a_2^{(t)}, a_{-1}^{(t)} = a_1^{(t)}, a_{K/2+1}^{(t)} = a_{K/2-1}^{(t)}$, and $a_{K/2+2}^{(t)} = a_{K/2-2}^{(t)}$. Thus, by (D.1), we have

$$a_{2l}^{(2k'+1)} = \frac{1}{4} \left( a_{2l-2}^{(2k'-1)} + 2a_{2l}^{(2k'-1)} + a_{2l+2}^{(2k'-1)} \right) = 0,$$

$$a_{2l-1}^{(2k'+2)} = \frac{1}{4} \left( a_{2l-3}^{(2k')} + 2a_{2l-1}^{(2k')} + a_{2l+1}^{(2k')} \right) = 0.$$

And, we have

$$a_{2l-1}^{(2k'+1)} - a_{2l+1}^{(2k'+1)} = \frac{1}{4} \left( a_{2l-3}^{(2k'-1)} + 2a_{2l-1}^{(2k'-1)} + a_{2l+1}^{(2k'-1)} \right)$$

$$- \frac{1}{4} \left( a_{2l-1}^{(2k'-1)} + 2a_{2l+1}^{(2k'-1)} + a_{2l+3}^{(2k'-1)} \right)$$

$$= \frac{1}{4} \left( a_{2l-3}^{(2k'-1)} + a_{2l-1}^{(2k'-1)} - a_{2l+1}^{(2k'-1)} - a_{2l+3}^{(2k'-1)} \right)$$

$$\geq 0,$$

where the inequality is by the induction and $a_{-1}^{(t)} = a_1^{(t)}, a_{K/2+1}^{(t)} = a_{K/2-1}^{(t)}, a_{K/2+2}^{(t)} = a_{K/2-2}^{(t)}$. This implies that $a_{2l_1-1}^{(2k'+1)} \geq a_{2l_2-1}^{(2k'+1)}$ for $l_1 < l_2$. Then, we also have

$$a_{2l}^{(2k'+2)} = \frac{1}{2} \left( a_{2l-1}^{(2k'+1)} + 2a_{2l+1}^{(2k'+1)} \right) \leq a_1^{(2k'+1)}.$$

Therefore, we prove that the results hold at $k = k' + 1$, which completes the proof. $\qquad\square$

**Lemma D.4.** If $\mathbf{\Pi} = \mathbf{\Pi}_1$, then it holds that for all $t \in \mathbb{N}$, $\left[ \mathbf{V}^{(t)} \right]_{i,i-k} = \left[ \mathbf{V}^{(t)} \right]_{i,i+k}$ and $\left[ \mathbf{V}^{(t)} \right]_{i_1,i_1-k} = \left[ \mathbf{V}^{(t)} \right]_{i_2,i_2-k}$ for $i, i_1, i_2, k \in \mathbb{N}$. Further, $\mathbf{V}^{(t)}$ is a symmetric matrix.

*Proof of Lemma D.4.* We use induction to prove the results. By Lemma D.2, we can easily check that $\mathbf{V}^{(1)}$ has the properties stated in Lemma D.4. Suppose that $\left[ \mathbf{V}^{(t)} \right]_{i,i-k} = \left[ \mathbf{V}^{(t)} \right]_{i,i+k}$ and $\left[ \mathbf{V}^{(t)} \right]_{i_1,i_2-k} = \left[ \mathbf{V}^{(t)} \right]_{i_2,i_2-k}$ for $i, i_1, i_2, k \in \mathbb{N}$. For $\mathbf{V}^{(t+1)}$, we first have

$$\mathbf{V}^{(t+1)} = \mathbf{V}^{(t)} - \eta \mathbb{E}[\nabla_{\mathbf{V}} \ell(\theta^{(t)})] = \mathbf{V}^{(t)} + \eta \mathbb{E} \left[ \frac{\mathbf{e}_y \sum_{i=1}^{N-1} \mathcal{S}_i^{(t)} \mathbf{x}_i^\top}{\mathbf{e}_y^\top \mathbf{V}^{(t)} \sum_{i=1}^{N-1} \mathcal{S}_i^{(t)} \mathbf{x}_i + \epsilon} \right].$$

Denote that $\mathbf{\Pi}(y) = \sum_{j=0}^{K-1} \mathbf{e}_{y-j} \mathbf{e}_{y+j}^\top$ which has the property that $\mathbf{\Pi}(y)^\top \mathbf{e}_i = \mathbf{e}_{2y-i}$. By the sampling method, we know that

$$(\mathbf{x}_1, \mathbf{x}_2, \ldots, \mathbf{x}_{N-1}, \mathbf{e}_y) \overset{d}{=} (\mathbf{\Pi}(y)\mathbf{x}_1, \mathbf{\Pi}(y)\mathbf{x}_2, \ldots, \mathbf{\Pi}(y)\mathbf{x}_{N-1}, \mathbf{e}_y), \tag{D.2}$$

$$(\mathbf{x}_1, \mathbf{x}_2, \ldots, \mathbf{x}_{N-1}, \mathbf{e}_y) \overset{d}{=} (\mathbf{\Pi}_0^l \mathbf{x}_1, \mathbf{\Pi}_0^l \mathbf{x}_2, \ldots, \mathbf{\Pi}_0^l \mathbf{x}_{N-1}, \mathbf{\Pi}_0^l \mathbf{e}_y) \text{ for } l \in \mathbb{N}. \tag{D.3}$$

Thus, we can get

$$\mathbf{e}_i^\top \mathbf{V}^{(t+1)} \mathbf{e}_{i-k} = \mathbf{e}_i^\top \mathbf{V}^{(t)} \mathbf{e}_{i-k} + \eta \mathbb{E} \left[ \frac{\mathbf{e}_i^\top \mathbf{e}_y \sum_{i'=1}^{N-1} \mathcal{S}_{i'}^{(t)} \mathbf{x}_{i'}^\top \mathbf{e}_{i-k}}{\mathbf{e}_y^\top \mathbf{V}^{(t)} \sum_{i'=1}^{N-1} \mathcal{S}_{i'}^{(t)} \mathbf{x}_{i'} + \epsilon} \right]$$

$$= \boldsymbol{e}_i^\top \boldsymbol{V}^{(t)} \boldsymbol{e}_{i-k} + \eta \mathbb{E} \left[ \frac{\sum_{i'=1}^{N-1} \mathcal{S}_{i'}^{(t)} \boldsymbol{x}_{i'}^\top \boldsymbol{e}_{i-k}}{\boldsymbol{e}_y^\top \boldsymbol{V}^{(t)} \sum_{i'=1}^{N-1} \mathcal{S}_{i'}^{(t)} \boldsymbol{x}_{i'} + \epsilon} \cdot \mathbf{1}\{y=i\} \right]$$

$$= \boldsymbol{e}_i^\top \boldsymbol{V}^{(t)} \boldsymbol{e}_{i-k} + \eta \mathbb{E} \left[ \frac{\sum_{i'=1}^{N-1} \mathcal{S}_{i'}^{(t)} \boldsymbol{x}_{i'}^\top \boldsymbol{\Pi}(y)^\top \boldsymbol{e}_{i-k}}{\boldsymbol{e}_y^\top \boldsymbol{V}^{(t)} \boldsymbol{\Pi}(y) \sum_{i'=1}^{N-1} \mathcal{S}_{i'}^{(t)} \boldsymbol{x}_{i'} + \epsilon} \cdot \mathbf{1}\{y=i\} \right]$$

$$= \boldsymbol{e}_i^\top \boldsymbol{V}^{(t)} \boldsymbol{e}_{i+k} + \eta \mathbb{E} \left[ \frac{\sum_{i'=1}^{N-1} \mathcal{S}_{i'}^{(t)} \boldsymbol{x}_{i'}^\top \boldsymbol{e}_{i+k}}{\boldsymbol{e}_y^\top \boldsymbol{V}^{(t)} \sum_{i'=1}^{N-1} \mathcal{S}_{i'}^{(t)} \boldsymbol{x}_{i'} + \epsilon} \cdot \mathbf{1}\{y=i\} \right]$$

$$= \boldsymbol{e}_i^\top \boldsymbol{V}^{(t)} \boldsymbol{e}_{i+k} + \eta \mathbb{E} \left[ \frac{\boldsymbol{e}_i^\top \boldsymbol{e}_y \sum_{i'=1}^{N-1} \mathcal{S}_{i'}^{(t)} \boldsymbol{x}_{i'}^\top \boldsymbol{e}_{i+k}}{\boldsymbol{e}_y^\top \boldsymbol{V}^{(t)} \sum_{i'=1}^{N-1} \mathcal{S}_{i'}^{(t)} \boldsymbol{x}_{i'} + \epsilon} \right]$$

$$= \boldsymbol{e}_i^\top \boldsymbol{V}^{(t+1)} \boldsymbol{e}_{i+k},$$

where the third equation is by (D.2) and the fourth equation is by induction. And, we can get

$$\boldsymbol{e}_{i_1}^\top \boldsymbol{V}^{(t+1)} \boldsymbol{e}_{i_1-k} = \boldsymbol{e}_{i_1}^\top \boldsymbol{V}^{(t)} \boldsymbol{e}_{i_1-k} + \eta \mathbb{E} \left[ \frac{\boldsymbol{e}_{i_1}^\top \boldsymbol{e}_y \sum_{i'=1}^{N-1} \mathcal{S}_{i'}^{(t)} \boldsymbol{x}_{i'}^\top \boldsymbol{e}_{i_1-k}}{\boldsymbol{e}_y^\top \boldsymbol{V}^{(t)} \sum_{i'=1}^{N-1} \mathcal{S}_{i'}^{(t)} \boldsymbol{x}_{i'} + \epsilon} \right]$$

$$= \boldsymbol{e}_{i_1}^\top \boldsymbol{V}^{(t)} \boldsymbol{e}_{i_1-k} + \eta \mathbb{E} \left[ \frac{\boldsymbol{e}_{i_1}^\top \boldsymbol{\Pi}_0^{i_2-i_1} \boldsymbol{e}_y \sum_{i'=1}^{N-1} \mathcal{S}_{i'}^{(t)} \boldsymbol{x}_{i'}^\top (\boldsymbol{\Pi}_0^\top)^{i_2-i_1} \boldsymbol{e}_{i_1-k}}{\boldsymbol{e}_y^\top (\boldsymbol{\Pi}_0^\top)^{i_2-i_1} \boldsymbol{V}^{(t)} \boldsymbol{\Pi}_0^{i_2-i_1} \sum_{i'=1}^{N-1} \mathcal{S}_{i'}^{(t)} \boldsymbol{x}_{i'} + \epsilon} \right]$$

$$= \boldsymbol{e}_{i_2}^\top \boldsymbol{V}^{(t)} \boldsymbol{e}_{i_2-k} + \eta \mathbb{E} \left[ \frac{\boldsymbol{e}_{i_2}^\top \boldsymbol{e}_y \sum_{i'=1}^{N-1} \mathcal{S}_{i'}^{(t)} \boldsymbol{x}_{i'}^\top \boldsymbol{e}_{i_2-k}}{\boldsymbol{e}_y^\top \boldsymbol{V}^{(t)} \sum_{i'=1}^{N-1} \mathcal{S}_{i'}^{(t)} \boldsymbol{x}_{i'} + \epsilon} \right]$$

$$= \boldsymbol{e}_{i_2}^\top \boldsymbol{V}^{(t+1)} \boldsymbol{e}_{i_2-k},$$

where the second equation is by (D.3) and the third equation is by induction. Therefore, we prove that the results hold for $\boldsymbol{V}^{(t+1)}$, which completes the proof. Further, we can get $[\boldsymbol{V}^{(t)}]_{i,i+k} = [\boldsymbol{V}^{(t)}]_{i,i-k} = [\boldsymbol{V}^{(t)}]_{i+k,i}$, which implies that $\boldsymbol{V}^{(t)}$ is symmetric. $\qquad\square$

The following two lemmas provide the properties of the weights for the second iteration.

**Lemma D.5.** If $\boldsymbol{\Pi} = \boldsymbol{\Pi}_1$, then it holds that $\|\boldsymbol{V}^{(2)}\|_{\max} \leq \frac{\eta}{\epsilon K} + 2\epsilon K^2$.

*Proof of Lemma D.5.* First, we have

$$\boldsymbol{V}^{(2)} = \boldsymbol{V}^{(1)} - \eta \mathbb{E}[\nabla_{\boldsymbol{V}} \ell(\theta^{(1)})] = \boldsymbol{V}^{(1)} + \eta \mathbb{E} \left[ \frac{\boldsymbol{e}_y \sum_{i=1}^{N-1} \mathcal{S}_i^{(1)} \boldsymbol{x}_i^\top}{\boldsymbol{e}_y^\top \boldsymbol{V}^{(1)} \sum_{i=1}^{N-1} \mathcal{S}_i^{(1)} \boldsymbol{x}_i + \epsilon} \right].$$

Thus,

$$\|\boldsymbol{V}^{(2)}\|_{\max} \leq \left\| \boldsymbol{V}^{(1)} \right\|_{\max} + \left\| \eta \mathbb{E} \left[ \frac{\frac{1}{N} \boldsymbol{e}_y \sum_{i=1}^{N-1} \boldsymbol{x}_i^\top}{\frac{1}{N} \boldsymbol{e}_y^\top \boldsymbol{V}^{(1)} \sum_{i=1}^{N-1} \boldsymbol{x}_i + \epsilon} \right] \right\|_{\max}$$

$$\leq \left\| \boldsymbol{V}^{(1)} \right\|_{\max} + \frac{\eta}{\frac{N-1}{N} \min_{i,j} [\boldsymbol{V}^{(1)}]_{i,j}} \cdot \left\| \mathbb{E} \left[ \frac{1}{N} \boldsymbol{e}_y \sum_{i=1}^{N-1} \boldsymbol{x}_i^\top \right] \right\|_{\max}$$

$$\leq \left\| \frac{\eta}{\epsilon N K} \sum_{i=1}^{N-1} \boldsymbol{\Pi}_1^{N-i} \right\|_{\max} + \frac{\eta}{\frac{N-1}{N} \min_{i,j} \left[ \frac{\eta}{\epsilon N K} \sum_{i=1}^{N-1} \boldsymbol{\Pi}_1^{N-i} \right]_{i,j}} \cdot \frac{N-1}{N}$$

$$\leq \frac{\eta}{\epsilon K} + 2\epsilon K^2,$$

where the second inequality is by $\boldsymbol{e}_y^\top \boldsymbol{V}^{(1)} \boldsymbol{x}_i \geq \min_{i,j} [\boldsymbol{V}^{(1)}]_{i,j}$, and the last inequality is by Lemma F.4. $\qquad\square$

**Lemma D.6.** If $\boldsymbol{\Pi} = \boldsymbol{\Pi}_1$ and $K$ is even, it holds that $\mathcal{S}_{N-1}^{(2)} \geq \mathcal{S}_j^{(2)} \exp(\Omega(N))$ for $j \neq N-1$. Further, $\mathcal{S}_{N-1}^{(2)} \geq 1 - \exp(-\Omega(N))$ and $\mathcal{S}_j^{(2)} \leq \exp(-\Omega(N))$ for $j \neq N-1$.

**Proof of Lemma D.6.** By Lemma C.1, we have

$$\mathbb{E}[\boldsymbol{A}^{(1)}]$$

$$= \mathbb{E}\left[\left(\sum_{i=1}^{N-1} \mathcal{S}_i^{(1)} \boldsymbol{x}_i \boldsymbol{x}_i^\top (\boldsymbol{V}^{(1)})^\top \boldsymbol{e}_y - \sum_{i_1=1}^{N-1}\sum_{i_2=1}^{N-1} \mathcal{S}_{i_1}^{(1)} \mathcal{S}_{i_2}^{(1)} \boldsymbol{x}_{i_1} \boldsymbol{x}_{i_2}^\top (\boldsymbol{V}^{(1)})^\top \boldsymbol{e}_y \right) \boldsymbol{p}_N^\top \right]$$

$$= \mathbb{E}\left[\left(\frac{\eta}{\epsilon N^2 K}\sum_{i=1}^{N-1}\boldsymbol{x}_i\boldsymbol{x}_i^\top \sum_{i'=1}^{N-1}\boldsymbol{\Pi}_1^{N-i'}\boldsymbol{e}_y - \frac{\eta}{\epsilon N^3 K}\sum_{i_1=1}^{N-1}\sum_{i_2=1}^{N-1}\boldsymbol{x}_{i_1}\boldsymbol{x}_{i_2}^\top \sum_{i'=1}^{N-1}\boldsymbol{\Pi}_1^{N-i'}\boldsymbol{e}_y \right)\boldsymbol{p}_N^\top\right]$$

$$= \mathbb{E}\left[\frac{\eta}{\epsilon N^2 K}\sum_{i=1}^{N-1}\boldsymbol{x}_i\boldsymbol{x}_i^\top \sum_{i'=1}^{N-1}\boldsymbol{\Pi}_1^{N-i'}(\boldsymbol{\Pi}_1^\top)^{N-i}\boldsymbol{x}_i \cdot \boldsymbol{p}_N^\top\right]$$

$$\quad - \mathbb{E}\left[\frac{\eta}{\epsilon N^3 K}\sum_{i_1=1}^{N-1}\sum_{i_2=1}^{N-1}\boldsymbol{x}_{i_1}\boldsymbol{x}_{i_1}^\top \boldsymbol{\Pi}_1^{i_2-i_1}\sum_{i'=1}^{N-1}\boldsymbol{\Pi}_1^{N-i'}(\boldsymbol{\Pi}_1^\top)^{N-i_1}\boldsymbol{x}_{i_1}\cdot\boldsymbol{p}_N^\top\right]$$

$$= \mathbb{E}\left[\frac{\eta}{\epsilon N^2 K}\sum_{i=1}^{N-1}\sum_{i'=1}^{N-1}\boldsymbol{x}_i\boldsymbol{x}_i^\top\boldsymbol{\Pi}_1^{2N-i'-i}\boldsymbol{x}_i\cdot\boldsymbol{p}_N^\top\right]$$

$$\quad - \mathbb{E}\left[\frac{\eta}{\epsilon N^3 K}\sum_{i_1=1}^{N-1}\sum_{i_2=1}^{N-1}\sum_{i'=1}^{N-1}\boldsymbol{x}_{i_1}\boldsymbol{x}_{i_1}^\top\boldsymbol{\Pi}_1^{2N-i'+i_2-2i_1}\boldsymbol{x}_{i_1}\cdot\boldsymbol{p}_N^\top\right]$$

$$= \left(\frac{\eta}{\epsilon N^2 K^2}\sum_{i=1}^{N-1}\sum_{i'=1}^{N-1}\mathrm{tr}(\boldsymbol{\Pi}_1^{2N-i'-i})\boldsymbol{1}_K - \frac{\eta}{\epsilon N^3 K^2}\sum_{i_1=1}^{N-1}\sum_{i_2=1}^{N-1}\sum_{i'=1}^{N-1}\mathrm{tr}(\boldsymbol{\Pi}_1^{2N-i'+i_2-2i_1})\boldsymbol{1}_K\right)\boldsymbol{p}_N^\top,$$

where the second equation is by Lemma D.1, the third equation is by the sampling method, the fourth equation is by $\boldsymbol{\Pi}_1 = \boldsymbol{\Pi}_1^\top$, and the fifth equation is by the fact that all the $\boldsymbol{x}_i$ is uniformly distributed in $\boldsymbol{E}$ for $i \in [N-1]$. Then, $\boldsymbol{W}_{12}^{(2)} = \boldsymbol{W}_{12}^{(1)} - \eta\mathbb{E}[\nabla_{\boldsymbol{W}}\ell(\theta^{(1)})]_{12} \propto \boldsymbol{1}_K\boldsymbol{p}_N^\top$. Thus, We also have

$$\mathbb{E}[\boldsymbol{B}^{(1)}]$$

$$= \mathbb{E}\left[\left(\sum_{i=1}^{N-1}\mathcal{S}_i^{(1)}\boldsymbol{p}_i\boldsymbol{x}_i^\top(\boldsymbol{V}^{(1)})^\top\boldsymbol{e}_y - \sum_{i=1}^{N}\mathcal{S}_i^{(1)}\boldsymbol{p}_i\cdot\sum_{i=1}^{N-1}\mathcal{S}_i^{(1)}\boldsymbol{x}_i^\top(\boldsymbol{V}^{(1)})^\top\boldsymbol{e}_y\right)\boldsymbol{p}_N^\top\right]$$

$$= \mathbb{E}\left[\left(\frac{\eta}{\epsilon N^2 K}\sum_{i=1}^{N-1}\boldsymbol{p}_i\boldsymbol{x}_i^\top\sum_{i'=1}^{N-1}\boldsymbol{\Pi}_1^{N-i'}\boldsymbol{e}_y - \frac{\eta}{\epsilon N^3 K}\sum_{i=1}^{N}\boldsymbol{p}_i\cdot\sum_{i=1}^{N-1}\boldsymbol{x}_i^\top\sum_{i'=1}^{N-1}\boldsymbol{\Pi}_1^{N-i'}\boldsymbol{e}_y\right)\boldsymbol{p}_N^\top\right]$$

$$= \mathbb{E}\left[\frac{\eta}{\epsilon N^2 K}\sum_{i=1}^{N-1}\boldsymbol{p}_i\boldsymbol{x}_i^\top\sum_{i'=1}^{N-1}\boldsymbol{\Pi}_1^{N-i'}(\boldsymbol{\Pi}_1^\top)^{N-i}\boldsymbol{x}_i\cdot\boldsymbol{p}_N^\top\right]$$

$$\quad - \mathbb{E}\left[\frac{\eta}{\epsilon N^3 K}\sum_{i=1}^{N}\boldsymbol{p}_i\cdot\sum_{i=1}^{N-1}\boldsymbol{x}_i^\top\sum_{i'=1}^{N-1}\boldsymbol{\Pi}_1^{N-i'}(\boldsymbol{\Pi}_1^\top)^{N-i}\boldsymbol{x}_i\cdot\boldsymbol{p}_N^\top\right]$$

$$= \mathbb{E}\left[\left(\frac{\eta}{\epsilon N^2 K}\sum_{i=1}^{N-1}\sum_{i'=1}^{N-1}\boldsymbol{p}_i\boldsymbol{x}_i^\top\boldsymbol{\Pi}_1^{2N-i'-i}\boldsymbol{x}_i - \frac{\eta}{\epsilon N^3 K}\sum_{i=1}^{N}\boldsymbol{p}_i\cdot\sum_{i=1}^{N-1}\sum_{i'=1}^{N-1}\boldsymbol{x}_i^\top\boldsymbol{\Pi}_1^{2N-i'-i}\boldsymbol{x}_i\right)\boldsymbol{p}_N^\top\right]$$

$$= \left(\frac{\eta}{\epsilon N^2 K^2}\sum_{i=1}^{N-1}\sum_{i'=1}^{N-1}\boldsymbol{p}_i\,\mathrm{tr}(\boldsymbol{\Pi}_1^{2N-i'-i}) - \frac{\eta}{\epsilon N^3 K^2}\sum_{i=1}^{N}\boldsymbol{p}_i\cdot\sum_{i=1}^{N-1}\sum_{i'=1}^{N-1}\mathrm{tr}(\boldsymbol{\Pi}_1^{2N-i'-i})\right)\boldsymbol{p}_N^\top,$$

where the second equation is by Lemma D.1, the third equation is by the sampling method, the fourth equation is by $\boldsymbol{\Pi}_1 = \boldsymbol{\Pi}_1^\top$, and the fifth equation is by the fact that all the $\boldsymbol{x}_i$ is uniformly distributed in $\boldsymbol{E}$ for $i \in [N-1]$. Since $\left[\widetilde{\boldsymbol{X}}\boldsymbol{W}^{(2)}\widetilde{\boldsymbol{x}}_N\right]_N = \boldsymbol{p}_N^\top\boldsymbol{W}_{22}^{(2)}\boldsymbol{p}_N$ and $\left[\widetilde{\boldsymbol{X}}\boldsymbol{W}^{(2)}\widetilde{\boldsymbol{x}}_N\right]_j = \boldsymbol{x}_j^\top\boldsymbol{W}_{12}^{(2)}\boldsymbol{p}_N + \boldsymbol{p}_j^\top\boldsymbol{W}_{22}^{(2)}\boldsymbol{p}_N$ for $j \in \{1, 2, \ldots, N-1\}$, we can obtain that

$$\left[\widetilde{\boldsymbol{X}}\boldsymbol{W}^{(2)}\widetilde{\boldsymbol{x}}_N\right]_N$$

$$= \boldsymbol{p}_N^\top \boldsymbol{W}_{22}^{(2)} \boldsymbol{p}_N$$

$$= \mathbb{E}\left[\boldsymbol{p}_N^\top \frac{\eta}{\boldsymbol{e}_y^\top \boldsymbol{V} \boldsymbol{X} \mathcal{S} + \epsilon} \boldsymbol{B}^{(1)} \boldsymbol{p}_N\right]$$

$$= \mathbb{E}\left[\frac{\eta}{\boldsymbol{e}_y^\top \boldsymbol{V} \boldsymbol{X} \mathcal{S} + \epsilon}\left(-\mathcal{S}_N^{(1)} \boldsymbol{p}_N^\top \boldsymbol{p}_N \cdot \sum_{i=1}^{N-1} \mathcal{S}_i^{(1)} \boldsymbol{x}_i^\top (\boldsymbol{V}^{(1)})^\top \boldsymbol{e}_y\right) \boldsymbol{p}_N^\top \boldsymbol{p}_N\right]$$

$$< 0,$$

where the third equation is by $\boldsymbol{p}_i^\top \boldsymbol{p}_j = 0$ for $i \neq j$. And for $j \in \{1, 2, \ldots, N-2\}$, we can get

$$\left[\widetilde{\boldsymbol{X}} \boldsymbol{W}^{(2)} \widetilde{\boldsymbol{x}}_N\right]_{N-1} - \left[\widetilde{\boldsymbol{X}} \boldsymbol{W}^{(2)} \widetilde{\boldsymbol{x}}_N\right]_j$$

$$= \boldsymbol{x}_{N-1}^\top \boldsymbol{W}_{12}^{(2)} \boldsymbol{p}_N + \boldsymbol{p}_{N-1}^\top \boldsymbol{W}_{22}^{(2)} \boldsymbol{p}_N - \boldsymbol{x}_j^\top \boldsymbol{W}_{12}^{(2)} \boldsymbol{p}_N - \boldsymbol{p}_j^\top \boldsymbol{W}_{22}^{(2)} \boldsymbol{p}_N$$

$$\overset{(i)}{=} \boldsymbol{p}_{N-1}^\top \boldsymbol{W}_{22}^{(2)} \boldsymbol{p}_N - \boldsymbol{p}_j^\top \boldsymbol{W}_{22}^{(2)} \boldsymbol{p}_N$$

$$\overset{(ii)}{=} \mathbb{E}\left[\frac{\eta}{\boldsymbol{e}_y^\top \boldsymbol{V}^{(1)} \boldsymbol{X} \mathcal{S}^{(1)} + \epsilon} \cdot \frac{1}{N}\left(\boldsymbol{p}_{N-1}^\top \boldsymbol{p}_{N-1} \boldsymbol{x}_{N-1}^\top \boldsymbol{V}^{(1)} \boldsymbol{e}_y - \boldsymbol{p}_j^\top \boldsymbol{p}_j \boldsymbol{x}_j^\top \boldsymbol{V}^{(1)} \boldsymbol{e}_y\right) \boldsymbol{p}_N^\top \boldsymbol{p}_N\right]$$

$$\overset{(iii)}{\geq} \frac{\eta}{\max_{i,j}[\boldsymbol{V}^{(1)}]_{i,j} + \epsilon} \cdot \frac{(\boldsymbol{p}_N^\top \boldsymbol{p}_N)^2}{N} \mathbb{E}\left[\boldsymbol{x}_{N-1}^\top \boldsymbol{V}^{(1)} \boldsymbol{e}_y - \boldsymbol{x}_j^\top \boldsymbol{V}^{(1)} \boldsymbol{e}_y\right]$$

$$\overset{(iv)}{=} \frac{\eta}{\max_{i,j}[\boldsymbol{V}^{(1)}]_{i,j} + \epsilon} \cdot \frac{\eta(\boldsymbol{p}_N^\top \boldsymbol{p}_N)^2}{\epsilon N^2 K} \mathbb{E}\left[\boldsymbol{x}_{N-1}^\top \sum_{i=1}^{N-1} \boldsymbol{\Pi}_1^{N-i+1} \boldsymbol{x}_{N-1} - \boldsymbol{x}_j^\top \sum_{i=1}^{N-1} \boldsymbol{\Pi}_1^{2N-i-j} \boldsymbol{x}_j\right]$$

$$\overset{(v)}{=} \frac{\eta}{\max_{i,j}[\boldsymbol{V}^{(1)}]_{i,j} + \epsilon} \cdot \frac{\eta(\boldsymbol{p}_N^\top \boldsymbol{p}_N)^2}{\epsilon N^2 K^2}\left[\sum_{i=1}^{N-1} \operatorname{tr}(\boldsymbol{\Pi}_1^{N+1-i}) - \sum_{i=1}^{N-1} \operatorname{tr}(\boldsymbol{\Pi}_1^{2N-i-j})\right]$$

$$\overset{(vi)}{\geq} \frac{\eta}{\max_{i,j}[\boldsymbol{V}^{(1)}]_{i,j} + \epsilon} \cdot \frac{\eta(\boldsymbol{p}_N^\top \boldsymbol{p}_N)^2}{\epsilon N^2 K^2}\left[\operatorname{tr}(\boldsymbol{\Pi}_1^2) - \operatorname{tr}(\boldsymbol{\Pi}_1^{N+1})\right]$$

$$\overset{(vii)}{\geq} \frac{\eta}{\frac{\eta}{\epsilon N K} \cdot (N-1) + \epsilon} \cdot \frac{\eta(\boldsymbol{p}_N^\top \boldsymbol{p}_N)^2}{\epsilon N^2 K^2}\left(\frac{K}{2} - 2 - \frac{K}{\sqrt{N+2}}\right)$$

$$\geq \Omega\left(\frac{\eta M^2}{N^2}\right)$$

$$\geq \Omega(N),$$

where $(i)$ is by $\boldsymbol{W}_{12}^{(2)} \propto \mathbf{1}_K \boldsymbol{p}_N^\top$, $(ii)$ is by $\boldsymbol{p}_i^\top \boldsymbol{p}_{i'} = 0$ for $i \neq i'$, $(iii)$ is by Lemma D.3 and the fact that $\boldsymbol{e}_y^\top \boldsymbol{V}^{(1)} \boldsymbol{X} \mathcal{S}^{(1)} \leq \max_{i,j}[\boldsymbol{V}^{(1)}]_{i,j}$, $(iv)$ is by Lemma D.1 and the sampling method, $(v)$ is by the fact that all the $\boldsymbol{x}_i$ is uniformly distributed in $\boldsymbol{E}$ for $i \in [N-1]$, $(vi)$ is by Lemma F.3, and $(vii)$ is by Lemma F.3. Therefore, we have $\mathcal{S}_{N-1}^{(2)}/\mathcal{S}_j^{(2)} = \exp\left(\left[\widetilde{\boldsymbol{X}} \boldsymbol{W}^{(2)} \widetilde{\boldsymbol{x}}_N\right]_{N-1} - \left[\widetilde{\boldsymbol{X}} \boldsymbol{W}^{(2)} \widetilde{\boldsymbol{x}}_N\right]_j\right) \geq \exp(\Omega(N))$ for $j \neq N-1$. Further,

$$\mathcal{S}_{N-1}^{(2)} = 1 - \sum_{j \neq N-1} \mathcal{S}_j^{(2)} \geq 1 - (N-1)\exp(-\Omega(N))\mathcal{S}_{N-1}^{(2)},$$

which implies that

$$\mathcal{S}_{N-1}^{(2)} \geq \frac{1}{1 + (N-1)\exp(-\Omega(N))} = 1 - \frac{N-1}{\exp(\Omega(N)) + N - 1} = 1 - \exp(-\Omega(N)).$$

Then, we have $\mathcal{S}_j^{(2)} \leq 1 - \mathcal{S}_{N-1}^{(2)} \leq \exp(-\Omega(N))$ for $j \neq N-1$. $\qquad\square$

Then, we can get the bounds of $\boldsymbol{V}^{(t)}$.

**Lemma D.7.** If $\boldsymbol{\Pi} = \boldsymbol{\Pi}_1$, then it holds for $t \geq 3$ that

$$\min_{i,j}[\boldsymbol{V}^{(t)}]_{i,j} \geq \frac{\eta}{2\epsilon K^2} \text{ and } \|\boldsymbol{V}^{(t)}\|_{\max} \leq \frac{\eta}{\epsilon K} + (t-2) \cdot 2\epsilon K^2.$$

*Proof of Lemma D.7.* First, we have

$$\min_{i,j}[\boldsymbol{V}^{(t)}]_{i,j} \geq \min_{i,j}[\boldsymbol{V}^{(1)}]_{i,j}$$

$$\geq \min_{i,j}\left[\frac{\eta}{\epsilon NK}\sum_{i'=1}^{N-1}\boldsymbol{\Pi}_1^{N-i'}\right]_{i,j}$$

$$\geq \frac{\eta}{\epsilon NK}\sum_{i'=1}^{N-1}\left(\frac{1}{K}-\frac{1}{\sqrt{i'+1}}\right)$$

$$\geq \frac{\eta}{\epsilon NK}\frac{N-1}{K}-\frac{\eta}{\epsilon NK}\sum_{i'=2}^{N}2(\sqrt{i'+1}-\sqrt{i'})$$

$$\geq \frac{\eta}{2\epsilon K^2},$$

where the third inequality is by Lemma F.4, and the last inequality is by $N > 4K$. Then, we can get that

$$\|\boldsymbol{V}^{(t)}\|_{\max} \leq \|\boldsymbol{V}^{(t-1)}\|_{\max} + \left\|\mathbb{E}\left[\frac{\eta\boldsymbol{e}_y\sum_{i=1}^{N-1}\mathcal{S}_i^{(t-1)}\boldsymbol{x}_i^\top}{\boldsymbol{e}_y^\top\boldsymbol{V}^{(t-1)}\sum_{i=1}^{N-1}\mathcal{S}_i^{(t-1)}\boldsymbol{x}_i+\epsilon}\right]\right\|_{\max}$$

$$\leq \|\boldsymbol{V}^{(t-1)}\|_{\max} + \mathbb{E}\left[\frac{\eta\left\|\boldsymbol{e}_y\sum_{i=1}^{N-1}\mathcal{S}_i^{(t-1)}\boldsymbol{x}_i^\top\right\|_{\max}}{\min\left[\boldsymbol{e}_y^\top\boldsymbol{V}^{(t-1)}\sum_{i=1}^{N-1}\mathcal{S}_i^{(t-1)}\boldsymbol{x}_i\right]}\right]$$

$$\leq \|\boldsymbol{V}^{(t-1)}\|_{\max} + \frac{\eta}{\min_{i,j}[\boldsymbol{V}^{(t-1)}]_{i,j}}$$

$$\leq \|\boldsymbol{V}^{(t-1)}\|_{\max} + 2\epsilon K^2$$

$$\leq \|\boldsymbol{V}^{(2)}\|_{\max} + (t-3)\cdot 2\epsilon K^2$$

$$\leq \frac{\eta}{\epsilon K} + (t-2)\cdot 2\epsilon K^2,$$

where the third inequality is by $\boldsymbol{e}_y^\top\boldsymbol{V}\boldsymbol{x}_i \geq \min_{i,j}[\boldsymbol{V}]_{i,j}$, and the last inequality is by Lemma D.5. $\qquad\square$

Next, we can analyze the training dynamics over multiple iterations.

**Lemma D.8.** Assume that $\boldsymbol{\Pi} = \boldsymbol{\Pi}_1$ and $K$ is an even integer. For $2 \leq t \leq T^*$, it holds that $\mathcal{S}_{N-1}^{(t)} \geq 1 - \exp(-\Omega(N))$ and $\boldsymbol{V}^{(t)} = \beta^{(t)}\boldsymbol{\Pi}_1 + \widetilde{\boldsymbol{V}}^{(t)}$ where $\left\|\widetilde{\boldsymbol{V}}^{(t)}\right\|_{\max} \leq \gamma^{(t)}$. Here, $\beta^{(t)} \geq \sqrt{\eta t} - \frac{2\eta}{\epsilon K}$ and $\gamma^{(t)} \leq \frac{2\eta}{\epsilon K} + 2(t-1)\epsilon K^2 N \exp(-\Omega(N))$.

*Proof of Lemma D.8.* We use induction to prove the results that

$$\beta^{(t)} \geq \sqrt{\eta t} - \frac{2\eta}{\epsilon K},$$

$$\gamma^{(t)} \leq \frac{2\eta}{\epsilon K} + 2(t-1)\epsilon K^2 N \exp(-\Omega(N)),$$

$$\left[\widetilde{\boldsymbol{X}}\boldsymbol{W}^{(t)}\widetilde{\boldsymbol{x}}_N\right]_{N-1} - \left[\widetilde{\boldsymbol{X}}\boldsymbol{W}^{(t)}\widetilde{\boldsymbol{x}}_N\right]_j \geq \Omega(N),$$

$$\mathcal{S}_{N-1}^{(t)} \geq 1 - \exp(-\Omega(N)).$$

By Lemma D.5 and Lemma D.6, it can be easily checked that the results hold for $t = 2$. Suppose that the results hold for $\boldsymbol{V}^{(t)}$ and $\mathcal{S}^{(t)}$. We aim to prove that the results hold for $t + 1$.

For $\boldsymbol{V}^{(t+1)}$, we can get

$$\boldsymbol{V}^{(t+1)} = \boldsymbol{V}^{(t)} - \eta\mathbb{E}[\nabla_{\boldsymbol{V}}\ell(\theta^{(t)})]$$

$$= \boldsymbol{V}^{(t)} + \eta \mathbb{E}\left[\frac{\boldsymbol{e}_y \sum_{i=1}^{N-1} \mathcal{S}_i^{(t)} \boldsymbol{x}_i^\top}{\boldsymbol{e}_y^\top \boldsymbol{V}^{(t)} \sum_{i=1}^{N-1} \mathcal{S}_i^{(t)} \boldsymbol{x}_i + \epsilon}\right]$$

$$= \boldsymbol{V}^{(t)} + \mathbb{E}\left[\frac{\eta \mathcal{S}_{N-1}^{(t)} \boldsymbol{e}_y \boldsymbol{x}_{N-1}^\top}{\boldsymbol{e}_y^\top \boldsymbol{V}^{(t)} \sum_{i=1}^{N-1} \mathcal{S}_i^{(t)} \boldsymbol{x}_i + \epsilon}\right] + \mathbb{E}\left[\frac{\eta \boldsymbol{e}_y \sum_{i=1}^{N-2} \mathcal{S}_i^{(t)} \boldsymbol{x}_i^\top}{\boldsymbol{e}_y^\top \boldsymbol{V}^{(t)} \sum_{i=1}^{N-1} \mathcal{S}_i^{(t)} \boldsymbol{x}_i + \epsilon}\right].$$

Then, we have

$$\left[\mathbb{E}\left[\frac{\eta \mathcal{S}_{N-1}^{(t)} \boldsymbol{e}_y \boldsymbol{x}_{N-1}^\top}{\boldsymbol{e}_y^\top \boldsymbol{V}^{(t)} \sum_{i=1}^{N-1} \mathcal{S}_i^{(t)} \boldsymbol{x}_i + \epsilon}\right]\right]_{1,2} \geq \frac{\eta \mathcal{S}_{N-1}^{(t)}}{\|\boldsymbol{V}^{(t)}\|_{\max} + \epsilon} \cdot \left[\mathbb{E}\left[\boldsymbol{e}_y \boldsymbol{x}_{N-1}^\top\right]\right]_{1,2}$$

$$\geq \frac{\eta[1 - \exp(-\Omega(N))]}{\beta^{(t)} + \gamma^{(t)} + \epsilon} \cdot \frac{1}{2}$$

$$\geq \frac{\eta}{4\left[\beta^{(t)} + \frac{\eta}{\epsilon K} + 2\epsilon K^2 + 2t\epsilon K^2 N \exp(-\Omega(N)) + \epsilon\right]}$$

$$\geq \frac{\eta}{4\left(\beta^{(t)} + \frac{2\eta}{\epsilon K}\right)}, \tag{D.4}$$

where the first inequality is by $\boldsymbol{e}_y^\top \boldsymbol{V}^{(t)} \boldsymbol{x}_i \leq \|\boldsymbol{V}^{(t)}\|_{\max}$, the second inequality is by induction, and the third inequality is by the assumption of $\epsilon$. And, we have

$$\left\|\mathbb{E}\left[\frac{\eta \boldsymbol{e}_y \sum_{i=1}^{N-2} \mathcal{S}_i^{(t)} \boldsymbol{x}_i^\top}{\boldsymbol{e}_y^\top \boldsymbol{V}^{(t)} \sum_{i=1}^{N-1} \mathcal{S}_i^{(t)} \boldsymbol{x}_i + \epsilon}\right]\right\|_{\max} \leq \frac{\eta \exp(-\Omega(N))}{\min_{i,j}[\boldsymbol{V}^{(t)}]_{i,j}} \cdot \left\|\mathbb{E}\left[\sum_{i=1}^{N-2} \boldsymbol{e}_y \boldsymbol{x}_i^\top\right]\right\|_{\max}$$

$$\leq 2\epsilon K^2 \exp(-\Omega(N)) \cdot N, \tag{D.5}$$

where the first inequality is by induction and $\boldsymbol{e}_y^\top \boldsymbol{V}^{(t)} \boldsymbol{x}_i \geq \min_{i,j}[\boldsymbol{V}^{(t)}]_{i,j}$, and the second inequality is by Lemma D.7. Thus, we can get that

$$\beta^{(t+1)} \geq \beta^{(t)} + 2\left[\mathbb{E}\left[\frac{\eta \mathcal{S}_{N-1}^{(t)} \boldsymbol{e}_y \boldsymbol{x}_{N-1}^\top}{\boldsymbol{e}_y^\top \boldsymbol{V}^{(t)} \sum_{i=1}^{N-1} \mathcal{S}_i^{(t)} \boldsymbol{x}_i + \epsilon}\right]\right]_{1,2}$$

$$\geq \beta^{(t)} + \frac{\eta}{2\beta^{(t)} + \frac{4\eta}{\epsilon K}}$$

$$\geq \sqrt{\eta t} - \frac{2\eta}{\epsilon K} + \frac{\eta}{2\sqrt{\eta t}}$$

$$\geq \sqrt{\eta t} - \frac{2\eta}{\epsilon K} + \frac{\sqrt{\eta}}{\sqrt{t+1} + \sqrt{t}}$$

$$= \sqrt{\eta(t+1)} - \frac{2\eta}{\epsilon K}$$

where the second inequality is by (D.4), and the third inequality is by induction and the fact that $x + \frac{\eta}{2x + \frac{4\eta}{\epsilon K}}$ is monotonically increasing for $x \geq \frac{\sqrt{\eta}}{\sqrt{2}} - \frac{2\eta}{\epsilon K}$. And,

$$\gamma^{(t+1)} \leq \gamma^{(t)} + \left\|\mathbb{E}\left[\frac{\eta \boldsymbol{e}_y \sum_{i=1}^{N-2} \mathcal{S}_i^{(t)} \boldsymbol{x}_i^\top}{\boldsymbol{e}_y^\top \boldsymbol{V}^{(t)} \sum_{i=1}^{N-1} \mathcal{S}_i^{(t)} \boldsymbol{x}_i + \epsilon}\right]\right\|_{\max}$$

$$\leq \gamma^{(t)} + 2\epsilon K^2 N \exp(-\Omega(N))$$

$$\leq \frac{2\eta}{\epsilon K} + 2t\epsilon K^2 N \exp(-\Omega(N)),$$

where the second inequality is by (D.5), and the third inequality is by induction.

Next, we consider $\mathcal{S}^{(t+1)}$. Recall that

$$\boldsymbol{W}_{12}^{(t+1)} = \boldsymbol{W}_{12}^{(t)} + \eta \mathbb{E}\left[\frac{\boldsymbol{A}^{(t)}}{\boldsymbol{e}_y^\top \boldsymbol{V} \boldsymbol{X} \mathcal{S} + \epsilon}\right] \text{ and } \boldsymbol{W}_{22}^{(t+1)} = \boldsymbol{W}_{22}^{(t)} + \eta \mathbb{E}\left[\frac{\boldsymbol{B}^{(t)}}{\boldsymbol{e}_y^\top \boldsymbol{V} \boldsymbol{X} \mathcal{S} + \epsilon}\right],$$

where

$$\boldsymbol{A}^{(t)} = \left( \sum_{i=1}^{N-1} \mathcal{S}_i^{(t)} \boldsymbol{x}_i \boldsymbol{x}_i^\top (\boldsymbol{V}^{(t)})^\top \boldsymbol{e}_y - \sum_{i_1=1}^{N-1} \sum_{i_2=1}^{N-1} \mathcal{S}_{i_1}^{(t)} \mathcal{S}_{i_2}^{(t)} \boldsymbol{x}_{i_1} \boldsymbol{x}_{i_2}^\top (\boldsymbol{V}^{(t)})^\top \boldsymbol{e}_y \right) \boldsymbol{p}_N^\top,$$

$$\boldsymbol{B}^{(t)} = \left( \sum_{i=1}^{N-1} \mathcal{S}_i^{(t)} \boldsymbol{p}_i \boldsymbol{x}_i^\top (\boldsymbol{V}^{(t)})^\top \boldsymbol{e}_y - \sum_{i=1}^{N} \mathcal{S}_i^{(t)} \boldsymbol{p}_i \cdot \sum_{i=1}^{N-1} \mathcal{S}_i^{(t)} \boldsymbol{x}_i^\top (\boldsymbol{V}^{(t)})^\top \boldsymbol{e}_y \right) \boldsymbol{p}_N^\top.$$

We also have $\left[ \widetilde{\boldsymbol{X}} \boldsymbol{W}^{(t)} \widetilde{\boldsymbol{x}}_N \right]_N = \boldsymbol{p}_N^\top \boldsymbol{W}_{22}^{(t)} \boldsymbol{p}_N$ and $\left[ \widetilde{\boldsymbol{X}} \boldsymbol{W}^{(t)} \widetilde{\boldsymbol{x}}_N \right]_j = \boldsymbol{x}_j^\top \boldsymbol{W}_{12}^{(t)} \boldsymbol{p}_N + \boldsymbol{p}_j^\top \boldsymbol{W}_{22}^{(t)} \boldsymbol{p}_N$ for $j \in \{1, 2, \ldots, N-1\}$. Then, for $j = N$, we have

$$\boldsymbol{p}_N^\top \boldsymbol{W}_{22}^{(t+1)} \boldsymbol{p}_N = \boldsymbol{p}_N^\top \boldsymbol{W}_{22}^{(t)} \boldsymbol{p}_N + \eta \mathbb{E} \left[ \frac{\boldsymbol{p}_N^\top \boldsymbol{B}^{(t)} \boldsymbol{p}_N}{\boldsymbol{e}_y^\top \boldsymbol{V}^{(t)} \sum_{i=1}^{N-1} \mathcal{S}_i^{(t)} \boldsymbol{x}_i + \epsilon} \right]$$

$$= \boldsymbol{p}_N^\top \boldsymbol{W}_{22}^{(t)} \boldsymbol{p}_N - \eta M \mathcal{S}_N^{(t)} \mathbb{E} \left[ \frac{\sum_{i=1}^{N-1} \mathcal{S}_i^{(t)} \boldsymbol{x}_i^\top \boldsymbol{V}^{(t)} \boldsymbol{e}_y}{\boldsymbol{e}_y^\top \boldsymbol{V}^{(t)} \sum_{i=1}^{N-1} \mathcal{S}_i^{(t)} \boldsymbol{x}_i + \epsilon} \right]$$

$$\leq \boldsymbol{p}_N^\top \boldsymbol{W}_{22}^{(t)} \boldsymbol{p}_N.$$

For $j \in \{1, 2, \ldots, N-2\}$, we have

$$\boldsymbol{x}_j^\top \boldsymbol{W}_{12}^{(t+1)} \boldsymbol{p}_N = \boldsymbol{x}_j^\top \boldsymbol{W}_{12}^{(t)} \boldsymbol{p}_N + \eta \mathbb{E} \left[ \frac{\boldsymbol{x}_j^\top \boldsymbol{A}^{(t)} \boldsymbol{p}_N}{\boldsymbol{e}_y^\top \boldsymbol{V}^{(t)} \sum_{i=1}^{N-1} \mathcal{S}_i^{(t)} \boldsymbol{x}_i + \epsilon} \right]$$

$$= \boldsymbol{x}_j^\top \boldsymbol{W}_{12}^{(t)} \boldsymbol{p}_N + \eta \sqrt{M} \mathcal{S}_j^{(t)} \mathbb{E} \left[ \frac{\boldsymbol{x}_j^\top \boldsymbol{V}^{(t)} \boldsymbol{e}_y - \sum_{i_2=1}^{N-1} \mathcal{S}_{i_2}^{(t)} \boldsymbol{x}_{i_2}^\top \boldsymbol{V}^{(t)} \boldsymbol{e}_y}{\boldsymbol{e}_y^\top \boldsymbol{V}^{(t)} \sum_{i=1}^{N-1} \mathcal{S}_i^{(t)} \boldsymbol{x}_i + \epsilon} \right]$$

$$\leq \boldsymbol{x}_j^\top \boldsymbol{W}_{12}^{(t)} \boldsymbol{p}_N + \eta \sqrt{M} \mathcal{S}_j^{(t)} \frac{\|\boldsymbol{V}^{(t)}\|_{\max}}{\min_{i,j}[\boldsymbol{V}^{(t)}]_{i,j}}$$

$$\leq \boldsymbol{x}_j^\top \boldsymbol{W}_{12}^{(t)} \boldsymbol{p}_N + \eta \sqrt{M} \exp(-\Omega(N)) \left( 2K + \frac{4(t-2)\epsilon^2 K^4}{\eta} \right),$$

where the first inequality is by $\boldsymbol{e}_y^\top \boldsymbol{V}^{(t)} \boldsymbol{x}_{N-1} = \|\boldsymbol{V}^{(t)}\|_{\max}$, and the second inequality is by induction and Lemma D.7. And,

$$\boldsymbol{p}_j^\top \boldsymbol{W}_{22}^{(t+1)} \boldsymbol{p}_N = \boldsymbol{p}_j^\top \boldsymbol{W}_{22}^{(t)} \boldsymbol{p}_N + \eta \mathbb{E} \left[ \frac{\boldsymbol{p}_j^\top \boldsymbol{B}^{(t)} \boldsymbol{p}_N}{\boldsymbol{e}_y^\top \boldsymbol{V}^{(t)} \sum_{i=1}^{N-1} \mathcal{S}_i^{(t)} \boldsymbol{x}_i + \epsilon} \right]$$

$$= \boldsymbol{p}_j^\top \boldsymbol{W}_{22}^{(t)} \boldsymbol{p}_N + \eta M \mathcal{S}_j^{(t)} \mathbb{E} \left[ \frac{\boldsymbol{x}_j^\top \boldsymbol{V}^{(t)} \boldsymbol{e}_y - \sum_{i=1}^{N-1} \mathcal{S}_i^{(t)} \boldsymbol{x}_i^\top \boldsymbol{V}^{(t)} \boldsymbol{e}_y}{\boldsymbol{e}_y^\top \boldsymbol{V}^{(t)} \sum_{i=1}^{N-1} \mathcal{S}_i^{(t)} \boldsymbol{x}_i + \epsilon} \right]$$

$$\leq \boldsymbol{p}_j^\top \boldsymbol{W}_{22}^{(t)} \boldsymbol{p}_N + \eta M \mathcal{S}_j^{(t)} \frac{\|\boldsymbol{V}^{(t)}\|_{\max}}{\min_{i,j}[\boldsymbol{V}^{(t)}]_{i,j}}$$

$$\leq \boldsymbol{p}_j^\top \boldsymbol{W}_{22}^{(t)} \boldsymbol{p}_N + \eta M \exp(-\Omega(N)) \left( 2K + \frac{4(t-2)\epsilon^2 K^4}{\eta} \right),$$

where the first inequality is by $\boldsymbol{e}_y^\top \boldsymbol{V}^{(t)} \boldsymbol{x}_{N-1} = \|\boldsymbol{V}^{(t)}\|_{\max}$, and the second inequality is by induction and Lemma D.7. For $j = N - 1$, we have

$$\boldsymbol{x}_{N-1}^\top \boldsymbol{W}_{12}^{(t+1)} \boldsymbol{p}_N = \boldsymbol{x}_{N-1}^\top \boldsymbol{W}_{12}^{(t)} \boldsymbol{p}_N + \eta \mathbb{E} \left[ \frac{\boldsymbol{x}_{N-1}^\top \boldsymbol{A}^{(t)} \boldsymbol{p}_N}{\boldsymbol{e}_y^\top \boldsymbol{V}^{(t)} \sum_{i=1}^{N-1} \mathcal{S}_i^{(t)} \boldsymbol{x}_i + \epsilon} \right]$$

$$= \boldsymbol{x}_{N-1}^\top \boldsymbol{W}_{12}^{(t)} \boldsymbol{p}_N + \eta \sqrt{M} \mathcal{S}_{N-1}^{(t)} \mathbb{E} \left[ \frac{\boldsymbol{x}_{N-1}^\top \boldsymbol{V}^{(t)} \boldsymbol{e}_y - \sum_{i_2=1}^{N-1} \mathcal{S}_{i_2}^{(t)} \boldsymbol{x}_{i_2}^\top \boldsymbol{V}^{(t)} \boldsymbol{e}_y}{\boldsymbol{e}_y^\top \boldsymbol{V}^{(t)} \sum_{i=1}^{N-1} \mathcal{S}_i^{(t)} \boldsymbol{x}_i + \epsilon} \right]$$

$$\geq \boldsymbol{x}_{N-1}^\top \boldsymbol{W}_{12}^{(t)} \boldsymbol{p}_N + \eta \sqrt{M} \mathbb{E} \left[ \frac{-\sum_{i_2=1}^{N-2} \mathcal{S}_{i_2}^{(t)} \boldsymbol{x}_{i_2}^\top \boldsymbol{V}^{(t)} \boldsymbol{e}_y}{\boldsymbol{e}_y^\top \boldsymbol{V}^{(t)} \sum_{i=1}^{N-1} \mathcal{S}_i^{(t)} \boldsymbol{x}_i + \epsilon} \right]$$

$$\geq \boldsymbol{x}_{N-1}^\top \boldsymbol{W}_{12}^{(t)} \boldsymbol{p}_N - \eta \sqrt{M} \frac{\sum_{i_2=1}^{N-2} \mathcal{S}_{i_2}^{(t)} \|\boldsymbol{V}^{(t)}\|_{\max}}{\min_{i,j} [\boldsymbol{V}^{(t)}]_{i,j}}$$

$$\geq \boldsymbol{x}_{N-1}^\top \boldsymbol{W}_{12}^{(t)} \boldsymbol{p}_N - \eta \sqrt{M} N \exp(-\Omega(N)) \left( 2K + \frac{4(t-2)\epsilon^2 K^4}{\eta} \right),$$

where the second inequality is by $\boldsymbol{e}_y^\top \boldsymbol{V}^{(t)} \boldsymbol{x}_{N-1} = \|\boldsymbol{V}^{(t)}\|_{\max}$, and the third inequality is by induction and Lemma D.7. And,

$$\boldsymbol{p}_{N-1}^\top \boldsymbol{W}_{22}^{(t+1)} \boldsymbol{p}_N = \boldsymbol{p}_{N-1}^\top \boldsymbol{W}_{22}^{(t)} \boldsymbol{p}_N + \eta \mathbb{E} \left[ \frac{\boldsymbol{p}_{N-1}^\top \boldsymbol{B}^{(t)} \boldsymbol{p}_N}{\boldsymbol{e}_y^\top \boldsymbol{V}^{(t)} \sum_{i=1}^{N-1} \mathcal{S}_i^{(t)} \boldsymbol{x}_i + \epsilon} \right]$$

$$= \boldsymbol{p}_{N-1}^\top \boldsymbol{W}_{22}^{(t)} \boldsymbol{p}_N + \eta M \mathcal{S}_{N-1}^{(t)} \mathbb{E} \left[ \frac{\boldsymbol{x}_{N-1}^\top \boldsymbol{V}^{(t)} \boldsymbol{e}_y - \sum_{i=1}^{N-1} \mathcal{S}_i^{(t)} \boldsymbol{x}_i^\top \boldsymbol{V}^{(t)} \boldsymbol{e}_y}{\boldsymbol{e}_y^\top \boldsymbol{V}^{(t)} \sum_{i=1}^{N-1} \mathcal{S}_i^{(t)} \boldsymbol{x}_i + \epsilon} \right]$$

$$\geq \boldsymbol{p}_{N-1}^\top \boldsymbol{W}_{22}^{(t)} \boldsymbol{p}_N + \eta M \mathbb{E} \left[ \frac{-\sum_{i=1}^{N-2} \mathcal{S}_i^{(t)} \boldsymbol{x}_i^\top \boldsymbol{V}^{(t)} \boldsymbol{e}_y}{\boldsymbol{e}_y^\top \boldsymbol{V}^{(t)} \sum_{i=1}^{N-1} \mathcal{S}_i^{(t)} \boldsymbol{x}_i + \epsilon} \right]$$

$$\geq \boldsymbol{p}_{N-1}^\top \boldsymbol{W}_{22}^{(t)} \boldsymbol{p}_N - \eta M \frac{\sum_{i_2=1}^{N-2} \mathcal{S}_{i_2}^{(t)} \|\boldsymbol{V}^{(t)}\|_{\max}}{\min_{i,j} [\boldsymbol{V}^{(t)}]_{i,j}}$$

$$\geq \boldsymbol{p}_{N-1}^\top \boldsymbol{W}_{22}^{(t)} \boldsymbol{p}_N - \eta M N \exp(-\Omega(N)) \left( 2K + \frac{4(t-2)\epsilon^2 K^4}{\eta} \right),$$

where the second inequality is by $\boldsymbol{e}_y^\top \boldsymbol{V}^{(t)} \boldsymbol{x}_{N-1} = \|\boldsymbol{V}^{(t)}\|_{\max}$, and the third inequality is by induction and Lemma D.7. Therefore, we can get that for $j \neq N-1$,

$$\left[ \widetilde{\boldsymbol{X}} \boldsymbol{W}^{(t+1)} \widetilde{\boldsymbol{x}}_N \right]_{N-1} - \left[ \widetilde{\boldsymbol{X}} \boldsymbol{W}^{(t+1)} \widetilde{\boldsymbol{x}}_N \right]_j$$

$$= \boldsymbol{x}_{N-1}^\top \boldsymbol{W}_{12}^{(t+1)} \boldsymbol{p}_N + \boldsymbol{p}_{N-1}^\top \boldsymbol{W}_{22}^{(t+1)} \boldsymbol{p}_N - \boldsymbol{x}_j^\top \boldsymbol{W}_{12}^{(t+1)} \boldsymbol{p}_N - \boldsymbol{p}_j^\top \boldsymbol{W}_{22}^{(t+1)} \boldsymbol{p}_N$$

$$\geq \boldsymbol{x}_{N-1}^\top \boldsymbol{W}_{12}^{(t)} \boldsymbol{p}_N + \boldsymbol{p}_{N-1}^\top \boldsymbol{W}_{22}^{(t)} \boldsymbol{p}_N - \boldsymbol{x}_j^\top \boldsymbol{W}_{12}^{(t)} \boldsymbol{p}_N - \boldsymbol{p}_j^\top \boldsymbol{W}_{22}^{(t)} \boldsymbol{p}_N$$

$$\quad - 4\eta M N \exp(-\Omega(N)) \left( 2K + \frac{4(t-2)\epsilon^2 K^4}{\eta} \right)$$

$$= \left[ \widetilde{\boldsymbol{X}} \boldsymbol{W}^{(t)} \widetilde{\boldsymbol{x}}_N \right]_{N-1} - \left[ \widetilde{\boldsymbol{X}} \boldsymbol{W}^{(t)} \widetilde{\boldsymbol{x}}_N \right]_j - \exp(-\Omega(N))$$

$$\geq \Omega(N),$$

where the last inequality is by induction. Thus, $\mathcal{S}_{N-1}^{(t+1)}/\mathcal{S}_j^{(t+1)} \geq \exp(\Omega(N))$ for $j \neq N-1$, which implies that $\mathcal{S}_{N-1}^{(t+1)} \geq 1 - \exp(-\Omega(N))$. Therefore, we prove that the results hold for $t+1$, which completes the proof. $\qquad\square$

The next two lemmas show the convergence rates of $\boldsymbol{V}^{(T)}/\|\boldsymbol{V}^{(T)}\|_F$ and $f_{\theta^T}(\boldsymbol{X})/\|f_{\theta^T}(\boldsymbol{X})\|_2$.

**Lemma D.9.** Assume that $\boldsymbol{\Pi} = \boldsymbol{\Pi}_1$ and $K$ is an even integer. For $\Omega(\eta \epsilon^{-2} K^{-2}) \leq T \leq T^*$, it holds that

$$\left\| \frac{\boldsymbol{V}^{(T)}}{\|\boldsymbol{V}^{(T)}\|_F} - \frac{\boldsymbol{\Pi}_1^\top}{\|\boldsymbol{\Pi}_1^\top\|_F} \right\|_F \leq \mathcal{O}\left( \frac{1}{\sqrt{T}} \right).$$

***Proof of Lemma D.9.*** By Lemma D.8, we can get that

$$\left\| \frac{\boldsymbol{V}^{(T)}}{\|\boldsymbol{V}^{(T)}\|_F} - \frac{\boldsymbol{\Pi}_1^\top}{\|\boldsymbol{\Pi}_1^\top\|_F} \right\|_F = \left\| \frac{\beta^{(T)} \boldsymbol{\Pi}_1 + \widetilde{\boldsymbol{V}}^{(T)}}{\|\boldsymbol{V}^{(T)}\|_F} - \frac{\boldsymbol{\Pi}_1}{\|\boldsymbol{\Pi}_1\|_F} \right\|_F$$

$$\leq \left\| \left( \frac{\beta^{(T)}}{\|\boldsymbol{V}^{(T)}\|_F} - \frac{1}{\|\boldsymbol{\Pi}_1\|_F} \right) \boldsymbol{\Pi}_1 \right\|_F + \left\| \frac{\widetilde{\boldsymbol{V}}^{(T)}}{\|\boldsymbol{V}^{(T)}\|_F} \right\|_F.$$

For the first part, we have

$$\left\|\left(\frac{\beta^{(T)}}{\|\boldsymbol{V}^{(T)}\|_F} - \frac{1}{\|\boldsymbol{\Pi}_1\|_F}\right)\boldsymbol{\Pi}_1\right\|_F = \left|\frac{\beta^{(T)}\|\boldsymbol{\Pi}_1\|_F}{\|\beta^{(T)}\boldsymbol{\Pi}_1 + \widetilde{\boldsymbol{V}}^{(T)}\|_F} - 1\right|$$

$$\leq 1 - \frac{\beta^{(T)}\|\boldsymbol{\Pi}_1\|_F}{\beta^{(T)}\|\boldsymbol{\Pi}_1\|_F + \|\widetilde{\boldsymbol{V}}^{(T)}\|_F}$$

$$\overset{(i)}{=} \frac{\|\widetilde{\boldsymbol{V}}^{(T)}\|_F}{\frac{\sqrt{2K}}{2}\beta^{(T)} + \|\widetilde{\boldsymbol{V}}^{(T)}\|_F}$$

$$\leq \frac{K\gamma^{(T)}}{\frac{\sqrt{2K}}{2}\beta^{(T)} + K\gamma^{(T)}}$$

$$\overset{(ii)}{\leq} \frac{\frac{2\eta}{\epsilon} + 2T\epsilon K^3 N\exp(-\Omega(N))}{\frac{\sqrt{2K}}{2}\left(\sqrt{\eta T} - \frac{2\eta}{\epsilon K}\right) + \frac{2\eta}{\epsilon} + 2T\epsilon K^3 N\exp(-\Omega(N))}$$

$$\leq \mathcal{O}\left(\frac{1}{\sqrt{T}}\right),$$

where $(i)$ is by $\|\boldsymbol{\Pi}_1\|_F = \sqrt{2K}/2$, and $(ii)$ is by Lemma D.8. For the second part, we have

$$\left\|\frac{\widetilde{\boldsymbol{V}}^{(T)}}{\|\boldsymbol{V}^{(T)}\|_F}\right\|_F = \frac{\|\widetilde{\boldsymbol{V}}^{(T)}\|_F}{\|\beta^{(T)}\boldsymbol{\Pi}_1 + \widetilde{\boldsymbol{V}}^{(T)}\|_F}$$

$$\leq \frac{\|\widetilde{\boldsymbol{V}}^{(T)}\|_F}{\|\beta^{(T)}\boldsymbol{\Pi}_1\|_F + \|\widetilde{\boldsymbol{V}}^{(T)}\|_F}$$

$$\overset{(i)}{\leq} \frac{K\gamma^{(T)}}{\frac{\sqrt{2K}}{2}\beta^{(T)} + K\gamma^{(T)}}$$

$$\overset{(ii)}{\leq} \frac{\frac{2\eta}{\epsilon} + 2T\epsilon K^3 N\exp(-\Omega(N))}{\frac{\sqrt{2K}}{2}\left(\sqrt{\eta T} - \frac{2\eta}{\epsilon K}\right) + \frac{2\eta}{\epsilon} + 2T\epsilon K^3 N\exp(-\Omega(N))}$$

$$\leq \mathcal{O}\left(\frac{1}{\sqrt{T}}\right),$$

where $(i)$ is by $\|\boldsymbol{\Pi}_1\|_F = \sqrt{2K}/2$, and $(ii)$ is by Lemma D.8. Therefore, we can obtain that

$$\left\|\frac{\boldsymbol{V}^{(T)}}{\|\boldsymbol{V}^{(T)}\|_F} - \frac{\boldsymbol{\Pi}_1^\top}{\|\boldsymbol{\Pi}_1^\top\|_F}\right\|_F \leq \mathcal{O}\left(\frac{1}{\sqrt{T}}\right).$$

$\square$

**Lemma D.10.** Assume that $\boldsymbol{\Pi} = \boldsymbol{\Pi}_1$ and $K$ is an even integer. For $\Omega(\eta\epsilon^{-2}K^{-2}) \leq T \leq T^*$, it holds that

$$\left\|\frac{f_{\theta^{(T)}}(\boldsymbol{X})}{\|f_{\theta^{(T)}}(\boldsymbol{X})\|_2} - \boldsymbol{\Pi}_1^\top \boldsymbol{x}_{N-1}\right\|_2 \leq \mathcal{O}\left(\frac{1}{\sqrt{T}}\right).$$

**Proof of Lemma D.10.** The output with $\theta = \theta^{(T)}$ is $f_{\theta^{(T)}}(\boldsymbol{X}) = \boldsymbol{V}^{(T)}\boldsymbol{X}\mathcal{S}(\widetilde{\boldsymbol{X}}^\top\boldsymbol{W}^{(T)}\widetilde{\boldsymbol{x}}_N) = \boldsymbol{V}^{(T)}\sum_{i=1}^{N-1}\mathcal{S}_i^{(T)}\boldsymbol{x}_i$. Then, we can get that

$$\left\|\frac{f_{\theta^{(T)}}(\boldsymbol{X})}{\|f_{\theta^{(T)}}(\boldsymbol{X})\|_2} - \boldsymbol{\Pi}_1^\top \boldsymbol{x}_{N-1}\right\|_2$$

$$= \left\|\frac{\left(\beta^{(T)}\boldsymbol{\Pi}_1 + \widetilde{\boldsymbol{V}}^{(T)}\right)\sum_{i=1}^{N-1}\mathcal{S}_i^{(T)}\boldsymbol{x}_i}{\left\|\boldsymbol{V}^{(T)}\sum_{i=1}^{N-1}\mathcal{S}_i^{(T)}\boldsymbol{x}_i\right\|_2} - \boldsymbol{\Pi}_1\boldsymbol{x}_{N-1}\right\|_2$$

$$\leq \left\| \left( \frac{\beta^{(T)} \mathcal{S}_{N-1}^{(T)}}{\left\| \mathbf{V}^{(T)} \sum_{i=1}^{N-1} \mathcal{S}_i^{(T)} \mathbf{x}_i \right\|_2} - 1 \right) \mathbf{\Pi}_1 \mathbf{x}_{N-1} \right\|_2$$

$$+ \left\| \frac{\beta^{(T)} \mathbf{\Pi}_1 \sum_{i=1}^{N-2} \mathcal{S}_i^{(T)} \mathbf{x}_i + \widetilde{\mathbf{V}}^{(T)} \sum_{i=1}^{N-1} \mathcal{S}_i^{(T)} \mathbf{x}_i}{\left\| \mathbf{V}^{(T)} \sum_{i=1}^{N-1} \mathcal{S}_i^{(T)} \mathbf{x}_i \right\|_2} \right\|_2.$$

For the first part, we have

$$\left\| \left( \frac{\beta^{(T)} \mathcal{S}_{N-1}^{(T)}}{\left\| \mathbf{V}^{(T)} \sum_{i=1}^{N-1} \mathcal{S}_i^{(T)} \mathbf{x}_i \right\|_2} - 1 \right) \mathbf{\Pi}_1 \mathbf{x}_{N-1} \right\|_2$$

$$\leq \left\| \left( 1 - \frac{\beta^{(T)} \mathcal{S}_{N-1}^{(T)}}{\left\| \beta^{(T)} \mathbf{\Pi}_1 \sum_{i=1}^{N-1} \mathcal{S}_i^{(T)} \mathbf{x}_i \right\|_2 + \left\| \widetilde{\mathbf{V}}^{(T)} \sum_{i=1}^{N-1} \mathcal{S}_i^{(T)} \mathbf{x}_i \right\|_2} \right) \mathbf{\Pi}_1 \mathbf{x}_{N-1} \right\|_2$$

$$\leq \left\| \left( 1 - \frac{\beta^{(T)} [1 - \exp(-\Omega(N))]}{\frac{\sqrt{2}}{2} \beta^{(T)} + \sqrt{K} \gamma^{(T)}} \right) \mathbf{\Pi}_1 \mathbf{x}_{N-1} \right\|_2$$

$$\leq \left( 1 - \frac{\left( \sqrt{\eta T} - \frac{2\eta}{\epsilon K} \right) [1 - \exp(-\Omega(N))]}{\left( \sqrt{\eta T} - \frac{2\eta}{\epsilon K} \right) + \sqrt{K} \left( \frac{2\eta}{\epsilon K} + 2T\epsilon K^2 N \exp(-\Omega(N)) \right)} \right) \cdot \frac{\sqrt{2}}{2}$$

$$\leq \mathcal{O} \left( \frac{\sqrt{K} \left( \frac{2\eta}{\epsilon K} + 2T\epsilon K^2 N \exp(-\Omega(N)) \right)}{\sqrt{\eta T} + 2T\epsilon K^2 N \exp(-\Omega(N))} \right)$$

$$\leq \mathcal{O} \left( \frac{1}{\sqrt{T}} \right),$$

where the first inequality is by Lemma D.8, the second inequality is by Lemma D.8 and $\|\mathbf{\Pi}_1 \mathbf{x}_i\|_2 = \sqrt{2}/2$, and the third inequality is by Lemma D.8. For the second part, we have

$$\left\| \frac{\beta^{(T)} \mathbf{\Pi}_1 \sum_{i=1}^{N-2} \mathcal{S}_i^{(T)} \mathbf{x}_i + \widetilde{\mathbf{V}}^{(T)} \sum_{i=1}^{N-1} \mathcal{S}_i^{(T)} \mathbf{x}_i}{\left\| \mathbf{V}^{(T)} \sum_{i=1}^{N-1} \mathcal{S}_i^{(T)} \mathbf{x}_i \right\|_2} \right\|_2$$

$$\leq \frac{\left\| \beta^{(T)} \mathbf{\Pi}_1 \sum_{i=1}^{N-2} \mathcal{S}_i^{(T)} \mathbf{x}_i \right\|_2 + \left\| \widetilde{\mathbf{V}}^{(T)} \sum_{i=1}^{N-1} \mathcal{S}_i^{(T)} \mathbf{x}_i \right\|_2}{\left\| \beta^{(T)} \mathbf{\Pi}_1 \sum_{i=1}^{N-1} \mathcal{S}_i^{(T)} \mathbf{x}_i \right\|_2}$$

$$\leq \frac{\exp(-\Omega(N)) \|\mathbf{V}^{(T)}\|_{\max} + \sqrt{K} \gamma^{(T)}}{\frac{\sqrt{2}}{2} \beta^{(T)}}$$

$$\leq \frac{\exp(-\Omega(N)) \left( \frac{\eta}{\epsilon K} + 2T\epsilon K^2 \right) + \sqrt{K} \left( \frac{2\eta}{\epsilon K} + 2T\epsilon K^2 N \exp(-\Omega(N)) \right)}{\frac{\sqrt{2}}{2} \cdot \left( \sqrt{\eta T} - \frac{2\eta}{\epsilon K} \right)}$$

$$\leq \mathcal{O} \left( \frac{1}{\sqrt{T}} \right)$$

where the second inequality is by Lemma D.8 and $\|\mathbf{\Pi}_1 \mathbf{x}_i\|_2 = \sqrt{2}/2$, and the third inequality is by Lemma D.8. Therefore, we can obtain that

$$\left\| \frac{f_{\theta^{(T)}}(\mathbf{X})}{\|f_{\theta^{(T)}}(\mathbf{X})\|_2} - \mathbf{\Pi}_1^\top \mathbf{x}_{N-1} \right\|_2 \leq \mathcal{O} \left( \frac{1}{\sqrt{T}} \right).$$

$\square$

# E   DETERMINISTIC WALK

In this section, we consider the case of the deterministic walk. We assume that the transition matrix is $\mathbf{\Pi} = \mathbf{\Pi}_2$. The following lemma shows the results of the first iteration.

**Lemma E.1.** If $\mathbf{\Pi} = \mathbf{\Pi}_2$, then it holds that

$$\boldsymbol{V}^{(1)} = \frac{\eta r}{\epsilon N K} \mathbf{1}_{K \times K} \text{ and } \boldsymbol{W}^{(1)} = \mathbf{0}_{(K+M) \times (K+M)}.$$

***Proof of Lemma E.1.*** By Lemma C.1, we have

$$\mathbb{E}[\nabla_{\boldsymbol{V}} \ell(\theta^{(0)})] = -\frac{1}{\epsilon N} \sum_{i=1}^{N-1} \mathbb{E}[\boldsymbol{e}_y \boldsymbol{x}_i^\top]$$

$$= -\frac{1}{\epsilon N} \sum_{i=1}^{N-1} \mathbb{E}[(\mathbf{\Pi}_2^\top)^{N-i} \boldsymbol{x}_i \boldsymbol{x}_i^\top]$$

$$= -\frac{1}{\epsilon N K} \sum_{i=1}^{N-1} (\mathbf{\Pi}_2^\top)^{N-i}$$

$$= -\frac{r}{\epsilon N K} \mathbf{1}_{K \times K},$$

where the first equation is by the initialization of $\boldsymbol{V}^{(0)}$ and $\boldsymbol{W}^{(0)}$, the second equation is by the sampling method, the third equation is by $\mathbb{E}[\boldsymbol{x}_i \boldsymbol{x}_i^\top] = \frac{1}{K} \boldsymbol{I}_K$ for $i \in [N-1]$ since $\boldsymbol{x}_i$ is uniformly distributed in $\boldsymbol{E}$, and the last equation is by Lemma F.2. Thus, by the update, we can get

$$\boldsymbol{V}^{(1)} = \boldsymbol{V}^{(0)} - \eta \mathbb{E}[\nabla_{\boldsymbol{V}} \ell(\theta^{(0)})] = \frac{\eta r}{\epsilon N K} \mathbf{1}_{K \times K}.$$

Since $\boldsymbol{V}^{(0)} = \mathbf{0}_{K \times K}$ and $\boldsymbol{W}^{(0)} = \mathbf{0}_{(K+M) \times (K+M)}$, we can get $\mathbb{E}[\nabla_{\boldsymbol{W}} \ell(\theta^{(0)})] = \mathbf{0}_{(K+M) \times (K+M)}$. Thus,

$$\boldsymbol{W}^{(1)} = \boldsymbol{W}^{(0)} - \eta \mathbb{E}[\nabla_{\boldsymbol{W}} \ell(\theta^{(0)})] = \mathbf{0}_{(K+M) \times (K+M)}.$$

$\square$

The following lemma states the results of the second iteration.

**Lemma E.2.** If $\mathbf{\Pi} = \mathbf{\Pi}_2$, then it holds that

$$\boldsymbol{V}^{(2)} = \left( \frac{\eta r}{\epsilon N K} + \frac{\eta \epsilon r N}{\eta r^2 K + \epsilon^2 N^2 K} \right) \mathbf{1}_{K \times K},$$

$$\boldsymbol{W}_{12}^{(2)} = \frac{\eta^2 r^2}{\eta r^2 N K + \epsilon^2 N^3 K} \mathbf{1}_K \boldsymbol{p}_N^\top,$$

$$\boldsymbol{W}_{22}^{(2)} = \left( \frac{\eta^2 r}{\eta r^2 N K + \epsilon^2 N^3 K} \sum_{i=1}^{N-1} \boldsymbol{p}_i - \frac{\eta^2 r^2}{\eta r^2 N + \epsilon^2 N^3} \boldsymbol{p}_N \right) \boldsymbol{p}_N^\top.$$

***Proof of Lemma E.2.*** By Lemma C.1, we have

$$\mathbb{E}[\nabla_{\boldsymbol{V}} \ell(\theta^{(1)})] = -\mathbb{E}\left[ \frac{1}{\boldsymbol{e}_y^\top \boldsymbol{V}^{(1)} \boldsymbol{X} \mathcal{S}^{(1)} + \epsilon} \cdot \boldsymbol{e}_y \sum_{i=1}^{N-1} \mathcal{S}_i^{(1)} \boldsymbol{x}_i^\top \right]$$

$$= -\frac{1}{\frac{\eta r^2}{\epsilon N} + \epsilon N} \sum_{i=1}^{N-1} \mathbb{E}[\boldsymbol{e}_y \boldsymbol{x}_i^\top]$$

$$= -\frac{\epsilon N}{\eta r^2 + \epsilon^2 N^2} \sum_{i=1}^{N-1} \mathbb{E}[(\mathbf{\Pi}_2^\top)^{N-i} \boldsymbol{x}_i \boldsymbol{x}_i^\top]$$

$$= -\frac{\epsilon N}{\eta r^2 K + \epsilon^2 N^2 K} \sum_{i=1}^{N-1} (\mathbf{\Pi}_2^\top)^{N-i}$$

$$= -\frac{\epsilon r N}{\eta r^2 K + \epsilon^2 N^2 K} \mathbf{1}_{K \times K},$$

where the second equation is by Lemma E.1, the third equation is by the sampling method, the fourth equation is by $\mathbb{E}[\boldsymbol{x}_i \boldsymbol{x}_i^\top] = \frac{1}{K}\mathbf{I}_K$ for $i \in [N-1]$ since $\boldsymbol{x}_i$ is uniformly distributed in $\boldsymbol{E}$, and the last equation is by Lemma F.2. Thus, we can get

$$\boldsymbol{V}^{(2)} = \boldsymbol{V}^{(1)} - \eta \mathbb{E}[\nabla_{\boldsymbol{V}} \ell(\theta^{(1)})]$$

$$= \frac{\eta r}{\epsilon N K}\mathbf{1}_{K \times K} + \frac{\eta \epsilon r N}{\eta r^2 K + \epsilon^2 N^2 K}\mathbf{1}_{K \times K}.$$

By Lemma C.1, we have

$$\mathbb{E}[\boldsymbol{A}^{(1)}] = \mathbb{E}\left[\left(\sum_{i=1}^{N-1} \mathcal{S}_i^{(1)} \boldsymbol{x}_i \boldsymbol{x}_i^\top (\boldsymbol{V}^{(1)})^\top \boldsymbol{e}_y - \sum_{i_1=1}^{N-1}\sum_{i_2=1}^{N-1} \mathcal{S}_{i_1}^{(1)} \mathcal{S}_{i_2}^{(1)} \boldsymbol{x}_{i_1} \boldsymbol{x}_{i_2}^\top (\boldsymbol{V}^{(1)})^\top \boldsymbol{e}_y\right)\boldsymbol{p}_N^\top\right]$$

$$= \mathbb{E}\left[\left(\frac{\eta r}{\epsilon N^2 K}\sum_{i=1}^{N-1} \boldsymbol{x}_i \boldsymbol{x}_i^\top \mathbf{1}_K - \frac{\eta r}{\epsilon N^3 K}\sum_{i_1=1}^{N-1}\sum_{i_2=1}^{N-1} \boldsymbol{x}_{i_1} \boldsymbol{x}_{i_2}^\top \mathbf{1}_K\right)\boldsymbol{p}_N^\top\right]$$

$$= \mathbb{E}\left[\left(\frac{\eta r}{\epsilon N^2 K}\sum_{i=1}^{N-1} \boldsymbol{x}_i - \frac{\eta r}{\epsilon N^3 K}\sum_{i_1=1}^{N-1}\sum_{i_2=1}^{N-1} \boldsymbol{x}_{i_1}\right)\boldsymbol{p}_N^\top\right]$$

$$= \left(\frac{\eta r^2}{\epsilon N^2 K}\mathbf{1}_K - \frac{\eta r^2 (N-1)}{\epsilon N^3 K}\mathbf{1}_K\right)\boldsymbol{p}_N^\top$$

$$= \frac{\eta r^2}{\epsilon N^3 K}\mathbf{1}_K \boldsymbol{p}_N^\top,$$

where the second equation is by Lemma E.1, and the fourth equation is by the fact that all the $\boldsymbol{x}_i$ is uniformly distributed in $\boldsymbol{E}$. We also have

$$\mathbb{E}[\boldsymbol{B}^{(1)}] = \mathbb{E}\left[\left(\sum_{i=1}^{N-1} \mathcal{S}_i^{(1)} \boldsymbol{p}_i \boldsymbol{x}_i^\top (\boldsymbol{V}^{(1)})^\top \boldsymbol{e}_y - \sum_{i=1}^{N} \mathcal{S}_i^{(1)} \boldsymbol{p}_i \cdot \sum_{i=1}^{N-1} \mathcal{S}_i^{(1)} \boldsymbol{x}_i^\top (\boldsymbol{V}^{(1)})^\top \boldsymbol{e}_y\right)\boldsymbol{p}_N^\top\right]$$

$$= \mathbb{E}\left[\left(\frac{\eta r}{\epsilon N^2 K}\sum_{i=1}^{N-1} \boldsymbol{p}_i \boldsymbol{x}_i^\top \mathbf{1}_K - \frac{\eta r}{\epsilon N^3 K}\sum_{i=1}^{N} \boldsymbol{p}_i \cdot \sum_{i=1}^{N-1} \boldsymbol{x}_i^\top \mathbf{1}_K\right)\boldsymbol{p}_N^\top\right]$$

$$= \left(\frac{\eta r}{\epsilon N^2 K}\sum_{i=1}^{N-1} \boldsymbol{p}_i - \frac{\eta r (N-1)}{\epsilon N^3 K}\sum_{i=1}^{N} \boldsymbol{p}_i\right)\boldsymbol{p}_N^\top$$

$$= \left(\frac{\eta r}{\epsilon N^3 K}\sum_{i=1}^{N-1} \boldsymbol{p}_i - \frac{\eta r^2}{\epsilon N^3}\boldsymbol{p}_N\right)\boldsymbol{p}_N^\top,$$

where the second equation is by Lemma E.1. Thus, we can get that

$$\boldsymbol{W}_{12}^{(2)} = \boldsymbol{W}_{12}^{(1)} - \eta \mathbb{E}[\nabla_{\boldsymbol{W}} \ell(\theta^{(1)})]_{12}$$

$$= \mathbb{E}\left[\frac{\eta}{\boldsymbol{e}_y^\top \boldsymbol{V}^{(1)} \boldsymbol{X} \mathcal{S}^{(1)} + \epsilon} \cdot \boldsymbol{A}^{(1)}\right]$$

$$= \frac{\eta}{\frac{\eta r^2}{\epsilon N^2} + \epsilon} \cdot \frac{\eta r^2}{\epsilon N^3 K}\mathbf{1}_K \boldsymbol{p}_N^\top$$

$$= \frac{\eta^2 r^2}{\eta r^2 N K + \epsilon^2 N^3 K}\mathbf{1}_K \boldsymbol{p}_N^\top,$$

and

$$\boldsymbol{W}_{22}^{(2)} = \boldsymbol{W}_{22}^{(1)} - \eta \mathbb{E}[\nabla_{\boldsymbol{W}} \ell(\theta^{(1)})]_{22}$$

$$= \mathbb{E}\left[\frac{\eta}{\boldsymbol{e}_y^\top \boldsymbol{V}^{(1)} \boldsymbol{X} \mathcal{S}^{(1)} + \epsilon} \cdot \boldsymbol{B}^{(1)}\right]$$

$$= \frac{\eta}{\frac{\eta r^2}{\epsilon N^2} + \epsilon} \cdot \left(\frac{\eta r}{\epsilon N^3 K}\sum_{i=1}^{N-1} \boldsymbol{p}_i - \frac{\eta r^2}{\epsilon N^3}\boldsymbol{p}_N\right)\boldsymbol{p}_N^\top$$

$$= \left( \frac{\eta^2 r}{\eta r^2 NK + \epsilon^2 N^3 K} \sum_{i=1}^{N-1} \boldsymbol{p}_i - \frac{\eta^2 r^2}{\eta r^2 N + \epsilon^2 N^3} \boldsymbol{p}_N \right) \boldsymbol{p}_N^\top.$$

$\square$

Next, we can analyze the gradient descent dynamics over multiple iterations.

**Lemma E.3.** If $\boldsymbol{\Pi} = \boldsymbol{\Pi}_2$, then for any $t \geq 0$ and any sequence of learning rates $\{\eta_t\}$, it holds that

$$\boldsymbol{V}^{(t)} \propto \mathbf{1}_{K \times K}, \text{ and } \mathcal{S}_1^{(t)} = \mathcal{S}_2^{(t)} = \cdots = \mathcal{S}_{N-1}^{(t)}.$$

***Proof of Lemma E.3.*** We use induction to prove that for some scalar $\alpha_1^{(t)}, \alpha_2^{(t)}, \alpha_3^{(t)}, \alpha_4^{(t)}$, it holds that for $t \geq 2$, $\boldsymbol{V}^{(t)} = \alpha_1^{(t)} \mathbf{1}_{K \times K}$, $\boldsymbol{W}_{12}^{(t)} = \alpha_2^{(t)} \mathbf{1}_K \boldsymbol{p}_N^\top$, and $\boldsymbol{W}_{22}^{(t)} = \left( \alpha_3^{(t)} \sum_{i=1}^{N-1} \boldsymbol{p}_i - \alpha_4^{(t)} \boldsymbol{p}_N \right) \boldsymbol{p}_N^\top$. By Lemma E.2, we know that the hypothesis holds for $t = 2$. Suppose that the hypothesis holds for $t = t'$. We aim to prove that the hypothesis holds for $t = t' + 1$. We have

$$\widetilde{\boldsymbol{X}} \boldsymbol{W}^{(t')} \widetilde{\boldsymbol{x}}_N = \begin{bmatrix} \boldsymbol{x}_1^\top \boldsymbol{W}_{12}^{(t')} \boldsymbol{p}_N + \boldsymbol{p}_1^\top \boldsymbol{W}_{22}^{(t')} \boldsymbol{p}_N \\ \boldsymbol{x}_2^\top \boldsymbol{W}_{12}^{(t')} \boldsymbol{p}_N + \boldsymbol{p}_2^\top \boldsymbol{W}_{22}^{(t')} \boldsymbol{p}_N \\ \vdots \\ \boldsymbol{x}_{N-1}^\top \boldsymbol{W}_{12}^{(t')} \boldsymbol{p}_N + \boldsymbol{p}_{N-1}^\top \boldsymbol{W}_{22}^{(t')} \boldsymbol{p}_N \\ \boldsymbol{p}_N^\top \boldsymbol{W}_{22}^{(t')} \boldsymbol{p}_N \end{bmatrix}$$

$$= \begin{bmatrix} \alpha_2^{(t')} \boldsymbol{p}_N^\top \boldsymbol{p}_N + \alpha_3^{(t')} \boldsymbol{p}_1^\top \boldsymbol{p}_1 \boldsymbol{p}_N^\top \boldsymbol{p}_N \\ \alpha_2^{(t')} \boldsymbol{p}_N^\top \boldsymbol{p}_N + \alpha_3^{(t')} \boldsymbol{p}_2^\top \boldsymbol{p}_2 \boldsymbol{p}_N^\top \boldsymbol{p}_N \\ \vdots \\ \alpha_2^{(t')} \boldsymbol{p}_N^\top \boldsymbol{p}_N + \alpha_3^{(t')} \boldsymbol{p}_{N-1}^\top \boldsymbol{p}_{N-1} \boldsymbol{p}_N^\top \boldsymbol{p}_N \\ -\alpha_4^{(t')} (\boldsymbol{p}_N^\top \boldsymbol{p}_N)^2 \end{bmatrix}. \tag{E.1}$$

Since $\boldsymbol{p}_1^\top \boldsymbol{p}_1 = \boldsymbol{p}_2^\top \boldsymbol{p}_2 = \cdots = \boldsymbol{p}_N^\top \boldsymbol{p}_N$, we have $[\widetilde{\boldsymbol{X}} \boldsymbol{W}^{(t')} \widetilde{\boldsymbol{x}}_N]_1 = [\widetilde{\boldsymbol{X}} \boldsymbol{W}^{(t')} \widetilde{\boldsymbol{x}}_N]_2 = \cdots = [\widetilde{\boldsymbol{X}} \boldsymbol{W}^{(t')} \widetilde{\boldsymbol{x}}_N]_{N-1}$. Thus, we can get that $\mathcal{S}_1^{(t')} = \mathcal{S}_2^{(t')} = \cdots = \mathcal{S}_{N-1}^{(t')} := s^{(t')}$. Then, we have

$$\mathbb{E}[\nabla_{\boldsymbol{V}} \ell(\theta^{(t')})] = -\mathbb{E}\left[ \frac{1}{\boldsymbol{e}_y^\top \boldsymbol{V}^{(t')} \boldsymbol{X} \mathcal{S}^{(t')} + \epsilon} \cdot \boldsymbol{e}_y \sum_{i=1}^{N-1} \mathcal{S}_i^{(t')} \boldsymbol{x}_i^\top \right]$$

$$= -\mathbb{E}\left[ \frac{1}{\alpha_1^{(t')} s^{(t')} (N-1) + \epsilon} \cdot \boldsymbol{e}_y \sum_{i=1}^{N-1} s^{(t')} \boldsymbol{x}_i^\top \right]$$

$$= -\frac{s^{(t')}}{\alpha_1^{(t')} s^{(t')} (N-1) + \epsilon} \sum_{i=1}^{N-1} \mathbb{E}[\boldsymbol{e}_y \boldsymbol{x}_i^\top]$$

$$= -\frac{s^{(t')}}{\alpha_1^{(t')} s^{(t')} (N-1) + \epsilon} \sum_{i=1}^{N-1} \mathbb{E}[(\boldsymbol{\Pi}_2^\top)^{N-i} \boldsymbol{x}_i \boldsymbol{x}_i^\top]$$

$$= -\frac{s^{(t')}}{\alpha_1^{(t')} s^{(t')} (N-1) K + \epsilon K} \sum_{i=1}^{N-1} (\boldsymbol{\Pi}_2^\top)^{N-i}$$

$$= -\frac{s^{(t')} r}{\alpha_1^{(t')} s^{(t')} (N-1) K + \epsilon K} \mathbf{1}_{K \times K},$$

where the second equation is by the induction, the fourth equation is by the sampling method, the fifth equation is by the fact that $\boldsymbol{x}_i$ is uniformly distributed in $\boldsymbol{E}$, and the last equation is by Lemma F.2. Thus, we can get $\boldsymbol{V}^{(t'+1)} = \boldsymbol{V}^{(t')} - \eta^{(t')} \mathbb{E}[\nabla_{\boldsymbol{V}} \ell(\theta^{(t')})] \propto \mathbf{1}_{K \times K}$. We also have

$$\mathbb{E}[\boldsymbol{A}^{(t')}] = \mathbb{E}\left[ \left( \sum_{i=1}^{N-1} \mathcal{S}_i^{(t')} \boldsymbol{x}_i \boldsymbol{x}_i^\top (\boldsymbol{V}^{(t')})^\top \boldsymbol{e}_y - \sum_{i_1=1}^{N-1} \sum_{i_2=1}^{N-1} \mathcal{S}_{i_1}^{(t')} \mathcal{S}_{i_2}^{(t')} \boldsymbol{x}_{i_1} \boldsymbol{x}_{i_2}^\top (\boldsymbol{V}^{(t')})^\top \boldsymbol{e}_y \right) \boldsymbol{p}_N^\top \right]$$

$$= \mathbb{E}\left[\left(\alpha_1^{(t')} s^{(t')} \sum_{i=1}^{N-1} \boldsymbol{x}_i \boldsymbol{x}_i^\top \mathbf{1}_K - \alpha_1^{(t')} (s^{(t')})^2 \sum_{i_1=1}^{N-1} \sum_{i_2=1}^{N-1} \boldsymbol{x}_{i_1} \boldsymbol{x}_{i_2}^\top \mathbf{1}_K\right) \boldsymbol{p}_N^\top\right]$$

$$= \mathbb{E}\left[\left(\alpha_1^{(t')} s^{(t')} \sum_{i=1}^{N-1} \boldsymbol{x}_i - \alpha_1^{(t')} (s^{(t')})^2 \sum_{i_1=1}^{N-1} \sum_{i_2=1}^{N-1} \boldsymbol{x}_{i_1}\right) \boldsymbol{p}_N^\top\right]$$

$$= \left(\frac{\alpha_1^{(t')} s^{(t')} (N-1)}{K} \mathbf{1}_K - \frac{\alpha_1^{(t')} (s^{(t')})^2 (N-1)^2}{K} \mathbf{1}_K\right) \boldsymbol{p}_N^\top,$$

where the second equation is by the induction, and the fourth equation is by the fact that all the $\boldsymbol{x}_i$ is uniformly distributed in $\boldsymbol{E}$. And,

$$\mathbb{E}[\boldsymbol{B}^{(t')}] = \mathbb{E}\left[\left(\sum_{i=1}^{N-1} \mathcal{S}_i^{(t')} \boldsymbol{p}_i \boldsymbol{x}_i^\top (\boldsymbol{V}^{(t')})^\top \boldsymbol{e}_y - \sum_{i=1}^{N} \mathcal{S}_i^{(t')} \boldsymbol{p}_i \cdot \sum_{i=1}^{N-1} \mathcal{S}_i^{(t')} \boldsymbol{x}_i^\top (\boldsymbol{V}^{(t')})^\top \boldsymbol{e}_y\right) \boldsymbol{p}_N^\top\right]$$

$$= \mathbb{E}\left[\left(\alpha_1^{(t')} s^{(t')} \sum_{i=1}^{N-1} \boldsymbol{p}_i \boldsymbol{x}_i^\top \mathbf{1}_K - \alpha_1^{(t')} (s^{(t')})^2 \sum_{i=1}^{N} \boldsymbol{p}_i \cdot \sum_{i=1}^{N-1} \boldsymbol{x}_i^\top \mathbf{1}_K\right) \boldsymbol{p}_N^\top\right]$$

$$= \left(\alpha_1^{(t')} s^{(t')} \sum_{i=1}^{N-1} \boldsymbol{p}_i - \alpha_1^{(t')} (s^{(t')})^2 (N-1) \sum_{i=1}^{N} \boldsymbol{p}_i\right) \boldsymbol{p}_N^\top,$$

where the second equation is by the induction. Therefore, we can get

$$\boldsymbol{W}_{12}^{(t'+1)} = \boldsymbol{W}_{12}^{(t')} - \eta \mathbb{E}[\nabla_{\boldsymbol{W}} \ell(\theta^{(t')})]_{12}$$

$$= \alpha_2^{(t')} \mathbf{1}_K \boldsymbol{p}_N^\top + \frac{\eta}{\frac{\eta r^2}{\epsilon N^2} + \epsilon} \mathbb{E}[\boldsymbol{A}^{(t')}]$$

$$= \alpha_2^{(t')} \mathbf{1}_K \boldsymbol{p}_N^\top + \frac{\eta}{\frac{\eta r^2}{\epsilon N^2} + \epsilon} \left(\frac{\alpha_1^{(t')} s^{(t')} (N-1)}{K} - \frac{\alpha_1^{(t')} (s^{(t')})^2 (N-1)^2}{K}\right) \mathbf{1}_K \boldsymbol{p}_N^\top$$

$$:= \alpha_2^{(t'+1)} \mathbf{1}_K \boldsymbol{p}_N^\top,$$

and

$$\boldsymbol{W}_{22}^{(t'+1)} = \boldsymbol{W}_{22}^{(t')} - \eta \mathbb{E}[\nabla_{\boldsymbol{W}} \ell(\theta^{(t')})]_{22}$$

$$= \left(\alpha_3^{(t')} \sum_{i=1}^{N-1} \boldsymbol{p}_i - \alpha_4^{(t')} \boldsymbol{p}_N\right) \boldsymbol{p}_N^\top + \frac{\eta}{\frac{\eta r^2}{\epsilon N^2} + \epsilon} \mathbb{E}[\boldsymbol{B}^{(t')}]$$

$$= \left(\alpha_3^{(t')} \sum_{i=1}^{N-1} \boldsymbol{p}_i - \alpha_4^{(t')} \boldsymbol{p}_N\right) \boldsymbol{p}_N^\top$$

$$+ \frac{\eta}{\frac{\eta r^2}{\epsilon N^2} + \epsilon} \left(\alpha_1^{(t')} s^{(t')} \sum_{i=1}^{N-1} \boldsymbol{p}_i - \alpha_1^{(t')} (s^{(t')})^2 (N-1) \sum_{i=1}^{N} \boldsymbol{p}_i\right) \boldsymbol{p}_N^\top$$

$$:= \left(\alpha_3^{(t'+1)} \sum_{i=1}^{N-1} \boldsymbol{p}_i - \alpha_4^{(t'+1)} \boldsymbol{p}_N\right) \boldsymbol{p}_N^\top.$$

Therefore, by induction, we can conclude that for all $t \geq 2$, $\boldsymbol{V}^{(t)} = \alpha_1^{(t)} \mathbf{1}_{K \times K}$, $\boldsymbol{W}_{12}^{(t)} = \alpha_2^{(t)} \mathbf{1}_K \boldsymbol{p}_N^\top$, and $\boldsymbol{W}_{22}^{(t)} = \left(\alpha_3^{(t)} \sum_{i=1}^{N-1} \boldsymbol{p}_i - \alpha_4^{(t)} \boldsymbol{p}_N\right) \boldsymbol{p}_N^\top$. Similar to (E.1), we have $[\widetilde{\boldsymbol{X}} \boldsymbol{W}^{(t)} \widetilde{\boldsymbol{x}}_N]_1 = [\widetilde{\boldsymbol{X}} \boldsymbol{W}^{(t)} \widetilde{\boldsymbol{x}}_N]_2 = \cdots = [\widetilde{\boldsymbol{X}} \boldsymbol{W}^{(t)} \widetilde{\boldsymbol{x}}_N]_{N-1}$, which implies that $\mathcal{S}_1^{(t)} = \mathcal{S}_2^{(t)} = \cdots = \mathcal{S}_{N-1}^{(t)}$. $\qquad\square$

## F AUXILIARY LEMMAS

In this section, we present some auxiliary lemmas. The following lemma provides the bound of some combinatorial numbers.

**Lemma F.1.** For all $n \in \mathbb{N}$, it holds that

$$\binom{2n}{n} < \frac{2^{2n}}{\sqrt{2n+1}} \text{ and } \binom{2n+1}{n} < \frac{2^{2n+1}}{\sqrt{2n+3}}.$$

***Proof of Lemma F.1.*** For $n \in \mathbb{N}$, we have

$$\binom{2n}{n} = 2^{2n} \cdot \frac{(2n-1)!!}{(2n)!!}$$

$$= 2^{2n} \cdot \prod_{k=1}^{n} \frac{2k-1}{2k}$$

$$< 2^{2n} \cdot \prod_{k=1}^{n} \frac{\sqrt{2k-1}\sqrt{2k-1}}{\sqrt{2k-1}\sqrt{2k+1}}$$

$$= 2^{2n} \cdot \prod_{k=1}^{n} \frac{\sqrt{2k-1}}{\sqrt{2k+1}}$$

$$= \frac{2^{2n}}{\sqrt{2n+1}}.$$

We also have

$$\binom{2n+1}{n} = 2^{2n+1} \cdot \frac{(2n+1)!!}{(2n+2)!!}$$

$$= 2^{2n+1} \cdot \prod_{k=1}^{n+1} \frac{2k-1}{2k}$$

$$< 2^{2n+1} \cdot \prod_{k=1}^{n+1} \frac{\sqrt{2k-1}\sqrt{2k-1}}{\sqrt{2k-1}\sqrt{2k+1}}$$

$$= 2^{2n+1} \cdot \prod_{k=1}^{n+1} \frac{\sqrt{2k-1}}{\sqrt{2k+1}}$$

$$= \frac{2^{2n+1}}{\sqrt{2n+3}}.$$

$\square$

The following lemma states the properties of $\mathbf{\Pi}_0$.

**Lemma F.2.** $\mathbf{\Pi}_0^K = \mathbf{I}_K$, $\mathbf{\Pi}_0\mathbf{\Pi}_0^\top = \mathbf{I}_K$, and $\sum_{k=1}^{K} \mathbf{\Pi}_0^k = \mathbf{1}_{K \times K}$.

***Proof of Lemma F.2.*** In this proof, the index $i$ larger than $K$ represents $i - K$. For $\mathbf{\Pi}_0$, only $[\mathbf{\Pi}_0]_{i+1,i} = 1$ for $i \in [K]$ and other elements are 0. We can get that for $\mathbf{\Pi}_0^k$, only $[\mathbf{\Pi}_0^k]_{i+k,i} = 1$ for $i \in [K]$ and other elements are 0. By this observation, we can derive that $\mathbf{\Pi}_0^K = \mathbf{I}_K$ and $\sum_{k=1}^{K} \mathbf{\Pi}_0^k = \mathbf{1}_{K \times K}$. Also, we have $\mathbf{\Pi}_0^\top = \mathbf{\Pi}_0^{K-1}$, so we can get $\mathbf{\Pi}_0\mathbf{\Pi}_0^\top = \mathbf{\Pi}_0^K = \mathbf{I}_K$. $\square$

The following two lemmas show some properties of $\mathbf{\Pi}_1$.

**Lemma F.3.** Assume that $R = rK + l$ with $r \in \mathbb{N}$ and $l \in \mathbb{N}$. For the case that $K$ is even and $l$ is odd,

$$\frac{1}{K} \operatorname{tr}(\mathbf{\Pi}_1^R) = 0.$$

For the case that $K$ is even and $l$ is odd,

$$\frac{1}{K} \operatorname{tr}(\mathbf{\Pi}_1^R) < \frac{2}{K} + \frac{1}{\sqrt{R+1}}.$$

For the case that $K$ is odd,

$$\frac{1}{K} \operatorname{tr}(\mathbf{\Pi}_1^R) < \frac{1}{K} + \frac{2}{\sqrt{R+1}}.$$

***Proof of Lemma F.3.*** By Lemma F.2, we know that

$$\frac{1}{K}\operatorname{tr}(\mathbf{\Pi}_0^P) = \begin{cases} 1, & \text{if } P = rK \text{ with } r \in \mathbb{N}; \\ 0, & \text{otherwise.} \end{cases} \tag{F.1}$$

Then, we can get

$$\frac{1}{K}\operatorname{tr}(\mathbf{\Pi}_1^R) = \frac{1}{K \cdot 2^R}\operatorname{tr}((\mathbf{\Pi}_0 + \mathbf{\Pi}_0^\top)^R)$$

$$= \frac{1}{K \cdot 2^R}\operatorname{tr}\left(\sum_{k=0}^{R}\binom{R}{k}\mathbf{\Pi}_0^{R-2k}\right)$$

$$= \frac{1}{2^R}\sum_{k=0}^{R}\binom{R}{k}\mathbf{1}_{\{K|2k-l\}}, \tag{F.2}$$

where the last equation is by (F.1).

When $K$ is even and $l$ is odd, we can directly get that

$$\frac{1}{K}\operatorname{tr}(\mathbf{\Pi}_1^R) = 0$$

by (F.2).

When $K$ is even and $l$ is even, we can get

$$\frac{1}{K}\operatorname{tr}(\mathbf{\Pi}_1^R) = \frac{1}{K \cdot 2^R}\operatorname{tr}((\mathbf{\Pi}_0 + \mathbf{\Pi}_0^\top)^R)$$

$$= \frac{1}{K \cdot 2^R}\operatorname{tr}\left(\sum_{k=0}^{R}\binom{R}{k}\mathbf{\Pi}_0^{R-2k}\right)$$

$$\overset{(i)}{=} \frac{1}{2^R}\sum_{s=0}^{2r}\binom{R}{s \cdot \frac{K}{2} + \frac{l}{2}}$$

$$\leq \frac{1}{2^R}\frac{2}{K}\sum_{s=0}^{r-1}\sum_{u=l/2}^{l/2+K/2-1}\binom{R}{s \cdot \frac{K}{2} + u} + \frac{1}{2^R}\frac{2}{K}\sum_{s=r+1}^{2r}\sum_{u=l/2-K/2+1}^{l/2}\binom{R}{s \cdot \frac{K}{2} + u}$$

$$+ \frac{1}{2^R}\binom{R}{R/2}$$

$$\overset{(ii)}{\leq} \frac{1}{2^R}\frac{2}{K}\left(2^R - \binom{R}{R/2}\right) + \frac{1}{2^R}\binom{R}{R/2}$$

$$= \frac{2}{K} + \frac{1}{2^R}\frac{K-2}{K}\binom{R}{R/2}$$

$$\overset{(iii)}{<} \frac{2}{K} + \frac{1}{\sqrt{R+1}},$$

where $(i)$ is by Lemma F.2, $(ii)$ is by $\sum_{k=0}^{R}\binom{R}{k} = 2^R$, and $(iii)$ is by Lemma F.1.

When $K$ is odd, $l$ is even and $r$ is even, we can get

$$\frac{1}{K}\operatorname{tr}(\mathbf{\Pi}_1^R) = \frac{1}{K \cdot 2^R}\operatorname{tr}((\mathbf{\Pi}_0 + \mathbf{\Pi}_0^\top)^R)$$

$$= \frac{1}{K \cdot 2^R}\operatorname{tr}\left(\sum_{k=0}^{R}\binom{R}{k}\mathbf{\Pi}_0^{R-2k}\right)$$

$$\overset{(i)}{=} \frac{1}{2^R}\sum_{s=0}^{r}\binom{R}{sK + \frac{l}{2}}$$

$$\leq \frac{1}{2^R}\frac{1}{K}\sum_{s=0}^{r/2-1}\sum_{u=l/2}^{l/2+K-1}\binom{R}{sK + u} + \frac{1}{2^R}\frac{1}{K}\sum_{s=r/2+1}^{r}\sum_{u=l/2-K+1}^{l/2}\binom{R}{sK + u}$$

$$+ \frac{1}{2^R}\binom{R}{R/2}$$

$$\overset{(ii)}{\leq} \frac{1}{2^R}\frac{1}{K}\left(2^R - \binom{R}{R/2}\right) + \frac{1}{2^R}\binom{R}{R/2}$$

$$= \frac{1}{K} + \frac{1}{2^R}\frac{K-1}{K}\binom{R}{R/2}$$

$$\overset{(iii)}{<} \frac{1}{K} + \frac{1}{\sqrt{R+1}},$$

where $(i)$ is by Lemma F.2, $(ii)$ is by $\sum_{k=0}^{R}\binom{R}{k} = 2^R$, and $(iii)$ is by Lemma F.1.

When $K$ is odd, $l$ is even and $r$ is odd, we can get

$$\frac{1}{K}\operatorname{tr}(\mathbf{\Pi}_1^R) = \frac{1}{K\cdot 2^R}\operatorname{tr}((\mathbf{\Pi}_0 + \mathbf{\Pi}_0^\top)^R)$$

$$= \frac{1}{K\cdot 2^R}\operatorname{tr}\left(\sum_{k=0}^{R}\binom{R}{k}\mathbf{\Pi}_0^{R-2k}\right)$$

$$\overset{(i)}{=} \frac{1}{2^R}\sum_{s=0}^{r}\binom{R}{sK + \frac{l}{2}}$$

$$= \frac{1}{2^R}\sum_{s=0}^{(r-1)/2-1}\binom{R}{sK + \frac{l}{2}} + \frac{1}{2^R}\sum_{s=(r+1)/2+1}^{r}\binom{R}{sK + \frac{l}{2}}$$

$$+ \frac{1}{2^R}\left[\binom{R}{\frac{r-1}{2}K + \frac{l}{2}} + \binom{R}{\frac{r+1}{2}K + \frac{l}{2}}\right]$$

$$\leq \frac{1}{2^R}\frac{1}{K}\sum_{s=0}^{(r-1)/2-1}\sum_{u=l/2}^{l/2+K-1}\binom{R}{sK + u} + \frac{1}{2^R}\frac{1}{K}\sum_{s=(r+1)/2+1}^{r}\sum_{u=l/2-K+1}^{l/2}\binom{R}{sK + u}$$

$$+ \frac{1}{2^R}\left[\binom{R}{\frac{R-1}{2}} + \binom{R}{\frac{R+1}{2}}\right]$$

$$\overset{(ii)}{<} \frac{1}{K} + \frac{2}{\sqrt{R+2}},$$

where $(i)$ is by Lemma F.2, $(ii)$ is by $\sum_{k=0}^{R}\binom{R}{k} = 2^R$ and Lemma F.1.

When $K$ is odd, $l$ is odd and $r$ is even, we can get

$$\frac{1}{K}\operatorname{tr}(\mathbf{\Pi}_1^R) = \frac{1}{K\cdot 2^R}\operatorname{tr}((\mathbf{\Pi}_0 + \mathbf{\Pi}_0^\top)^R)$$

$$= \frac{1}{K\cdot 2^R}\operatorname{tr}\left(\sum_{k=0}^{R}\binom{R}{k}\mathbf{\Pi}_0^{R-2k}\right)$$

$$\overset{(i)}{=} \frac{1}{2^R}\sum_{s=0}^{r-1}\binom{R}{sK + \frac{K+l}{2}}$$

$$= \frac{1}{2^R}\sum_{s=0}^{(r-2)/2-1}\binom{R}{sK + \frac{K+l}{2}} + \frac{1}{2^R}\sum_{r/2+1}^{r-1}\binom{R}{sK + \frac{K+l}{2}}$$

$$+ \frac{1}{2^R}\left[\binom{R}{\frac{r-2}{2}K + \frac{K+l}{2}} + \binom{R}{\frac{r}{2}K + \frac{K+l}{2}}\right]$$

$$\leq \frac{1}{2^R}\frac{1}{K}\sum_{s=0}^{(r-2)/2-1}\sum_{u=(K+l)/2}^{(K+l)/2+K-1}\binom{R}{sK + u}$$

$$+ \frac{1}{2^R} \frac{1}{K} \sum_{s=r/2+1}^{r-1} \sum_{u=(K+l)/2-K+1}^{(K+l)/2} \binom{R}{sK+u} + \frac{1}{2^R}\left[\binom{R}{\frac{R-1}{2}} + \binom{R}{\frac{R+1}{2}}\right]$$

$$\overset{(ii)}{<} \frac{1}{K} + \frac{2}{\sqrt{R+2}},$$

where $(i)$ is by Lemma F.2, $(ii)$ is by $\sum_{k=0}^R \binom{R}{k} = 2^R$ and Lemma F.1.

When $K$ is odd, $l$ is odd and $r$ is odd, we can get

$$\frac{1}{K}\operatorname{tr}(\mathbf{\Pi}_1^R) = \frac{1}{K \cdot 2^R}\operatorname{tr}((\mathbf{\Pi}_0 + \mathbf{\Pi}_0^\top)^R)$$

$$= \frac{1}{K \cdot 2^R}\operatorname{tr}\left(\sum_{k=0}^R \binom{R}{k}\mathbf{\Pi}_0^{R-2k}\right)$$

$$\overset{(i)}{=} \frac{1}{2^R}\sum_{s=0}^{r-1}\binom{R}{sK + \frac{K+l}{2}}$$

$$\leq \frac{1}{2^R}\frac{1}{K}\sum_{s=0}^{(r-1)/2-1}\sum_{u=(K+l)/2}^{(K+l)/2+K-1}\binom{R}{sK+u}$$

$$+ \frac{1}{2^R}\frac{1}{K}\sum_{s=(r-1)/2+1}^{r-1}\sum_{u=(K+l)/2-K+1}^{(K+l)/2}\binom{R}{sK+u} + \frac{1}{2^R}\binom{R}{R/2}$$

$$\overset{(ii)}{<} \frac{1}{K} + \frac{1}{\sqrt{R+1}},$$

where $(i)$ is by Lemma F.2, $(ii)$ is by $\sum_{k=0}^R \binom{R}{k} = 2^R$ and Lemma F.1. □

**Lemma F.4.** Assume that $R \geq K$ and $K$ is even. For the case that $R$ is even,

$$\left[\mathbf{\Pi}_1^R\right]_{i,j} = 0 \text{ for odd } (j-i);$$

$$\left[\mathbf{\Pi}_1^R\right]_{i,j} \geq \frac{2}{K} - \frac{2}{\sqrt{R+1}} \text{ for even } (j-i).$$

For the case that $R$ is even,

$$\left[\mathbf{\Pi}_1^R\right]_{i,j} \geq \frac{2}{K} - \frac{2}{\sqrt{R+1}} \text{ for odd } (j-i);$$

$$\left[\mathbf{\Pi}_1^R\right]_{i,j} = 0 \text{ for even } (j-i).$$

***Proof of Lemma F.4.*** First, we can get that

$$\left[\mathbf{\Pi}_1^R\right]_{i,j} = \frac{1}{2^R}\left[(\mathbf{\Pi}_0 + \mathbf{\Pi}_0^\top)^R\right]_{i,j}$$

$$= \frac{1}{2^R}\left[\left(\sum_{k=0}^R \binom{R}{k}\mathbf{\Pi}_0^{R-2k}\right)^R\right]_{i,j}$$

$$= \frac{1}{2^R}\sum_{k=0}^R \binom{R}{k}\mathbf{1}\{K|R-2k-j+i\}. \tag{F.3}$$

Next, we consider two cases and assume that $R = rK + l$. We can easily observe that

$$\frac{K}{2}\binom{R}{s} \geq \sum_{i=0}^{K/2-1}\binom{R}{s-i} \text{ for } s \leq R/2; \tag{F.4}$$

$$\frac{K}{2}\binom{R}{s} \geq \sum_{i=0}^{K/2-1}\binom{R}{s+i} \text{ for } s \geq R/2. \tag{F.5}$$

**Condition 1:** $R$ is even. We can directly get by (F.3) that $\left[\mathbf{\Pi}_1^R\right]_{i,j} = 0$ for odd $(j-i)$. When $(j-i)$ is even, we have

$$
\begin{aligned}
\left[\mathbf{\Pi}_1^R\right]_{i,j} &= \frac{1}{2^R} \sum_{k=0}^{R} \binom{R}{k} \mathbf{1}\{K|R-2k-j+i\} \\
&= \frac{1}{2^R} \sum_{s=0}^{2r} \binom{R}{s \cdot \frac{K}{2} + \frac{l}{2} - \frac{j-i}{2}} \\
&= \frac{1}{2^R} \frac{2}{K} \left( \sum_{s=0}^{2r} \frac{K}{2} \binom{R}{s \cdot \frac{K}{2} + \frac{l}{2} - \frac{j-i}{2}} + K \binom{R}{R/2} \right) - \frac{2}{2^R} \binom{R}{R/2} \\
&\geq \frac{1}{2^R} \frac{2}{K} \sum_{s=0}^{R} \binom{R}{s} - \frac{2}{2^R} \binom{R}{R/2} \\
&\geq \frac{2}{K} - \frac{2}{\sqrt{R+1}},
\end{aligned}
$$

where the first inequality is by (F.4) and (F.5), and the second inequality is by Lemma F.1.

**Condition 2:** $R$ is odd. We can directly get by (F.3) that $\left[\mathbf{\Pi}_1^R\right]_{i,j} = 0$ for even $(j-i)$. When $(j-i)$ is old, we have

$$
\begin{aligned}
\left[\mathbf{\Pi}_1^R\right]_{i,j} &= \frac{1}{2^R} \sum_{k=0}^{R} \binom{R}{k} \mathbf{1}\{K|R-2k-j+i\} \\
&= \frac{1}{2^R} \sum_{s=0}^{2r} \binom{R}{s \cdot \frac{K}{2} + \frac{l-(j-i)}{2}} \\
&= \frac{1}{2^R} \frac{2}{K} \left( \sum_{s=0}^{2r} \frac{K}{2} \binom{R}{s \cdot \frac{K}{2} + \frac{l-(j-i)}{2}} + K \binom{R}{R/2} \right) - \frac{2}{2^R} \binom{R}{R/2} \\
&\geq \frac{1}{2^R} \frac{2}{K} \sum_{s=0}^{R} \binom{R}{s} - \frac{2}{2^R} \binom{R}{R/2} \\
&\geq \frac{2}{K} - \frac{2}{\sqrt{R+1}},
\end{aligned}
$$

where the first inequality is by (F.4) and (F.5), and the second inequality is by Lemma F.1. $\qquad\square$

The following lemma shows the basic property of the positional embedding.

**Lemma F.5.** Assume that

$$
\boldsymbol{p}_i = \left[ \sin\left( \frac{i\pi}{M+1} \right), \sin\left( \frac{2i\pi}{M+1} \right), \ldots, \sin\left( \frac{Mi\pi}{M+1} \right) \right]^\top
$$

for $i \in [M]$. It holds that

$$
\boldsymbol{p}_{i_1}^\top \boldsymbol{p}_{i_2} = \begin{cases} \frac{M+1}{2} & \text{for } i_1 = i_2; \\ 0 & \text{for } i_1 \neq i_2. \end{cases}
$$

***Proof of Lemma F.5.*** When $i_1 \neq i_2$ and $i_1 + i_2$ are even, we have

$$
\begin{aligned}
\boldsymbol{p}_{i_1}^\top \boldsymbol{p}_{i_2} &= \sum_{j=1}^{M} \sin\left( \frac{j i_1 \pi}{M+1} \right) \sin\left( \frac{j i_2 \pi}{M+1} \right) \\
&= \sum_{j=0}^{M} \sin\left( \frac{j i_1 \pi}{M+1} \right) \sin\left( \frac{j i_2 \pi}{M+1} \right)
\end{aligned}
$$

$$= \frac{1}{4} \sum_{j=0}^{M} \left[ \exp\left( i\pi \frac{i_1 - i_2}{M+1} j \right) + \exp\left( -i\pi \frac{i_1 - i_2}{M+1} j \right) \right.$$

$$\left. - \exp\left( i\pi \frac{i_1 + i_2}{M+1} j \right) - \exp\left( -i\pi \frac{i_1 + i_2}{M+1} j \right) \right]$$

$$= \frac{1}{4} \cdot \frac{\exp\left( i\pi(i_1 - i_2) \right) - 1}{\exp\left( i\pi \frac{i_1-i_2}{M+1} \right) - 1} + \frac{1}{4} \cdot \frac{\exp\left( -i\pi(i_1 - i_2) \right) - 1}{\exp\left( -i\pi \frac{i_1-i_2}{M+1} \right) - 1}$$

$$- \frac{1}{4} \cdot \frac{\exp\left( i\pi(i_1 + i_2) \right) - 1}{\exp\left( i\pi \frac{i_1+i_2}{M+1} \right) - 1} - \frac{1}{4} \cdot \frac{\exp\left( -i\pi(i_1 + i_2) \right) - 1}{\exp\left( -i\pi \frac{i_1+i_2}{M+1} \right) - 1}$$

$$= 0,$$

where the third equation is by $\sin(x) = \frac{\exp(ix) - \exp(-ix)}{2i}$, and the last inequality is by $\exp(i\pi k) = 1$ for even $k$. When $i_1 \neq i_2$ and $i_1 + i_2$ are odd, we have

$$\boldsymbol{p}_{i_1}^\top \boldsymbol{p}_{i_2} = \sum_{j=1}^{M} \sin\left( \frac{ji_1\pi}{M+1} \right) \sin\left( \frac{ji_2\pi}{M+1} \right)$$

$$= \sum_{j=0}^{M} \sin\left( \frac{ji_1\pi}{M+1} \right) \sin\left( \frac{ji_2\pi}{M+1} \right)$$

$$= \frac{1}{4} \sum_{j=0}^{M} \left[ \exp\left( i\pi \frac{i_1 - i_2}{M+1} j \right) + \exp\left( -i\pi \frac{i_1 - i_2}{M+1} j \right) \right.$$

$$\left. - \exp\left( i\pi \frac{i_1 + i_2}{M+1} j \right) - \exp\left( -i\pi \frac{i_1 + i_2}{M+1} j \right) \right]$$

$$= \frac{1}{4} \cdot \frac{\exp\left( i\pi(i_1 - i_2) \right) - 1}{\exp\left( i\pi \frac{i_1-i_2}{M+1} \right) - 1} + \frac{1}{4} \cdot \frac{\exp\left( -i\pi(i_1 - i_2) \right) - 1}{\exp\left( -i\pi \frac{i_1-i_2}{M+1} \right) - 1}$$

$$- \frac{1}{4} \cdot \frac{\exp\left( i\pi(i_1 + i_2) \right) - 1}{\exp\left( i\pi \frac{i_1+i_2}{M+1} \right) - 1} - \frac{1}{4} \cdot \frac{\exp\left( -i\pi(i_1 + i_2) \right) - 1}{\exp\left( -i\pi \frac{i_1+i_2}{M+1} \right) - 1}$$

$$= -\frac{1}{2} \left( \frac{1}{\exp\left( i\pi \frac{i_1-i_2}{M+1} \right) - 1} + \frac{1}{\exp\left( -i\pi \frac{i_1-i_2}{M+1} \right) - 1} \right)$$

$$+ \frac{1}{2} \left( \frac{1}{\exp\left( i\pi \frac{i_1+i_2}{M+1} \right) - 1} + \frac{1}{\exp\left( -i\pi \frac{i_1+i_2}{M+1} \right) - 1} \right)$$

$$= 0,$$

where the third equation is by $\sin(x) = \frac{\exp(ix) - \exp(-ix)}{2i}$, the fifth inequality is by $\exp(i\pi k) = -1$ for odd $k$, and the last equation is by $\frac{1}{\exp(x)-1} + \frac{1}{\exp(-x)-1} = -1$. When $i_1 = i_2$, we have

$$\boldsymbol{p}_{i_1}^\top \boldsymbol{p}_{i_2} = \sum_{j=1}^{M} \sin\left( \frac{ji_1\pi}{M+1} \right) \sin\left( \frac{ji_2\pi}{M+1} \right)$$

$$= \sum_{j=0}^{M} \sin\left( \frac{ji_1\pi}{M+1} \right) \sin\left( \frac{ji_2\pi}{M+1} \right)$$

$$= \frac{1}{4} \sum_{j=0}^{M} \left[ \exp\left( i\pi \frac{i_1 - i_2}{M+1} j \right) + \exp\left( -i\pi \frac{i_1 - i_2}{M+1} j \right) \right.$$

$$\left. - \exp\left( i\pi \frac{i_1 + i_2}{M+1} j \right) - \exp\left( -i\pi \frac{i_1 + i_2}{M+1} j \right) \right]$$

$$= \frac{M+1}{2} - \frac{1}{4}\sum_{j=0}^{M}\left[\exp\left(i\pi\frac{i_1+i_2}{M+1}j\right) + \exp\left(-i\pi\frac{i_1+i_2}{M+1}j\right)\right]$$

$$= \frac{M+1}{2} - \frac{1}{4}\cdot\frac{\exp\left(i\pi(i_1+i_2)\right)-1}{\exp\left(i\pi\frac{i_1+i_2}{M+1}\right)-1} - \frac{1}{4}\cdot\frac{\exp\left(-i\pi(i_1+i_2)\right)-1}{\exp\left(-i\pi\frac{i_1+i_2}{M+1}\right)-1}$$

$$= \frac{M+1}{2},$$

where the third equation is by $\sin(x) = \frac{\exp(ix)-\exp(-ix)}{2i}$, and the last inequality is by $\exp(i\pi k) = 1$ for even $k$. □

## G    ADDITIONAL EXPERIMENTS

### G.1    ADDITIONAL EXPERIMENTS ON RANDOM/DETERMINISTIC WALKS

In this subsection, we provide additional experiments on synthetic data for Task 1 and Task 2 with $(K, N) = (20, 101)$. We consider the transformer model introduced in Section 2 with the length of the position embedding M = 1000. To train the model, we utilize gradient descent starting with zero initialization, where the learning rate $\eta = 1$ and the constant $\epsilon$ in the log-loss is set as $\epsilon = 0.1$. And, we run the gradient descent algorithm for T = 50 training epochs. Figure 8 and Figure 9 illustrate the experiments for Task 1 and Task 2 respectively. These experimental results match Theorem 3.1 and Theorem 3.2, which also strongly supports our theoretical results.

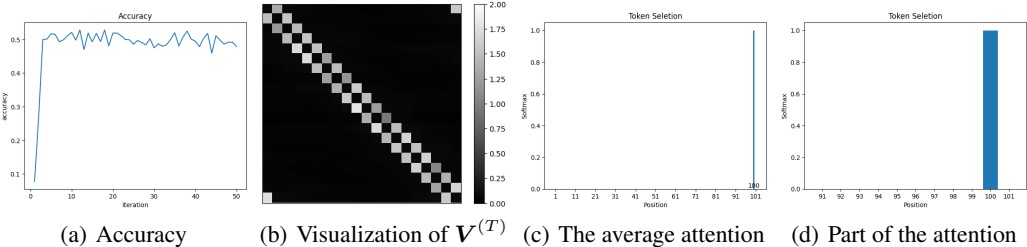

(a) Accuracy          (b) Visualization of $\boldsymbol{V}^{(T)}$   (c) The average attention    (d) Part of the attention

Figure 8: The results of the experiment on Task 1 with $(K, N) = (20, 101)$: (a) is the test accuracy; (b) is the visualization of $\boldsymbol{V}^{(T)}$; (c) and (d) present the average attention of the test data with x-axis representing the position of the token and y-axis representing the attention score.

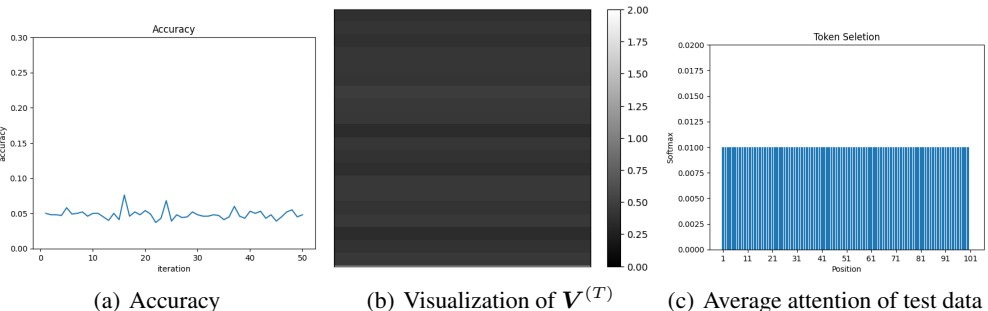

(a) Accuracy              (b) Visualization of $\boldsymbol{V}^{(T)}$        (c) Average attention of test data

Figure 9: The results of the experiment on Task 2 with $K = 20, N = 101$. (a) is the prediction accuracy with $x$-axis representing the iteration and $y$-axis representing the accuracy. (b) is the visualization of $\boldsymbol{V}$. (c) is the average attention of the test data with $x$-axis representing the position of the token and $y$-axis representing the attention score.

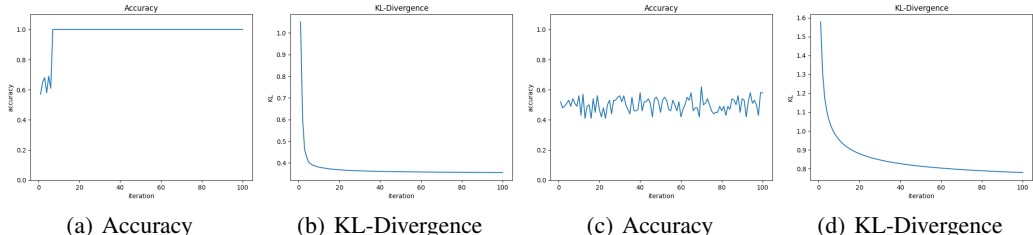

(a) Accuracy     (b) KL-Divergence     (c) Accuracy     (d) KL-Divergence

Figure 10: The results of the experiment conducted using a more complicated transformer for Task 3 and Task 4: (a) and (b) correspond to the experiment for Task 3; (c) and (d) correspond to the experiment for Task 4.

## G.2 ADDITIONAL EXPERIMENTS ON THE QUESTION ANSWERING TASKS IN SECTION 5.2

Here, we conduct some additional experiments for Task 3 and Task 4 discussed in Section 5.2, extending the single-layer transformer model to a more complicated model by adding a fully connected layer with ReLU activation to the transformer model. The new model has the form

$$f_\theta(\boldsymbol{X}) = \boldsymbol{A} \cdot \mathrm{ReLU}\left(\boldsymbol{V}\boldsymbol{X}\mathrm{Softmax}\left(\widetilde{\boldsymbol{X}}^\top \boldsymbol{W}\widetilde{\boldsymbol{x}}_N\right)\right), \tag{G.1}$$

where $\boldsymbol{A} \in \mathbb{R}^{K \times m}$, $\boldsymbol{V} \in \mathbb{R}^{m \times K}$, $\boldsymbol{W} \in \mathbb{R}^{(K+M) \times (K+M)}$ are the trainable parameter matrices, and $m$ is the number of neurons in the fully connected layer. For Task 3 and Task 4, the length of the vocabulary $K$ and the length of each input sequence $N$ are set as $(K, N) = (19, 17), (19, 19)$ respectively. In addition, we set the positional embedding $M = 1000$ and the number of neurons $m = 19$. To train the model, we consider the Gaussian random initialization $\boldsymbol{A}_{ij}^{(0)}, \boldsymbol{V}_{ij}^{(0)}, \boldsymbol{W}_{ij}^{(0)} \sim N(0, \sigma^2)$ with $\sigma = 0.01$, and use gradient descent with learning rate $\eta = 0.1$. The constant $\epsilon$ in the log-loss is set as $\epsilon = 0.1$. Both experiments are conducted on 1024 training data and 1024 test data. Here, most of the settings remain the same as in the previous experiments in Section 5.2.

Figure 10 shows the experiment results using the more complicated transformer in (G.1) to learn Task 3 and Task 4. In Figure 10(a) and Figure 10(c), we present the test accuracy achieved by the transformer model in learning Task 3 and Task 4 respectively. In Figure 10(b) and Figure 10(d), we first normalize the output of the trained transformer model to get a $K$-dimensional vector, representing the prediction distribution of $K$ words. Then, we report the KL-divergence between this prediction distribution and the true distribution of $y|\boldsymbol{x}_1, \boldsymbol{x}_2, ..., \boldsymbol{x}_{N-1}$. The experiment results show a clear difference between the performances of the transformer model in the two tasks. In Task 3, the trained transformer model can successfully approach the optimal accuracy (100%) within 100 iterations. However, in Task 4, the test accuracy always remains around 50%, which is the accuracy of a random guess.

Despite using a more complicated transformer model with an additional feedforward layer of nonlinearities compared to the one considered in our theoretical analysis and previous experiments, the experimental results are still similar to those reported in Section 5.2. These results demonstrate that more complex transformer models may still struggle with the relatively 'simple' Task 4 but excel at the relatively 'difficult' Task 3. This indicates that our findings can be applied to cases involving additional nonlinearities, implying their applicability to more complex and general conditions.

