# OpenReview forum: "Towards Understanding Token Selection in Self-Attention: Successes and Pitfalls in Learning Random Walks"
_ICLR.cc/2025/Conference — Submitted to ICLR 2025_

### Official Review · Reviewer_XcjE · 2024-10-25

**Soundness:** 3
**Presentation:** 4
**Contribution:** 4
**Rating:** 6
**Confidence:** 3

**Summary:**

This paper presents an intriguing observation about self-attention mechanisms. It finds that self-attention excels at learning the Markov property of random walks but struggles with deterministic walks, which are much simpler. This suggests that if all tokens contain similar informative features, self-attention may fail to learn them effectively. The paper also provides robust experiments to support this observation.

**Strengths:**

+ The paper is well-written and easy to follow.
+ The observation is intriguing and sheds light on training strategies and feature selection for transformer-based models.
+ It provides a solid theoretical analysis.
+ The experiments effectively support the findings.

**Weaknesses:**

- Lacks experiments with real-world data.
- Only tests on single-layer self-attention. It would be beneficial to scale up to multi-layer and multi-head self-attention to explore the results further.

**Questions:**

According to the weaknesses.

Similar to a single-layer neural network, where a naive perceptron struggles as a good learner, a deep neural network with multiple linear layers can fit various functions and tasks. Given the prevalence of large language models (LLMs) today, and the shift in focus towards them, I wonder if scaling up self-attention—whether in width (multi-heads) or depth (number of layers)—would still present the same issues identified in this study?

---

> ### Author Response · Authors · 2024-11-23
>
> Thanks for your insightful and supportive comments. In the revised paper, we have added new experimental results to demonstrate that our findings apply to more general cases, including (1) Gaussian initialization instead of zero initialization, and (2) problems beyond random and deterministic walks. Please refer to our response to all reviewers for an overview of the revisions. Below, we address your comments and questions in detail.
>
> >**Q1**: Lacks experiments with real-world data.
>
> **A1**: Our setting is specific in that we consider a random walk task and a deterministic task on a circle, making it challenging to find real-world datasets to satisfy the requirements of the task. However, the insight of our theoretical analysis is general: although there are numerous highly informative tokens, the transformer architecture may yield a bad performance if the average of tokens is not informative. Even though our analysis is conducted on a relatively simple task, we believe this insight can be applied to more general cases. For example, motivated by our insight about random/deterministic walks, we have proposed two NLP tasks in the revised paper (see lines 450-521). The experiment results show that the transformer fails to learn the relatively ‘simple’ Task 4 but can learn the relatively ‘difficult’ Task 3. This phenomenon can be explained by our theoretical finding that the self-attention mechanism struggles in the case that there are multiple highly informative tokens but the average of them is not informative. These results demonstrate that our theories and explanations for random and deterministic walks can guide the construction of various other learning tasks and predict the performance of a transformer model in these tasks.
>
>
>
> >**Q2**: Only tests on single-layer self-attention. It would be beneficial to scale up to multi-layer and multi-head self-attention to explore the results further. Similar to a single-layer neural network, where a naive perceptron struggles as a good learner, a deep neural network with multiple linear layers can fit various functions and tasks. Given the prevalence of large language models (LLMs) today, and the shift in focus towards them, I wonder if scaling up self-attention—whether in width (multi-heads) or depth (number of layers)—would still present the same issues identified in this study?
>
> **A2**: Thank you for your suggestion. We agree that scaling up the width or depth is meaningful and aligns better with the practical setting. However, precise theoretical analysis of such more complicated architectures can be extremely complicated. As a result, most of the recent theoretical studies on the training dynamics of transformers focus on simple one-layer transformer models [1,2,3,4,5]. In fact, to our knowledge, the setting considered in our paper is already more aligned with the practical transformer architecture compared to many of these recent theoretical studies. For example, [1] and [2] conduct the theoretical analysis on the transformer with a linear attention (instead of softmax attention). And, in [3], [4], and [5], the value matrix $V$ is not involved in the training process; instead, the value matrix is held constant while other parameters are trained. In comparison, we construct a transformer architecture with nonlinear softmax attention and analyze the training of the matrices $\mathbf{V}$ and $\mathbf{W}$ simultaneously, which is a step toward more practical study.
>
>
>
>  ---
>
> [1] Ruiqi Zhang, Spencer Frei, and Peter L Bartlett. Trained transformers learn linear models in-context. Journal of Machine Learning Research, 25(49):1–55, 2024.
>
> [2] Anwar, Usman, Johannes Von Oswald, Louis Kirsch, David Krueger, and Spencer Frei. "Adversarial Robustness of In-Context Learning in Transformers for Linear Regression." arXiv preprint arXiv:2411.05189 (2024).
>
>
> [3] Davoud Ataee Tarzanagh, Yingcong Li, Christos Thrampoulidis, and Samet Oymak. Transformers as support vector machines. In NeurIPS 2023 Workshop on Mathematics of Modern Machine Learning, 2023.
>
> [4] Yuchen Li, Yuanzhi Li, and Andrej Risteski. How do transformers learn topic structure: Towards a mechanistic understanding. International Conference on Machine Learning, 2023.
>
> [5] Zihao Li, Yuan Cao, Cheng Gao, Yihan He, Han Liu, Jason Matthew Klusowski, Jianqing Fan, and Mengdi Wang. One-Layer Transformer Provably Learns One-Nearest Neighbor In Context. Advances in Neural Information Processing Systems, 2024.

---

> ### Author Response · Authors · 2024-11-26
> **Looking forward to your feedback**
>
> Dear Reviewer XcjE,
>
> Thank you for your supportive and helpful comments. We have provided detailed responses to your questions and carefully revised the paper. We believe these revisions have significantly improved the paper. In particular, we would like to highlight the following points.
>
> First of all, in our revised paper, we have added discussions on two question answering tasks where the questions are of the forms:
>
> *Based on the list `apple, orange, apple, apple, orange', which type of fruit appears most frequently?*
>
> *Based on the sentence `I prefer an apple to an orange', which type of fruit do I prefer?*
>
> We believe that these additional results address your comment about real data experiments to a certain extent. Please note that the nature of our conclusion is to demonstrate that 'some seemingly simple learning tasks may be challenging for transformers.' Therefore, we believe that constructing such simple learning tasks with a clear practical background is the best way to demonstrate the impact and usefulness of our theory.
>
> In addition, regarding your comment about deeper transformer models, we would like to emphasize that most existing theoretical analyses on transformer training procedures focus on one-layer transformers, and, compared with most existing works, our work addresses a setting that is arguably closer to practical applications. We believe that our insights and proof techniques can help bridge the gap between theoretical studies and real-world applications.
>
> We are happy to answer any further questions you may have. Thank you once again for reviewing our paper.
>
> Best regards,
>
> Authors of Submission9023

---

> > ### Comment · Reviewer_XcjE · 2024-12-02
> > **Response to rebuttal**
> >
> > Thank you for your response. I will keep a positive score for this work.

---

### Official Review · Reviewer_G3VT · 2024-10-29

**Soundness:** 4
**Presentation:** 4
**Contribution:** 3
**Rating:** 8
**Confidence:** 4

**Summary:**

The current work introduces two case studies that highlight a rather counter-intuitive phenomena — that transformer models can effectively learn to predict the next token in a simple random walk task along a graph, but fail to learn this task when the walk is deterministic. The authors put forward theoretical reasons as to why this is the case and proceed to empirically show that their theory holds. The theoretical argument is elegant and makes an initially puzzling finding intuitively obvious, which I really liked.

**Strengths:**

1. The paper is overall well written and easy to follow (except for maybe Section 2.1, more later).
2. The authors give a theoretical account of the phenomena they are studying, complete with proofs supporting their point (though I only skimmed them, so cannot vouch for their correctness). Additionally, they give an intuitive, non-formal argument for why transformes succeed and fail in their test cases, which I find convincing.
3. The authors support their conclusion with empirical tests, though I think they could — and should do — more (see below).
4. The previous literature is well surveyed, though I admit that is not my primary area of expertise, so I am unlikely to know if there is anything missing.

**Weaknesses:**

There are two main issues in my opinion with this article:

1. The insight the authors gain from their test cases are interesting, but it their proof is based on a particular assumption — initialisation of the attention matrices to zero. How often does this happen in practice though? I understand that other assumptions such as aggregating key and query matrices into a single one to facilitate their analysis, but the other one seems a bit artificial. While this doesn’t invalidate the insight, it would be good to either see:
    1. Experimental evidence for what happens if they use standard initialisations
    2. A good reason as to why this is not needed beyond “it makes our analysis easier”.
2. Related to the above, while the work is well situated within it’s literature, I am not convinced on how this is “interesting”. To what does this novel insight apply? Is there a specific domain or task where the authors believe this is a problem? If it is, have other researchers proposed a solution, in which case this work would be an explanation for those issues (which is good in my opinion)?

**Questions:**

My questions are embedded within the weaknesses.

246 - “series” instead of “serious”

There might be more, I am terrible at spotting these.

As it stands, I recommend reject. But I think the issues are fixable, mostly by addressing the concerns outlined above.

---

> ### Author Response · Authors · 2024-11-23
>
> Thank you for your detailed comments. We have updated the paper based on your suggestions, including new experiments on (1) Gaussian initialization instead of zero initialization, and (2) problems beyond random and deterministic walks. For a summary of the revisions, please refer to our response to all reviewers. We address your concerns as follows.
> >**Q1**: Experimental evidence for what happens if they use standard initializations. A good reason as to why this is not needed beyond “it makes our analysis easier”.
>
> **A1**: Thanks for your suggestion. We have added the experiments with Gaussian random initialization to the revised paper (lines 370-400). The experimental results indicate that the transformer learns the true prediction distribution of deterministic walks much slower than learning that of random walks, which clearly demonstrates that Task 2 for learning deterministic walks is significantly more challenging even with random initialization.
>
>
> >**Q2**: Related to the above, while the work is well situated within its literature, I am not convinced on how this is “interesting”. To what does this novel insight apply? Is there a specific domain or task where the authors believe this is a problem? If it is, have other researchers proposed a solution, in which case this work would be an explanation for those issues (which is good in my opinion)?
>
> **A2**: We demonstrate that training a one-layer transformer model on the deterministic walk task leads to failure, yielding a poor performance no better than a random guess. This finding reveals a potential limitation of self-attention: even with numerous highly informative tokens, the transformer architecture may struggle due to insufficient information provided by the average of tokens. To our knowledge, this observation appears to be novel in the current literature, and we believe this insight can have broader implications in various scenarios. For example, motivated by our insight about random/deterministic walks, we have proposed two NLP tasks in the revised paper (see lines 450-521). The experiment results show that the transformer fails to learn the relatively ‘simple’ Task 4 but can learn the relatively ‘difficult’ Task 3. This phenomenon can be explained by our theoretical finding that the self-attention mechanism struggles in the case that there are multiple highly informative tokens but the average of them is not informative. These results demonstrate that our theories and explanations for random and deterministic walks can guide the construction of various other learning tasks and predict the performance of a transformer model in these tasks.
>
>
>
> >**Q3**: Typo
>
> **A3**: Thanks for pointing out the typo. We have revised the paper accordingly.

---

> ### Comment · Reviewer_G3VT · 2024-11-23
>
> I thank the authors for addressing my comments. I think the paper is much stronger now and have thus updated my score accordingly. I believe this is an interesting finding that points to some quirks of the attention mechanisms and thus I recommend accepting it.

---

> > ### Author Response · Authors · 2024-11-24
> >
> > Dear Reviewer G3VT,
> >
> > Thank you for acknowledging the significant improvement of our paper and for raising your score! Your comments and suggestions have greatly helped us in revising the paper, and we truly appreciate it. If you have any further suggestions, please let us know.
> >
> > Best regards,
> >
> > Authors of Submission9023

---

### Official Review · Reviewer_nbrK · 2024-11-02

**Soundness:** 2
**Presentation:** 2
**Contribution:** 2
**Rating:** 3
**Confidence:** 3

**Summary:**

The paper investigates the ability of the self-attention mechanism in transformers to perform token selection by examining its performance in learning two types of sequential data: random walks vs deterministic walks on circles. The authors theoretically demonstrate that a single-layer self-attention mechanism can successfully learn the transition probabilities of random walks but fails to predict deterministic walks. This contrast reveals a limitation of self-attention when dealing with multiple equally informative tokens.

**Strengths:**

- The paper provides a theoretically rigorous examination of token selection by self-attention in transformers. However, analysis is limited to a very specific implementation in a simplified model with a single attention layer.
- If confirmed in more complex architectures and outside of toy problems, the results could be important.

**Weaknesses:**

- The paper's focus on one-layer transformers and a single overly simple toy task limits its generalizability to more complex and realistic scenarios involving deep transformers. It is not clear whether the effect observed would arise in more complex settings.
- The initialization of all weights to zero seems to be the main cause for the problem, since it would break the symmetry in the initial softmax weighting (as per step 1 of "training dynamics in learning deterministic walks".
    Is there a particular reason why weights were initialized to zero?

- A more thorough validation with a wider range of token representations and architectures is required to support the conclusions of the paper.

- Minor: positional embeddings were concatenated here, but in practical applications they are typically added element-wise; I am not sure if this would have an impact on the observed phenomenon.

**Questions:**

See above in Weaknesses.

---

> ### Author Response · Authors · 2024-11-23
>
> We appreciate your constructive comments and have updated the paper based on your suggestions. Specifically, we have added new experimental results to demonstrate that our findings apply to more general cases, including (1) Gaussian initialization instead of zero initialization, and (2) problems beyond random and deterministic walks. For a summary of the revisions, please refer to our response to all reviewers. We address your questions as follows. Due to character limits, we separate our response into two parts (references are given in the second part of our response).
>
> >**Q1**: The paper's focus on one-layer transformers and a single overly simple toy task limits its generalizability to more complex and realistic scenarios involving deep transformers. It is not clear whether the effect observed would arise in more complex settings.
>
> **A1**: Although our theoretical analysis is based on a simple task, the underlying insight is applicable across various scenarios. Our analysis reveals that despite the presence of many highly informative tokens, the performance of the transformer may suffer if the average informativeness of the tokens is low. We believe that this insight can be observed in other tasks as well. For example, we have proposed two NLP tasks in the revised paper (see lines 450-521). The experiment results show that the transformer fails to learn the relatively ‘simple’ Task 4 but can learn the relatively ‘difficult’ Task 3. This phenomenon can be explained by our theoretical finding that the self-attention mechanism struggles in the case that there are multiple highly informative tokens but the average of them is not informative. These results demonstrate that our theories and explanations for random and deterministic walks can guide the construction of various other learning tasks and predict the performance of a transformer model in these tasks.
>
> Your suggestion about deep transformer architecture is insightful, but it seems to be beyond the scope of our paper. We would like to point out that existing theoretical analysis on the training dynamics of transformers mostly mainly focus on a single self-attention layer ([1], [2], [3], [4]). Studying similar problems with more complicated data structures and more complex transformer architecture could be an important future direction.
>
>
>
> >**Q2**: The initialization of all weights to zero seems to be the main cause for the problem since it would break the symmetry in the initial softmax weighting (as per step 1 of "training dynamics in learning deterministic walks". Is there a particular reason why weights were initialized to zero?
>
> **A2**: Our theoretical analysis focuses on zero initialization as it simplifies our analysis. When we use small random initialization to train the model, intuitively, the result of learning deterministic walks may not be as bad as the zero initialization case, but we can still expect that the training for deterministic walks is more difficult and the performance is worse than that for random walks.
>
> We have added the experiments with Gaussian random initialization to the revised paper (see lines 370-400). The experimental results indicate that the transformer learns the true prediction distribution of deterministic walks much slower than learning that of random walks, which clearly demonstrates that Task 2 for learning deterministic walks is significantly more challenging even with random initialization.
>
>
>
> >**Q3**: A more thorough validation with a wider range of token representations and architectures is required to support the conclusions of the paper.
>
> **A3**: Thanks for your suggestion. We are sure that our findings can be extended to any token representations where different states are represented by orthogonal vectors, such as one-hot encoding and the encoding corresponding to sine and cosine functions (as proposed in [5]).
>
> In terms of architecture, we would like to acknowledge the difficulty and complexity of precisely analyzing the training dynamics of more complicated transformer architectures. As a result, most of the recent theoretical studies on the training dynamics of transformers focus on simple one-layer transformer models [4,6,7,8,9]. In fact, to our knowledge, the setting considered in our paper is already more aligned with the practical transformer architecture compared to many of these recent theoretical studies. For example, [6] and [7] conduct the theoretical analysis on the transformer with a linear attention (instead of softmax attention). And, in [4], [8], and [9], the value matrix $\boldsymbol{V}$ is not involved in the training process; instead, the value matrix is held constant while other parameters are trained. In comparison, we construct a transformer architecture with nonlinear softmax attention and analyze the training of the matrices $\boldsymbol{V}$ and $\boldsymbol{W}$ simultaneously, which is a step toward more practical study.

---

> ### Author Response · Authors · 2024-11-23
>
> >**Q4**: Minor: positional embeddings were concatenated here, but in practical applications, they are typically added element-wisely; I am not sure if this would have an impact on the observed phenomenon.
>
> **A4**: Concatenated positional embeddings can significantly simplify the complexity of theoretically analyzing the transformer model, which is the reason why we utilize this kind of embedding. And, we would like to point out that concatenated positional embeddings have been utilized in most theoretical studies, such as [4], [10], [11], and [12].
>
> Although our precise theoretical analysis relies on concatenated positional embeddings, we would like to point out that the insight provided by our study can be applied to the case where positional embeddings are added element-wisely. In our revised paper (lines 408-448), we have discussed that the key reason the transformer fails to learn deterministic walks efficiently is that the token average is not informative. When positional embeddings are added element-wisely, it is still true that *all deterministic walks will give exactly the same token average, and therefore the token average is not informative in the prediction task.*
>
> ---
>
>
> [1] Davoud Ataee Tarzanagh, Yingcong Li, Xuechen Zhang, and Samet Oymak. Max-margin token selection in attention mechanism. Advances in Neural Information Processing Systems, 36:48314–48362, 2023.
>
> [2] Alberto Bietti, Vivien Cabannes, Diane Bouchacourt, Herve Jegou, and Leon Bottou. Birth of a transformer: A memory viewpoint. Advances in Neural Information Processing Systems, 36, 2024.
>
> [3] Yuandong Tian, Yiping Wang, Beidi Chen, and Simon S Du. Scan and snap: Understanding training dynamics and token composition in 1-layer transformer. Advances in Neural Information Processing Systems, 36:71911–71947, 2023.
>
> [4] Zihao Li, Yuan Cao, Cheng Gao, Yihan He, Han Liu, Jason Matthew Klusowski, Jianqing Fan, and Mengdi Wang. One-Layer Transformer Provably Learns One-Nearest Neighbor In Context. Advances in Neural Information Processing Systems, 2024.
>
> [5] Ashish Vaswani, Noam Shazeer, Niki Parmar, Jakob Uszkoreit, Llion Jones, Aidan N Gomez, Lukasz Kaiser, and Illia Polosukhin. Attention is all you need. Advances in Neural Information Processing Systems, 2017.
>
> [6] Ruiqi Zhang, Spencer Frei, and Peter L Bartlett. Trained transformers learn linear models in-context. Journal of Machine Learning Research, 25(49):1–55, 2024.
>
> [7] Usman Anwar, Johannes Von Oswald, Louis Kirsch, David Krueger, and Spencer Frei. "Adversarial Robustness of In-Context Learning in Transformers for Linear Regression." arXiv preprint arXiv:2411.05189 (2024).
>
>
> [8] Davoud Ataee Tarzanagh, Yingcong Li, Christos Thrampoulidis, and Samet Oymak. Transformers as support vector machines. In NeurIPS 2023 Workshop on Mathematics of Modern Machine Learning, 2023.
>
> [9] Yuchen Li, Yuanzhi Li, and Andrej Risteski. "How do transformers learn topic structure: Towards a mechanistic understanding." International Conference on Machine Learning. 2023.
>
> [10] Zixuan Wang, Stanley Wei, Daniel Hsu, and Jason D. Lee. "Transformers Provably Learn Sparse Token Selection While Fully-Connected Nets Cannot." International Conference on Machine Learning, 2024
>
> [11] Eshaan Nichani, Alex Damian, and Jason D. Lee. "How Transformers Learn Causal Structure with Gradient Descent." International Conference on Machine Learning, 2024
>
> [12] Yu Bai, Fan Chen, Huan Wang, Caiming Xiong, and Song Mei. "Transformers as statisticians: Provable in-context learning with in-context algorithm selection." Advances in Neural Information Processing Systems, 2024.

---

> ### Author Response · Authors · 2024-11-26
> **Looking forward to your feedback**
>
> Dear Reviewer nbrK,
>
> Thank you for your time and effort in reviewing our paper. We believe that our response and revision have addressed your concerns. We would greatly appreciate it if you could review our response and revision. Here, we would like to particularly highlight the following points:
>
> - To address your concern that “the initialization of all weights to zero seems to be the main cause of the problem”, we have added additional experiments using Gaussian random initialization (see Figure 5 in the revised paper). The results demonstrate that **even with Gaussian random initialization, deterministic walks remain more challenging to learn with transformers compared to random walks**. This observation aligns well with our theoretical findings.
>
> - In response to your concerns about our simplified problem setting and questions about extensions to other settings, we would like to emphasize that, as a paper focusing on theoretical analysis, it is necessary to consider a relatively clean setting. As we mentioned in our earlier response, our work already addresses a setting that is arguably closer to practice than many existing theoretical studies, and **our proof techniques can help advance theoretical studies towards more practical settings**. In the revision, we have included experiments on two new question-answering tasks (see Section 5.2 in the revised paper). Our results demonstrate that **the insights gained from studying random/deterministic walks can help predict the performance of transformers in other learning tasks as well**. We believe that conducting rigorous theoretical analysis in a clean setting and providing insights applicable to other settings is precisely what a theory paper should aim to do, and our paper accomplishes this.
>
> We are confident that, thanks to your constructive feedback, our revised paper is now of much higher quality. Therefore, we sincerely hope you can review our response and the revised paper, and reconsider your evaluation in light of the points mentioned above.
>
> Thank you.
>
> Best regards,
>
> Authors of Submission9023

---

> ### Author Response · Authors · 2024-11-30
>
> Dear Reviewer nbrK,
>
> We have not received your response since we submitted our replies to your original review. We are confident that our response and revision have addressed your concerns, and we are eager to know whether you have any additional questions. As the discussion period is ending soon, we would greatly appreciate it if you could review our responses and let us know if you have any further questions. Thank you.
>
> Best regards,
>
> Authors of Submission9023

---

> ### Author Response · Authors · 2024-12-02
>
> Dear Reviewer nbrK,
>
> Apologies for our repeated messages. Since the deadline for you to give us your feedback is only one day away, we sincerely hope you can reevaluate our paper based on our responses and revisions.
>
> In your original review, your concerns were mainly about the simplicity of the setting we considered. To address these concerns, we have added experiments on (1) learning of random/deterministic walks with random initialization, (2) extensions to other learning problems in NLP, and (3) extensions to transformer models with additional nonlinearities (both (2) and (3) also use random initializations). We believe these new results fully address your concerns.
>
> We are confident that our revised paper is much stronger, and we truly hope the improvements can be recognized. We would also like to reemphasize that conducting rigorous theoretical analysis in a clean setting and providing insights applicable to other settings is precisely what a theory paper should aim to do, and we believe our paper accomplishes this.
>
> Thank you.
>
> Best regards,
>
> Authors of Submission9023

---

### Official Review · Reviewer_tPAR · 2024-11-04

**Soundness:** 3
**Presentation:** 3
**Contribution:** 2
**Rating:** 6
**Confidence:** 4

**Summary:**

This paper analyzes the gradient dynamics of one-layer transformer (with parameters V and W, which is self-attention pairwise logits) on predicting the next state of Markov chains in two synthetic cases (1) when the Markov chain is a random walk on a circular graph and (2) when the Markov chain is a deterministic walk (either clockwise or counter-clockwise). The conclusion is surprising: for random walk the Transformer is able to fully model the transition matrix of Markov chain (in V), and the prediction accuracy is optimal, while for deterministic walk, the prediction is random and V converges to all 1 matrix (Theorem 3.2). The paper also performs experiments to justify the results.

**Strengths:**

1. The paper performs rigorous mathematical study to analyze gradient dynamics in the specific Markov cases.
2. The paper is relatively well-written.

**Weaknesses:**

1. I am a bit skeptical about whether Theorem 3.2 is empirically meaningful. The initial condition in the theoretical analysis is W = V = 0, which may lead to the all 1 matrix V in Theorem 3.2. What if the symmetry is broken and there is some small initial noise of W and V? In that case, will the transformer converge to something similar to Theorem 3.1? I checked the appendix and it looks like the Theorem 3.2 is indeed due to perfect symmetry in attention scores (Lemma C.3) and the gradient of V (Lemma C.2). Note that Task 1 already has noise in its input/output relationship, which Task 2 does not have. This may lead to sharp contrast between Theorem 3.1 and 3.2, which is not an issue empirically.

If Theorem 3.2 is purely because the symmetry initialization, then it would hurt the generalization of the main conclusion. If theoretical study is non-trivial with random initialization, authors can also use experiments to demonstrate that the conclusion still holds with small random initialization.

2. How does this analysis extend to more general cases? Can it handle more general Markov chains? What if there is FFN and nonlinearity on top of the attention layer?

**Questions:**

1. Could the authors investigate how small random initializations of W and V affect the convergence behavior described in Theorem 3.2? This would help clarify whether the observed behavior is solely due to the symmetric initialization or if it persists under more realistic conditions.

2. Could the authors comment on how their analysis might extend to more complex Markov chains, such as those with large and compositional state/action spaces, or non-uniform transition probabilities? Additionally, it would be helpful if they could discuss the potential impact of adding nonlinearities or feed-forward layers to the transformer architecture on their theoretical results.

------
After the discussion, I raised the score from 5 to 6.

---

> ### Author Response · Authors · 2024-11-23
>
> Thank you for your detailed comments. We have revised the paper according to your suggestions and added new experiment results on more general settings. Please refer to our response to all reviewers for a summary of the changes made in the revision. Below, we address your concerns and questions in detail.
>
> >**Q1**: I am a bit skeptical about whether Theorem 3.2 is empirically meaningful. The initial condition in the theoretical analysis is W = V = 0, which may lead to the all 1 matrix V in Theorem 3.2. What if the symmetry is broken and there is some small initial noise of W and V? In that case, will the transformer converge to something similar to Theorem 3.1?
>
> **A1**: Thanks for your suggestion. We have added the experiments with Gaussian random initialization to the revised paper (see lines 370-400). We recognize that the experimental results can not perfectly match our theoretical analysis with zero initialization as stated in Theorem 3.2. However, the experiment results indicate that the transformer learns the true prediction distribution of deterministic walks much slower than learning that of random walks, which still clearly demonstrates that Task 2 for learning deterministic walks is significantly more challenging even with random initialization.
>
> >**Q2**: How does this analysis extend to more general cases? Can it handle more general Markov chains? What if there is FFN and nonlinearity on top of the attention layer?
>
> **A2**: The practical insight of our theoretical analysis is that while there are numerous highly informative tokens, the transformer architecture could still yield a bad performance as long as the average of tokens is not informative. Despite our analysis focusing on a relatively simple task, we suggest that this insight can be extended to broader scenarios. For example, motivated by our insight about random/deterministic walks, we have proposed two NLP tasks in the revised paper (see lines 450-521). The experiment results show that the transformer fails to learn the relatively ‘simple’ Task 4 but can learn the relatively ‘difficult’ Task 3. This phenomenon can be explained by our theoretical finding that the self-attention mechanism struggles in the case that there are multiple highly informative tokens but the average of them is not informative. These results demonstrate that our theories and explanations for random and deterministic walks can guide the construction of various other learning tasks and predict the performance of a transformer model in these tasks.
>
> We appreciate your suggestion regarding FFN and additional nonlinear layers. However, it seems to be beyond the scope of our paper. We would like to point out that existing theoretical analysis on the training dynamics of transformers mostly mainly focus on a single self-attention layer ([1], [2], [3], [4]). Studying similar problems with more complicated data structures and more complex transformer architectures could be an important future direction.
>
> ---
>
> [1] Davoud Ataee Tarzanagh, Yingcong Li, Xuechen Zhang, and Samet Oymak. Max-margin token selection in attention mechanism. Advances in Neural Information Processing Systems, 36:48314–48362, 2023.
>
> [2] Alberto Bietti, Vivien Cabannes, Diane Bouchacourt, Herve Jegou, and Leon Bottou. Birth of a transformer: A memory viewpoint. Advances in Neural Information Processing Systems, 36, 2024.
>
> [3] Yuandong Tian, Yiping Wang, Beidi Chen, and Simon S Du. Scan and snap: Understanding training dynamics and token composition in 1-layer transformer. Advances in Neural Information Processing Systems, 36:71911–71947, 2023.
>
> [4] Zihao Li, Yuan Cao, Cheng Gao, Yihan He, Han Liu, Jason Matthew Klusowski, Jianqing Fan, and Mengdi Wang. One-Layer Transformer Provably Learns One-Nearest Neighbor In Context. Advances in Neural Information Processing Systems, 2024.

---

> ### Author Response · Authors · 2024-11-26
> **Looking forward to your feedback**
>
> Dear Reviewer tPAR,
>
> Thank you for your helpful and constructive review. We have carefully addressed your concerns and questions in our response and revision.
>
> In particular, following your suggestion, we have added experiments with Gaussian random initialization (see Figure 5 in the revised paper), and demonstrate that transformers struggle in learning deterministic walks even with small random initialization.
>
> Moreover, regarding your question about extensions to more complex settings, we have also added discussions and experiments beyond random/deterministic walks to demonstrate that our insight can be extended to other settings. Specifically, we have proposed two new question answering tasks (please refer to Section 5.2 in the revised paper) and conducted experiments to study the performance of a one-layer transformer. Our experiment results demonstrate that the insights obtained from studying random/deterministic walks can guide us to predict the performance of a transformer model in various other learning tasks.
>
> Please also refer to our general response to all reviewers, where we provide an overview of the major changes made in the revision. We are confident that our revised paper is much stronger than the previous version, and we are eager to hear back from you. If you have any additional questions, please let us know. Thank you.
>
> Best regards,
>
> Authors of Submission9023

---

> > ### Comment · Reviewer_tPAR · 2024-11-26
> > **Thanks for your rebuttal**
> >
> > Thanks for your detailed explanation.
> >
> > 1. The fact that "transformer learns the true prediction distribution of deterministic walks much slower than learning that of random walks" suggests that $W=V=0$ is likely a saddle point in the optimization landscape and with sufficient noise you can escape from that. Do you have more theoretical analysis to show that this is true?
> >
> > 2. "self-attention mechanism struggles in the case that there are multiple highly informative tokens but the average of them is not informative." This is a typical behavior of linear model. Is that because you are only considering one-layer transformer without nonlinearity? What if you impose some nonlinearity? Will this problem be addressed?
> >
> > Overall I still feel that the theoretical analysis in its current form, is a bit straightforward. More discussions about the two cases above would be great. If the authors have them then I will raise my score.

---

> > > ### Author Response · Authors · 2024-11-28
> > >
> > > Dear Reviewer tPAR,
> > >
> > > Thank you for your comments. We address your further concerns as follows.
> > >
> > > > The fact that "transformer learns the true prediction distribution of deterministic walks much slower than learning that of random walks" suggests that $\boldsymbol{W} = \boldsymbol{V} = \boldsymbol{0}$ is likely a saddle point in the optimization landscape and with sufficient noise you can escape from that. Do you have more theoretical analysis to show that this is true?
> > >
> > > We would like to point out that $\boldsymbol{W}=\boldsymbol{V}=\boldsymbol{0}$ is not a saddle point, even for learning deterministic walks. Saddle points, by definition, are points where the gradient of the loss function is zero. However, please note that, at $\boldsymbol{W}=\boldsymbol{V}=\boldsymbol{0}$, the gradients of $\boldsymbol{W}$ and $\boldsymbol{V}$ are both non-zero. Therefore, $\boldsymbol{W}$ and $\boldsymbol{V}$ are not saddle points.
> > >
> > > Our theory actually shows that, through the training of gradient descent, $||\boldsymbol{W}||\_{F}$ and $||\boldsymbol{V}||\_{F}$ will both grow and diverge to infinity. However, along the training path, the softmax scores on all tokens remain balanced, and $\boldsymbol{V}$ is always proportional to the all-one matrix when learning deterministic walks. In our paper, we have carefully avoided using the term ''saddle points’’ to describe the point $ \boldsymbol{W} = \boldsymbol{V} = \boldsymbol{0}$ .
> > >
> > >
> > > > "self-attention mechanism struggles in the case that there are multiple highly informative tokens but the average of them is not informative." This is a typical behavior of linear model. Is that because you are only considering one-layer transformer without nonlinearity? What if you impose some nonlinearity? Will this problem be addressed?
> > >
> > >
> > > In response to your question, in our latest revision, we have added experiments on a more complicated transformer model with an additional fully connected layer with ReLU activation function (see lines 1944-1986 in the revised paper). We test the performance of this more complicated transformer model on the question answering tasks (Task 3 and Task 4) we discussed in Section 5.2 by training the model with gradient descent starting from Gaussian random initialization. We can observe that the results are still similar to those reported in Section 5.2, which demonstrate that more complex transformer models may still struggle with the relatively ‘simple’ Task 4 but excel at the relatively ‘difficult’ Task 3. This demonstrates that our theoretical findings can be applied to cases involving additional nonlinearities.
> > >
> > >
> > > > Overall I still feel that the theoretical analysis in its current form, is a bit straightforward.
> > >
> > >
> > > We would like to highlight our technical contributions and strengths as follows:
> > >
> > > - We understand that our analyses and discussions on deterministic walks provide clear insights and are easy to follow. However, we believe that this clarity should not be dismissed as ‘straightforward’. Providing clear insights should be considered a strength of our paper, not a weakness. We also hope that our clarification above, demonstrating that $\boldsymbol{W} = \boldsymbol{V} = \boldsymbol{0}$ is not a saddle point, can convince you that our analysis is not straightforward.
> > >
> > > - Importantly, please do not overlook that our paper also provides positive guarantees for transformers to learn random walks (Theorem 3.1). Our results not only demonstrate that the prediction accuracy will be optimal but also clearly characterize how the value matrix $\boldsymbol{V}$ and the softmax score vector $\mathcal{S}$ function in a well-trained transformer. These results are highly non-trivial. We are confident that even if we were to remove all analyses on deterministic walks and only present the guarantees in Theorem 3.1 for random walks, our paper would still stand as a strong theoretical contribution, particularly due to our precise analysis (please also refer to our earlier responses to you regarding comparisons with existing theoretical analyses).
> > >
> > >
> > > We believe that our response above addresses your remaining concerns, and we sincerely hope that you can reconsider your evaluation of our paper taking the points above into consideration. Thank you.
> > >
> > > Best regards,
> > >
> > > Authors of Submission9023

---

> > > ### Author Response · Authors · 2024-11-30
> > >
> > > Dear Reviewer tPAR,
> > >
> > > We are writing to follow up on our previous discussion. We hope that our earlier response has addressed your concerns. We are particularly confident in the theoretical contribution of our study, and we are willing to address any additional questions you may have. Thank you.
> > >
> > > Best regards,
> > >
> > > Authors of Submission9023

---

> > > > ### Comment · Reviewer_tPAR · 2024-12-03
> > > > **Thanks for your update.**
> > > >
> > > > > Our theory actually shows that, through the training of gradient descent, and W and V will both grow and diverge to infinity. However, along the training path, the softmax scores on all tokens remain balanced.
> > > >
> > > > What I mean by "saddle" is just like that. Module this norm-growing direction, the dynamics of W and V stay on this "ridge" where softmax scores are all balanced. But if there is any deviation from the perfect balance, then it will learn something similar to random Markov walk. The slowness is due to that it takes time to move away from the ridge, which can be arbitrarily long in the worst case.
> > > >
> > > > If you can characterize the saddle point in a mathematically rigorous manner, it would be great. For now given the contribution, I will raise the score to 6.

---

> > > > > ### Author Response · Authors · 2024-12-03
> > > > >
> > > > > Dear Reviewer tPAR,
> > > > >
> > > > > Thank you for clarifying your questions and for raising your score. You are correct that, in the case of zero initialization, the dynamics of $\boldsymbol{W}$ and $\boldsymbol{V}$ stay on a ‘ridge’ where the softmax scores are balanced. Our experiments demonstrate that with small random initialization, the weights will indeed move away from the ‘ridge’. However, in the experiments, this deviation from the ‘ridge’ can be slow, leading to worse performance in learning deterministic walks compared to learning random walks. We believe that mathematically characterizing the time it takes for gradient descent to move away from the ‘ridge’ is a challenging but important future research direction. We will add discussions about this in our camera-ready version.
> > > > >
> > > > > Best regards,
> > > > >
> > > > > Authors of Submission9023

---

### Author Response · Authors · 2024-11-23
**General Response to All Reviewers**

Dear Reviewers,

We deeply appreciate your time and effort in reviewing our paper. We have revised the paper according to your comments, and thanks to your insightful and constructive feedback, we believe that our revised paper is much stronger. Here, we would like to provide an overview of the major changes we made in the revision to address your common questions.

**Experiments with random initialization**

We have added experiments with Gaussian random initialization to the revised paper (see lines 370-400). In the experiments, for random walks, the transformer achieves near-optimal accuracy after approximately 400 training iterations, while for deterministic walks, it does not achieve near-optimal accuracy even after 1000 iterations. These results demonstrate that, although the transformer performs better with random initialization compared to zero initialization, training remains significantly more challenging for deterministic walks than for random walks. Therefore, our theory and conclusion are still relevant in more practical settings with random initialization.

**New results on simple question answering tasks in NLP**

We have renamed the section “Successes & Pitfalls in Learning Random/Deterministic Walks” to “Successes & Pitfalls **Beyond** Random/Deterministic Walks”, and moved this section after the experiment section. In this revised section, we clarified the main reason the transformer performs relatively poorly in learning deterministic walks. More importantly, we added experimental results on two simple NLP tasks that were constructed based on the insights obtained from studying random/deterministic walks. These two tasks are:

---

**Task 3.** The question answering task covers possible questions of the form

*Based on the list `apple, orange, apple, apple, orange', which type of fruit appears most frequently?*

Here, the list stated in the question can be any combination of 'apple' and 'orange' with a fixed length of 5. Therefore, there are a total of $32$ possible questions the model may see, and each of these questions occur with probability $1/32$. The correct response is the fruit that appears most frequently in the list.

---

**Task 4.** There are only two possible questions

*Based on the sentence `I prefer an apple to an orange', which type of fruit do I prefer?*

*Based on the sentence `I prefer an orange to an apple', which type of fruit do I prefer?*

Here, each of the two questions above occurs with probability $1/2$. The correct response is 'apple' for the first question above, and 'orange' for the second question above.

---

Comparing these two 'NLP' tasks, we observe that in Task 3, no single word can determine the answer; instead, we must combine all five words in the list to solve the question. In contrast, in Task 4, the single word in the 8th or 11th position can uniquely determine the answer. Thus, Task 4 can be naturally considered a 'simpler' task and easier to learn. However, our experiment results (where the transformers are trained with Gaussian random initialization) show that the transformer fails to learn the relatively 'simple' Task 4 but can learn the relatively 'difficult' Task 3.

This surprising result can be explained by our insights from studying random/deterministic walks: In Task 3, the average of the word embeddings in a question can still help the model find the correct response. In contrast, in Task 4, the two questions produce the *same* average of word embeddings, rendering it uninformative for answering the question. As a result, the transformer struggles to learn Task 4 for the same reason it struggles  to learn deterministic walks.


We have adjusted the third contribution bullet in the revised paper accordingly, and moved the additional related work section and part of the old “Successes & Pitfalls in Learning Random/Deterministic Walks” section to the appendix to save space.

We are confident that, thanks to your helpful comments, our revised paper is of much higher quality. We sincerely hope you can check whether our revisions and responses have addressed your questions and concerns.

Thank you!

Best regards,

Authors of Submission9023

---

### Meta-Review · Area_Chair_xkFL · 2024-12-23

**Metareview:**

This paper shows that a one-layer self-attention-based network can learn the transition matrix of one-dimensional Brownian motion but cannot learn if the transition matrix is deterministic. The reviewers are concerned that the mathematical setting of the paper is too narrow to understand whether these findings generalize to a broader class of data and architectures. I would encourage the authors to work further in this direction, because the problem of ascertaining what kinds of stochastic processes can be learned effectively by transformer architectures is certainly important. I would also like to point out that connecting the experiments in Tasks 3 and 4 to the mathematical set up of this paper is very difficult, it is not clear whether Task 3 is more/less “difficult” than Task 4, it is important to make such statements mathematically precise.

**Additional Comments On Reviewer Discussion:**

Reviewer tPAR said that the mathematical proof rested on assuming that the weights for key/query and value are initialized to zero. This diminishes the merits of this work. The second important concern is whether this result has something to say about general Markov chains. The authors have added new toy experiments to analyze these questions, but since this is primarily a theoretical paper, it is not clear whether the concerns of the reviewer are assuaged.

Reviewer nbrK had largely the same concerns as Reviewer tPAR (single layer transformer is a very simplistic setting, initializing weights to zero is a very special condition and any deviation from this would change the mathematical result). In their response, the authors gave references to a lot of existing papers that also study single layer transformers.

Reviewer G3VT had similar points as the other two reviewers. But their score was exceedingly high, 8/10. I am going to discount this score a bit to calibrate it against the scores of other reviewers with the same concerns.

Reviewer XcjE was concerned about the fact that there are no experiments in the paper on real data. The authors rectified this by adding a toy NLP task.

Altogether, I am in agreement with the reviewers that the mathematical setting of the paper is too narrow to understand whether these findings generalize to a broader class of data and architectures.

---

### Decision · Program_Chairs · 2025-01-22

Reject